# A non-canonical role of the inner kinetochore in regulating sister-chromatid cohesion at centromeres

Lu Yan[1,7], Xueying Yuan[1,7], Mingjie Liu[2,7], Qinfu Chen [3], Miao Zhang[1], Junfen Xu[3], Ling-Hui Zeng[4], Long Zhang [1,5], Jun Huang [1,5], Weiguo Lu [3,5], Xiaojing He [2✉], Haiyan Yan [4✉] & Fangwei Wang [1,3,5,6✉]

## Abstract

The 16-subunit Constitutive Centromere-associated Network (CCAN)-based inner kinetochore is well-known for connecting centromeric chromatin to the spindle-binding outer kinetochore. Here, we report a non-canonical role for the inner kinetochore in directly regulating sister-chromatid cohesion at centromeres. We provide biochemical, X-ray crystal structure, and intracellular ectopic localization evidence that the inner kinetochore directly binds cohesin, a ring-shaped multi-subunit complex that holds sister chromatids together from S-phase until anaphase onset. This interaction is mediated by binding of the 5-subunit CENP-OPQUR sub-complex of CCAN to the Scc1-SA2 sub-complex of cohesin. Mutation in the CENP-U subunit of the CENP-OPQUR complex that abolishes its binding to the composite interface between Scc1 and SA2 weakens centromeric cohesion, leading to premature separation of sister chromatids during delayed metaphase. We further show that CENP-U competes with the cohesin release factor Wapl for binding the interface of Scc1-SA2, and that the cohesion-protecting role for CENP-U can be bypassed by depleting Wapl. Taken together, this study reveals an inner kinetochore-bound pool of cohesin, which strengthens centromeric sister-chromatid cohesion to resist metaphase spindle pulling forces.

**Keywords** Centromere; Kinetochore; Cohesin; Sister-Chromatid Cohesion; Mitosis
**Subject Categories** Cell Cycle; Chromatin, Transcription & Genomics; Structural Biology

## Introduction

In the eukaryotic cell cycle, chromosomes are duplicated in the S-phase and subsequently segregated in mitosis, thereby ensuring that each of the two daughter cells receives a copy of the genome identical to that of the mother cell. Chromosome missegregation during mitosis results in aneuploidy, which is a hallmark of cancer and may promote tumorigenesis (Vasudevan et al, 2021).

During mitosis, spindle microtubules emanating from two spindle poles attach to sister kinetochores to align chromosomes on the metaphase plate. Prior to the metaphase-to-anaphase transition, the cohesion between sister chromatids must be strong enough to counteract spindle pulling forces. Sister-chromatid cohesion is mediated by the ring-shaped multi-subunit cohesin complex consisting of core subunits SMC1, SMC3, Scc1 (also called Mcd1 or Rad21), and SA2 (or its paralog SA1) (Guacci et al, 1997; Losada et al, 1998; Michaelis et al, 1997) (Fig. 1A). Cohesin is loaded onto chromatin in telophase and G1-phase (Ciosk et al, 2000; Krantz et al, 2004; Tonkin et al, 2004; Watrin et al, 2006), and topologically entraps sister DNAs to physically hold them together from S-phase in preparation for their segregation in anaphase (Hoencamp and Rowland, 2023).

In vertebrate mitosis, cohesin is released from chromosomes in a two-step manner. Starting from prophase till early metaphase, Wapl releases the bulk of cohesin from chromosome arms, a process referred to as the prophase pathway (Gandhi et al, 2006; Goto et al, 2017; Haarhuis et al, 2013; Kueng et al, 2006; Tedeschi et al, 2013). Cohesin at centromeres is protected against Wapl, which leads to the characteristic "X-shape" of metaphase chromosomes, particularly in response to drug-induced mitosis arrest (Dreier et al, 2011; Gimenez-Abian et al, 2004; Haarhuis et al, 2014; Hauf et al, 2005; Losada et al, 2002; Nakajima et al, 2007; Nishiyama et al, 2013; Sumara et al, 2002). When all chromosomes have bi-oriented on the metaphase plate, and the spindle assembly checkpoint is silenced, cohesin is cleaved by the protease Separase following the degradation of its inhibitory partner Securin, thereby allowing synchronous separation of sister chromatids in anaphase

[1]Life Sciences Institute, State Key Laboratory of Transvascular Implantation Devices of the Second Affiliated Hospital of Zhejiang University School of Medicine, MOE Laboratory of Biosystems Homeostasis and Protection, Zhejiang University, Hangzhou 310058, China. [2]Key Laboratory of Molecular Biophysics of the Ministry of Education, College of Life Science and Technology, Huazhong University of Science and Technology, Wuhan 430074, China. [3]Department of Gynecological Oncology, Women's Hospital, Zhejiang University School of Medicine, Hangzhou 310006, China. [4]Key Laboratory of Novel Targets and Drug Study for Neural Repair of Zhejiang Province, School of Medicine, Hangzhou City University, Hangzhou 310015, China. [5]Cancer Center, Zhejiang University, Hangzhou 310058, China. [6]Zhejiang Provincial Key Laboratory of Geriatrics and Geriatrics Institute of Zhejiang Province, Affiliated Zhejiang Hospital, Zhejiang University School of Medicine, Hangzhou 310058, China. [7]These authors contributed equally: Lu Yan, Xueying Yuan, Mingjie Liu. ✉E-mail: hexj@hust.edu.cn; yanhy@hzcu.edu.cn; fwwang@zju.edu.cn

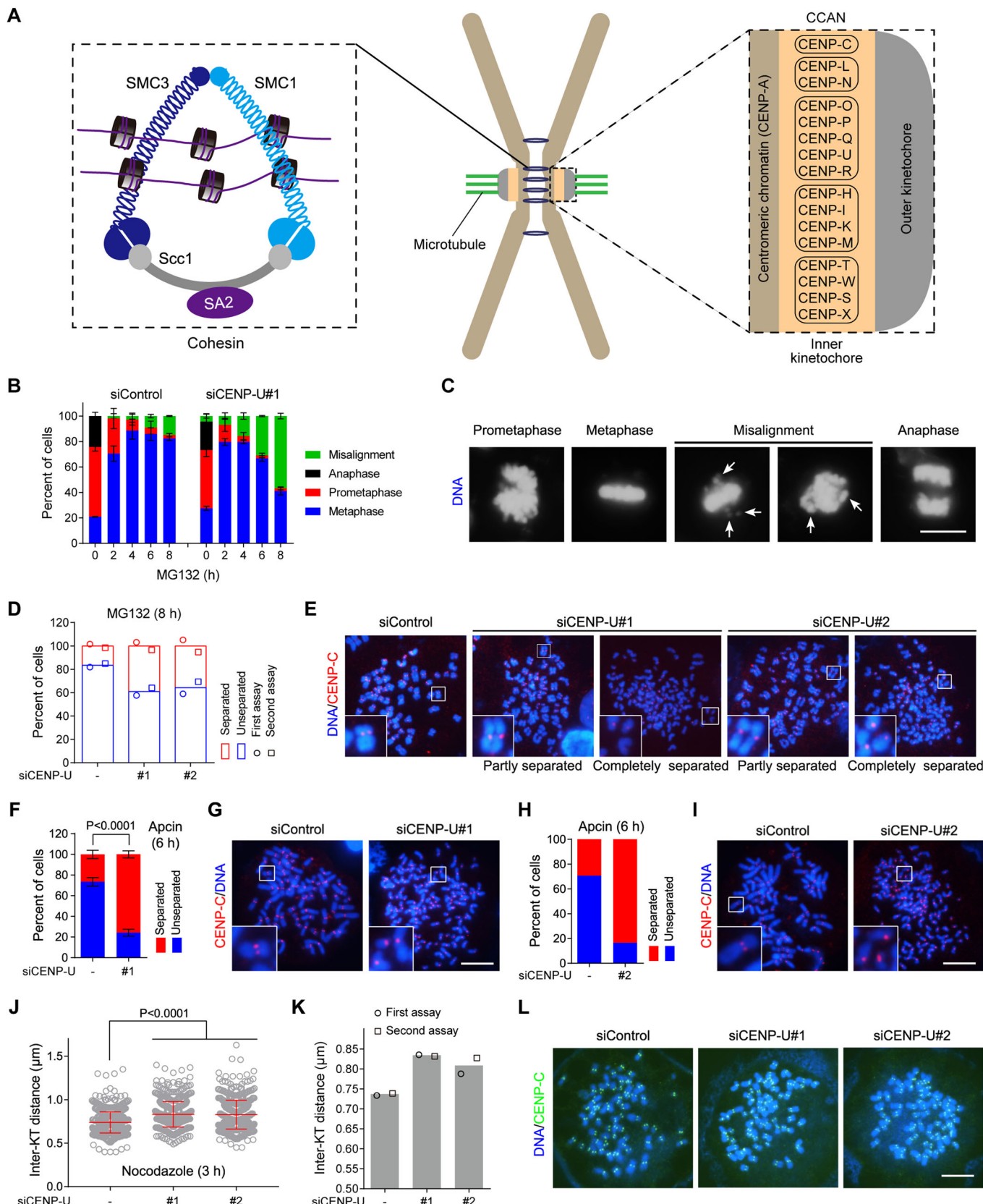

**Figure 1.   CENP-U strengthens centromeric cohesion and promotes metaphase sister-chromatid cohesion.**

(A) A schematic showing a metaphase chromosome with centromeric cohesin comprising core subunits, as well as the kinetochore organization with CCAN sub-complexes. (B, C) HeLa cells were transfected with control siRNA or CENP-U siRNA. At 48 h post-transfection, cells were treated with MG132 and then fixed at the indicated time points for DNA staining with DAPI. The percentage of mitotic cells in prometaphase, metaphase, metaphase with misaligned chromosomes, and anaphase was determined in over 300 cells for each condition from three independent experiments (B). Example images are shown (C). Arrows point to misaligned chromosomes. (D, E) HeLa cells were transfected with control siRNA or two independent CENP-U siRNAs. At 48 h post-transfection, cells were treated with MG132 for 8 h, then mitotic cells were collected to prepare chromosome spreads and then stained with the CENP-C antibody and DAPI. The percentage of cells in which the majority of sister chromatids was separated or unseparated was determined in over 200 cells for each condition. The means and individual data points from two independent experiments are plotted (D). Example images are shown (E). (F, G) HeLa cells were transfected with control siRNA or CENP-U siRNA. At 48 h post-transfection, cells were treated with Apcin for 6 h, then mitotic chromosome spreads were prepared, stained, and counted as in (D) and (E) in over 300 cells for each condition from three independent experiments, with statistics being analyzed for cells with separated chromatids (F). Example images are shown (G). (H, I) HeLa cells were transfected with siRNA and treated with Apcin as in F and G, then mitotic chromosome spreads were analyzed in 100 cells for each condition (H). Example images are shown (I). (J–L) HeLa cells were transfected with control siRNA or CENP-U siRNA. At 48 h post-transfection, cells were treated with nocodazole for 3 h. Mitotic chromosome spreads were stained with the CENP-C antibody and DAPI. The inter-KT distance was measured on over 1000 chromosomes in 20 cells (J). The means and individual data points from two independent experiments are plotted (K). Example images are shown (L). Data information: Statistics were performed using unpaired Student's *t*-test (F, J). Means and standard deviations (SDs) are shown (B, F, J). Scale bars, 10 μm (C, E, G, I, L). Source data are available online for this figure.

(McAinsh and Kops, 2023; Uhlmann et al, 2000). Therefore, centromeric cohesin protects centromere cohesion to withstand the metaphase spindle pulling forces until anaphase onset (Mirkovic and Oliveira, 2017). The precise localization of cohesin at centromeres and the proteins it interacts with specifically in this region remain poorly defined.

Centromeric chromatin contains the nucleosomes in which canonical histone H3 is replaced with its variant CENP-A (McKinley and Cheeseman, 2016) (Fig. 1A). Throughout the cell cycle, the CENP-A-containing nucleosomes specify the constitutive binding of centromeric chromatin to the inner kinetochore 16-subunit Constitutive Centromere-associated Network (CCAN) comprising CENP-C and the complexes of CENP-L/N, CENP-O/P/Q/U/R, CENP-H/I/K/M and CENP-T/W/S/X (Hara and Fukagawa, 2018; Pesenti et al, 2022; Tian et al, 2022; Yatskevich et al, 2022). The ten-subunit outer kinetochore KMN network (Knl1/Mis12 complex/Ndc80 complex) is further recruited by the CCAN in early mitosis to mediate the attachment between chromosomes and the mitotic spindle. Therefore, the CCAN-based inner kinetochore plays a canonical role in connecting centromeric chromatin to the mitotic spindle, which is essential for proper chromosome alignment and segregation.

In this study, we find an unexpected, non-canonical role for the CCAN-based inner kinetochore in binding cohesin and regulating sister-chromatid cohesion. Mechanistically, the CENP-U subunit of the CENP-OPQUR directly interacts with the composite interface of the Scc1-SA2 sub-complex of cohesin to antagonize Wapl binding, thereby retaining a subset of cohesin at the inner kinetochore to strengthen centromeric cohesion and resist metaphase spindle pulling forces.

## Results

### CENP-U strengthens centromeric cohesion and promotes sister-chromatid cohesion

We and others previously showed that CENP-U depletion by siRNA only moderately affected chromosome alignment in otherwise unperturbed mitosis or when cells were briefly arrested in metaphase for up to 3 h by treatment with MG132 (Chen et al, 2021; Nguyen et al, 2021), a proteasome inhibitor which prevents

anaphase onset by inhibiting the anaphase-promoting complex or cyclosome (APC/C)-mediated ubiquitination and destruction of the Cdk1 partner Cyclin B and the Separase inhibitor Securin. Interestingly, during prolonged metaphase arrest induced by MG132, there was a much more obvious defect in maintaining metaphase chromosome alignment in CENP-U-depleted cells than in control HeLa cells (Fig. 1B,C). Upon MG132 treatment for 6 h and 8 h, only 9.1 and 14.9% of control HeLa cells showed chromosome misalignment, respectively. Strikingly, 30.9% and 56.8% CENP-U-depleted cells displayed the respective defect.

We further examined the effect of CENP-U knockdown on chromosome alignment by time-lapse live cell imaging, using HeLa cells stably expressing histone H2B, which was C-terminally fused to GFP (H2B-GFP). To synchronize cells in mitosis, cells were transiently delayed in monopolar mitosis with the kinesin-5/Eg5 inhibitor S-trityl-L-cysteine (STLC), and then released into fresh medium containing MG132. Both control cells and CENP-U knockdown cells completed the formation of the metaphase plate within around 76 min after STLC release (Fig. EV1A,B; see Movies EV1, EV2). As expected from our previous study (Zhou et al, 2017), 92.8% of control cells were able to maintain chromosome alignment during the course of live imaging for 879 min. Remarkably, only 48.4% cells were able to do so after CENP-U depletion, whereas 51.6% cells underwent chromosome scattering from the metaphase plate. Thus, CENP-U is required to maintain chromosome alignment on the metaphase plate during prolonged metaphase arrest.

Immunofluorescence microscopy showed that the core inner kinetochore protein CENP-C frequently appeared as unpaired fluorescence foci on misaligned chromosomes in CENP-U depleted metaphase cells (Fig. EV1C), implying a precocious separation of sister chromatids. We then examined the effect of CENP-U knockdown on sister-chromatid cohesion, using metaphase chromosome spreads prepared from HeLa cells treated with MG132 for 8 h. Around 16% of control cells underwent premature sister-chromatid separation (PSCS) (Fig. 1D,E), presumably cohesion fatigue (Daum et al, 2011; Gorbsky, 2013; Sapkota et al, 2018). Strikingly, when CENP-U was depleted by two independent siRNAs (Fig. EV1D–F), sister chromatids were separated in 35.7–39.0% cells (Fig. 1D,E). We further assessed the effect of CENP-U depletion on sister-chromatid cohesion when cells were delayed in metaphase by treatment with Apcin, a small molecule

which binds to the APC/C activator Cdc20 and competitively inhibits the E3 ligase activity of APC/C (Sackton et al, 2014). After Apcin treatment for 6 h, 26.6–29.5% and 75.9–83.5%, mitotic cells underwent sister-chromatid separation in control HeLa cells and CENP-U-depleted cells, respectively (Fig. 1F–I). Thus, CENP-U depletion causes a defect in maintaining sister-chromatid cohesion during metaphase delay, which is induced by inhibition of either proteasome or the APC/C activity.

Given that CENP-U is a constitutive inner kinetochore protein, we wondered whether CENP-U depletion would loosen centromeric cohesion. For this purpose, we measured the inter-kinetochore (inter-KT) distance between sister kinetochores of chromosome spreads prepared from HeLa cells which were briefly arrested in mitosis with the microtubule destabilizer nocodazole, a condition where the PSCS was less obvious in both control cells and CENP-U depleted cells when compared to MG132 treatment. As shown in Fig. 1J–L, the inter-KT distance was around 12% further apart in CENP-U depleted cells than in control cells, indicative of weakened centromeric cohesion.

Therefore, CENP-U promotes the strength of centromeric cohesion, which is important to counteract the metaphase spindle pulling forces to prevent premature loss of sister-chromatid cohesion.

## CENP-U directly interacts with the Scc1-SA2 sub-complex of cohesin

To dissect the mechanism by which CENP-U protects sister-chromatid cohesion, we set out to identify proteins that associate with CENP-U. We found that the bacterially expressed glutathione S-transferase (GST)-fused human CENP-U protein in the forms of full-length (amino acid residues 1–418) and fragments encompassing residues 101–418 and 201–418 were unstable (Fig. EV2A–C), consistent with a previous study (Pesenti et al, 2018). We succeeded in purifying the GST-fused CENP-U fragment spanning residues 1–200 in *E. coli*, and thus used it to pull down proteins from HeLa cell lysates using GST as a negative control (Fig. EV2D). Mass spectrometry (MS) analysis showed that the cohesin core subunits SA2 and Scc1 were relatively enriched in the GST-CENP-U (1–200) pull-down sample (Dataset EV1). In line with the MS data, the immunoblotting analysis revealed that GST-CENP-U (1–200) specifically pulled down Scc1 (Fig. EV2E). Moreover, GST-CENP-U (1–100) and GST-CENP-U (1-60) were also able to pull down the cohesin core subunits Scc1, SA2, and SMC1, but not the regulatory subunit Pds5B (Figs. 2A and EV2F).

We next assessed the contribution of individual core subunit to the binding of cohesin complex to CENP-U. Knockdown of SMC3 or SMC1 moderately reduced the binding of Scc1 and SA2 to GST-CENP-U (1–60) (Fig. 2B,C). Interestingly, knockdown of Scc1 or SA2 prevented GST-CENP-U (1–60) from binding SMC1 and SMC3 (Fig. 2D,E). Thus, CENP-U associates with the core cohesin complex in a manner dependent on Scc1 and SA2, but not on SMC1 and SMC3.

To determine how Scc1 and SA2 mediate the interaction between cohesin and CENP-U, we examined the interaction of CENP-U with Scc1 and SA2 using a cell line in which multiple copies of Lac operator (LacO) repeats were stably integrated into the genome of U2OS cells (Janicki et al, 2004). SFB (a triple tag of S-tag, Flag-tag, and streptavidin-binding peptide)-fused CENP-U

did not localize to the LacO locus, which was visualized by binding of enhanced green fluorescent protein (EGFP)-fused Lac repressor (EGFP-LacI) to the LacO repeats (Fig. 2F,G). Tethering EGFP-LacI-fused Scc1 (EGFP-LacI-Scc1) to the LacO repeats moderately recruited SFB-CENP-U. Strikingly, co-expression of Myc-tagged SA2 (Myc-SA2) with EGFP-LacI-Scc1 caused strong recruitment of SFB-CENP-U to the LacO repeats (Fig. 2F,G). Thus, SA2 strongly promotes Scc1 interaction with CENP-U in cells.

We next performed GST pull-down assays using lysates prepared from HEK-293T cells transiently expressing Scc1 C-terminally fused to GFP (Scc1-GFP) and/or Myc-SA2. Scc1-GFP or Myc-SA2 alone was hardly pulled down by GST-CENP-U (1–60), whereas the co-expressed Scc1-GFP and Myc-SA2 efficiently bound to GST-CENP-U (1-60) (Fig. 2H), indicating that Scc1 and SA2 bind to CENP-U (1–60) as a complex. The failure of GST-CENP-U (1–60) in binding Scc1-GFP or Myc-SA2 when exogenously expressed alone suggests that endogenous SA2 and Scc1 in the cell lysates exist as a sub-complex with a molecular ratio of 1:1, leaving little endogenous SA2 and Scc1 available for binding Scc1-GFP and Myc-SA2, respectively.

It was previously shown that SA2 interacted with the Scc1 fragment spanning residues 171-450 (Zhang et al, 2013), and that recombinant protein fragments of Scc1 (281–420) and SA2 (80–1060) purified from insect cells were co-crystalized as a complex (Hara et al, 2014; Li et al, 2020). We, therefore, co-expressed Myc-SA2 with various fragments of Scc1-GFP for pull-down assays. GST-CENP-U (1-60) bound to full-length (residues 1–631) Scc1-GFP and the Scc1 fragment containing residues 281–420, but not to the fragments encompassing residues 1–281 and 420–631 (Fig. 2I). Consistently, Myc-SA2 was pulled down by GST-CENP-U (1–60) only when co-expressed with Scc1-GFP or Scc1 (281–420)-GFP.

Next, we bacterially co-expressed GST-Scc1 (281–420) and untagged SA2 (80–1060) and used glutathione beads to purify them as a complex (hereafter referred to as the GST-Scc1-SA2 sub-complex), as previously reported (Hara et al, 2014). Pull-down assays showed that SFB-CENP-U expressed in HEK-293T cells bound to the GST-Scc1-SA2 sub-complex, but not GST-Scc1 (281–420) alone (Fig. 2J).

Using bacterially expressed and purified recombinant proteins, we further found that the GST-Scc1-SA2 sub-complex was pulled down by MBP-fused CENP-U (1–200), but not by the negative control protein MBP-fused histone H2A (Fig. 2K). Thus, CENP-U directly binds to a sub-complex formed between Scc1 and SA2. We failed to bacterially purify full-length proteins of CENP-O, CENP-P, CENP-Q, CENP-R, and CENP-U, which prevented us from determining whether the whole CENP-OPQUR complex directly interacts with the Scc1-SA2 sub-complex in vitro.

CENP-Q and CENP-U form a sub-complex within the CENP-OPQUR complex in cells (Pesenti et al, 2022; Tian et al, 2022; Yatskevich et al, 2022), which prompted us to determine whether the CENP-U and CENP-Q sub-complex binds to the Scc1-SA2 sub-complex. Pull-down assays showed that GST-CENP-U (1–100) bound to Scc1 but not SFB-CENP-Q in HEK-293T cell lysates (Fig. EV2G). In sharp contrast, GST-CENP-U (101–418) pulled down SFB-CENP-Q but not Scc1. Thus, CENP-U uses the N-terminus and C-terminal region to bind the Scc1-SA2 sub-complex and CENP-Q, respectively. Moreover, SFB-CENP-Q did not bind to the GST-Scc1-SA2 sub-complex when expressed alone

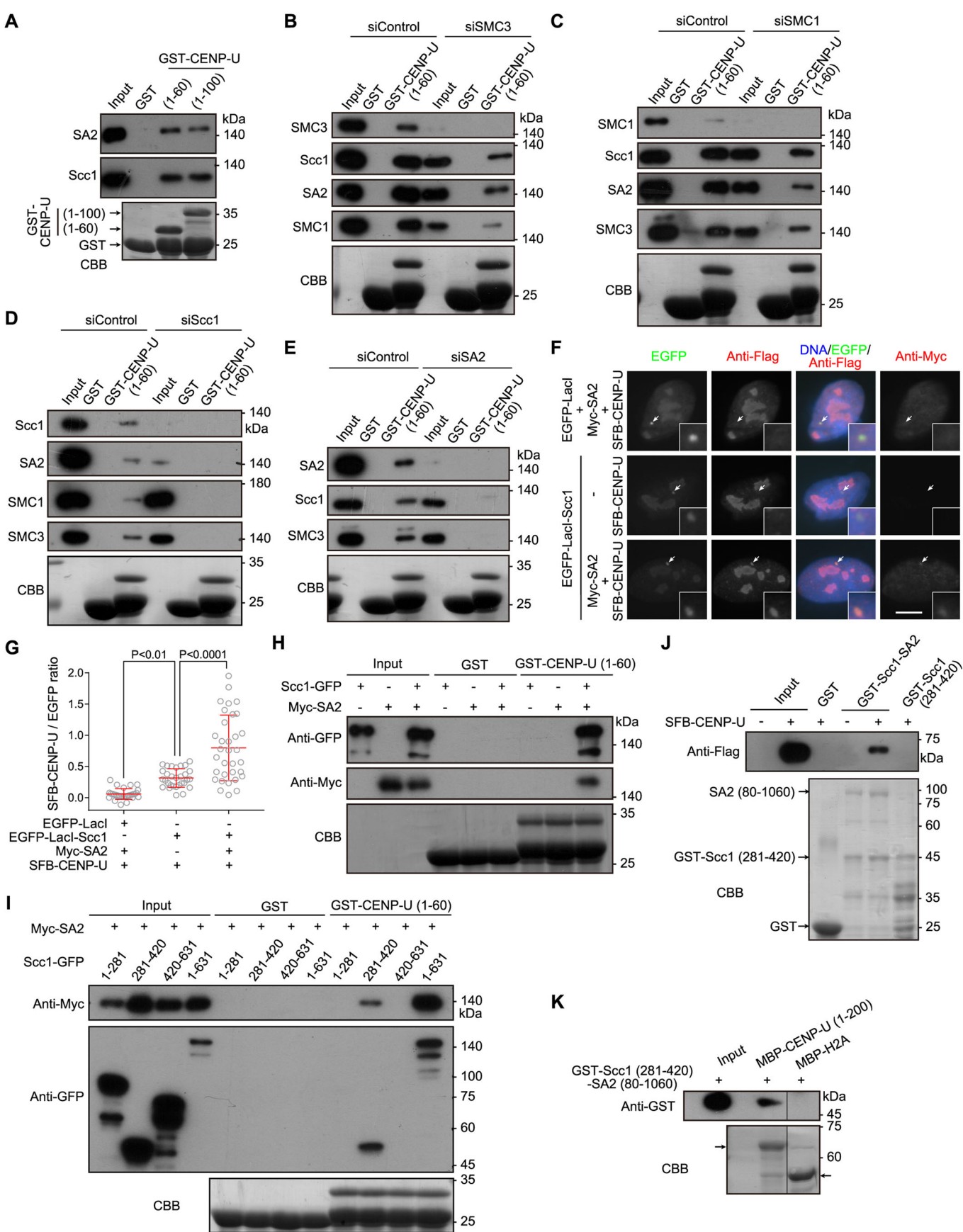

**Figure 2.  CENP-U directly interacts with the Scc1-SA2 sub-complex of cohesin.**

(A) HeLa cell lysates were subjected to pull-down with GST, GST-CENP-U (1–60), or GST-CENP-U (1–100), followed by immunoblotting with antibodies for Scc1 and SA2, and Coomassie brilliant blue (CBB) staining. (B–E) HeLa cell lysates were transfected with control siRNA or siRNA for SMC3 (B), SMC1 (C), Scc1 (D), or SA2 (E). At 48 h post-transfection, cell lysates were subjected to pull-down with GST or GST-CENP-U (1–60), followed by immunoblotting with antibodies for SMC3, SMC1, Scc1, or SA2, and CBB staining. (F, G) U2OS-LacO cells transiently expressing the indicated proteins were stained with antibodies for the Flag-tag, Myc-tag, and DAPI. Example images are shown (F). The white arrows point to the LacO repeats. Scale bars, 10 µm. The fluorescence intensity ratio of SFB-CENP-U/EGFP at the LacO repeats was quantified in 30 cells for each condition, with statistics being performed using one-way ANOVA (G). Means and SDs are shown. (H) Lysates prepared from HEK-293T cells transiently expressing Scc1-GFP and/or Myc-SA2 were subjected to pull-down with GST or GST-CENP-U (1–60), followed by immunoblotting with antibodies for GFP and the Myc-tag, and CBB staining. (I) Lysates prepared from HEK-293T cells transiently expressing Myc-SA2 and Scc1-GFP in the forms of full-length (residues 1–631), and the indicated fragments were subjected to pull-down with GST or GST-CENP-U (1–60), followed by immunoblotting with antibodies for GFP and the Myc-tag, and CBB staining. (J) Lysates prepared from HEK-293T cells expressing SFB-CENP-U were subjected to pull-down with GST, GST-Scc1 (281–420)-SA2 (80–1060), or GST-Scc1 (281–420), followed by immunoblotting with the antibody for the Flag-tag, and CBB staining. (K) The GST-Scc1 (281–420)-SA2 (80–1060) sub-complex was subjected to pull-down with MBP-CENP-U (1–200), or MBP-H2A as a negative control, followed by immunoblotting with the antibody for GST, and CBB staining. An irrelevant lane was removed. Source data are available online for this figure.

(Fig. EV2H). Interestingly, co-expression with CENP-U-GFP, but not CENP-U (1–100)-GFP, enabled the interaction. We confirmed that both CENP-U-GFP and CENP-U (1–100)-GFP were efficiently pulled down by GST-Scc1-SA2. These results indicate that CENP-Q indirectly binds Scc1-SA2 through interacting with the C-terminal region of CENP-U. In line with this, as well as the CENP-Q-dependent localization of CENP-U at kinetochores (Chen et al, 2021; Hori et al, 2008), the knockdown of CENP-Q in CENP-U-depleted cells did not cause a further defect in maintaining metaphase sister-chromatid cohesion (Fig. EV2I,J). These data imply that CENP-Q indirectly strengthens sister-chromatid cohesion through binding CENP-U that interacts with Scc1-SA2.

## The FDF motif of CENP-U directly binds to the composite interface between Scc1 and SA2

We next investigated how CENP-U interacts with the Scc1-SA2 sub-complex. Pull-down assays showed that endogenous Scc1 and SA2 in HeLa cell lysates bound to GST-CENP-U (1–60) and GST-CENP-U (1–50), but not GST-CENP-U (1–39) (Fig. 3A), indicating that the CENP-U fragment containing residues 40–50 is important for binding Scc1-SA2.

In the human CENP-U (40–60) fragment, we noticed a (42)-DVFDF-(46) motif which is conserved in mammals but not in vertebrates (Fig. 3B). This prompted us to compare the binding of human CENP-U (*Hs*CENP-U) and chicken CENP-U (*Gg*CENP-U) to Scc1 and SA2. The GST-Scc1-SA2 sub-complex pulled down *Hs*CENP-U-GFP but not *Gg*CENP-U-GFP, which were transiently expressed in HEK-293T cells (Fig. 3C). Strikingly, when the corresponding residues RKFLP in *Gg*CENP-U were replaced with residues DVFDF, the "humanized" *Gg*CENP-U-DVFDF-GFP chimeric protein was efficiently pulled down by GST-Scc1-SA2. Thus, the DVFDF motif enables chicken CENP-U to bind the Scc1-SA2 sub-complex. This is in line with a previous study which used peptide arrays to show that a CENP-U peptide containing the DVFDF motif bound to Scc1-SA2 in vitro with a Kd of around 4.0 µM (Li et al, 2020).

To elucidate the molecular basis underlying the interaction between the DVFDF motif of CENP-U and the Scc1-SA2 sub-complex, we obtained the crystals of the Scc1 (281–420)-SA2 (80–1060) complex soaking with a peptide PIDVFDFPDNS encompassing residues 40–50 of CENP-U, and solved the complex structure. The FDF motif-containing peptide bound to a hydrophobic pocket formed by Scc1 and SA2 (Figs. 3D and EV3A,B), similar to that of the previously reported CTCF peptide with a Y-x-

F motif (x represents any amino acid) (Li et al, 2020). The aromatic side chains of two phenylalanines dominantly mediated the interaction. While F44 was inserted into the hydrophobic pocket formed by L366, F367, and F371 from SA2, F46 formed hydrophobic interaction with Y297, W334 of SA2 and I337, L341 of Scc1. Thus, the FDF motif of CENP-U directly interacts with the binding interface between Scc1 and SA2.

We then generated CENP-U mutants with the phenylalanine-to-alanine or aspartic acid-to-lysine mutations in the FDF motif. Pull-down assays showed that mutating the FDF motif to ADA, FDA, and ADF all prevented GST-CENP-U (1–60) from binding Scc1, SA2, and SMC1 in cell lysates (Fig. 3E), whereas the FDF-to-FKF mutation did not affect GST-CENP-U (1–60) binding to Scc1 and SA2 (Fig. EV3C). Importantly, the FDF-to-ADA mutation blocked the direct binding of MBP-CENP-U (1–200) to GST-Scc1-SA2 (Fig. 3F). When transiently expressed in HEK-293T cells, CENP-U C-terminally fused to GFP (CENP-U-GFP), but not the CENP-U-ADA-GFP mutant, was pulled down from cell lysates by GST-Scc1-SA2 (Fig. 3G). In line with this, CENP-U (1–100)-GFP but not CENP-U (101–418)-GFP was pulled down by GST-Scc1-SA2 (Fig. EV3D). Moreover, in U2OS-LacO cells transiently expressing Myc-SA2, tethering EGFP-LacI-Scc1 to the LacO repeats recruited SFB-CENP-U, but not the SFB-CENP-U-ADA mutant (Fig. 3H,I). Similarly, when C-terminally fused to EGFP-LacI, CENP-U, but not the ADA mutant, recruited co-expressed Scc1-Myc and SA2-Flag (Fig. EV3E). Both CENP-U-EGFP-LacI and CENP-U-ADA-EGFP-LacI efficiently recruited Plk1 (Fig. EV3F), indicating that mutating the FDF motif does not affect CENP-U interaction with Plk1. Thus, the FDF motif is necessary for CENP-U binding to the Scc1-SA2 sub-complex in vitro and in cells.

Additionally, SFB-CENP-Q was pulled down by GST-Scc1-SA2 when co-expressed with CENP-U-GFP, but not the CENP-U-ADA-GFP mutant (Fig. EV3G), further confirming the indirect association of CENP-Q with the Scc1-SA2 sub-complex.

Taken together, these data indicate that the FDF motif of CENP-U directly interacts with the interface between Scc1 and SA2, and that this interaction is necessary for the CENP-U and CENP-Q sub-complex to bind cohesin.

## The FDF motif is required for CENP-U to maintain metaphase sister-chromatid cohesion

To investigate the functional significance of the FDF motif in CENP-U, we stably expressed siRNA-resistant CENP-U-GFP (wild-type/WT and

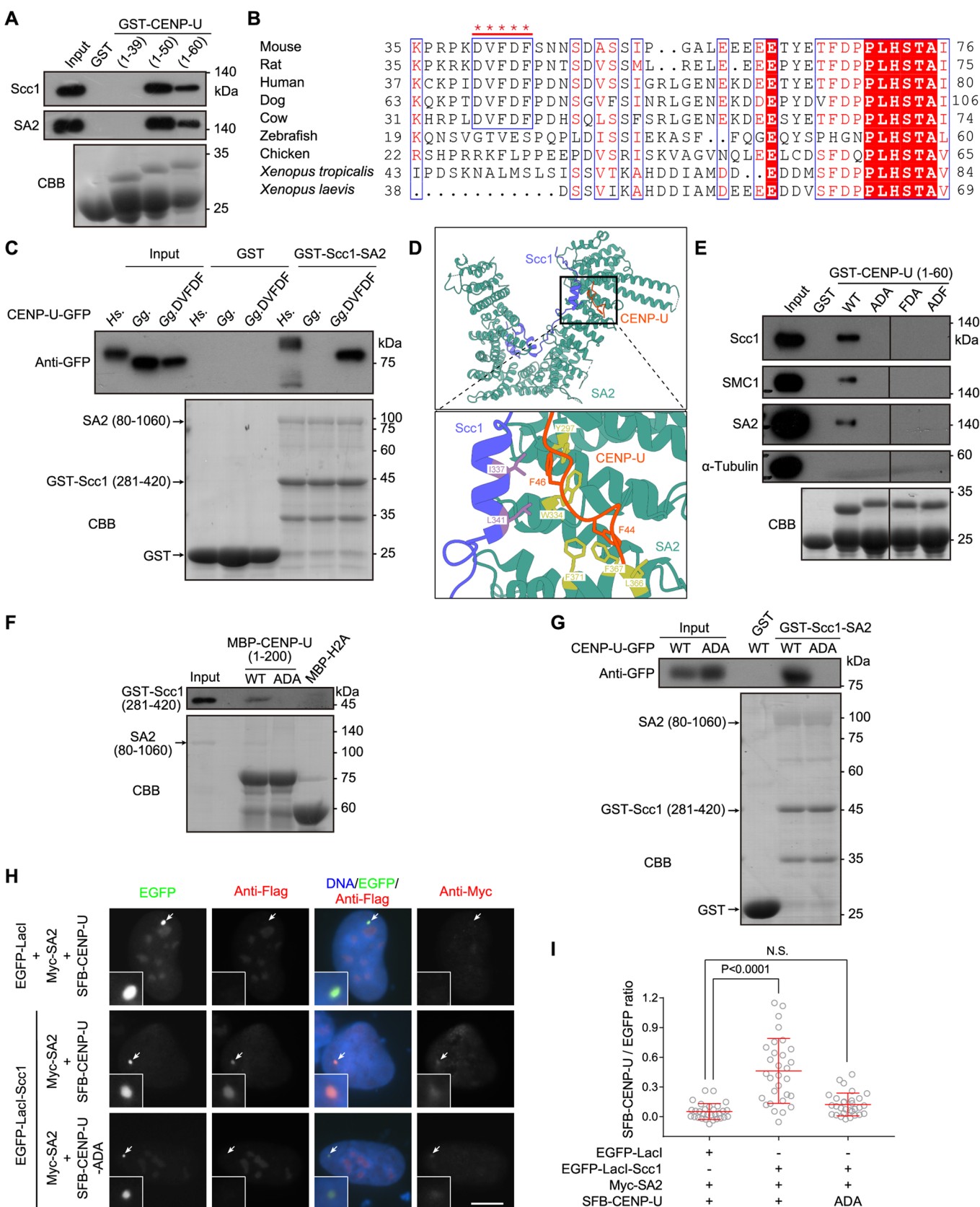

**Figure 3. The FDF motif of CENP-U directly binds to the composite interface between Scc1 and SA2.**

(A) HeLa cell lysates were subjected to pull-down with GST, GST-CENP-U (1–39), GST-CENP-U (1–50), or GST-CENP-U (1–60), followed by immunoblotting with antibodies for Scc1 and SA2, and CBB staining. (B) Multiple sequence alignment for the N-terminus of CENP-U. The DVFDF motif that is conserved in mammals are marked with *****. (C) Lysates prepared from HEK-293T cells transiently expressing CENP-U-GFP (human, chicken, and chimeric chicken with the DVFDF motif) were subjected to pull-down with GST or GST-Scc1 (281–420)-SA2 (80–1060), followed by immunoblotting with the antibody for GFP, and CBB staining. (D) Cartoon presentation of the overall structure of Scc1-SA2-CENP-U complex colored in purple, cyan, and orange, respectively (left), and the zoom view for binding details between F44 and F46 of CENP-U and Scc1-SA2 (right). (E) HeLa cell lysates were subjected to pull-down with GST or GST-CENP-U (1–60) in the forms of WT, ADA, FDA, and ADF, followed by immunoblotting with antibodies for Scc1, SMC1, SA2, and α-Tubulin, and CBB staining. Irrelevant lanes were removed. (F) The GST-Scc1 (281–420)-SA2 (80–1060) subcomplex was subjected to pull-down with MBP-H2A or MBP-CENP-U (1–200) in the forms of WT and ADA, followed by immunoblotting with the antibody for GST, and CBB staining. (G) Lysates prepared from HEK-293T cells transiently expressing CENP-U-GFP in the forms of WT and ADA were subjected to pull-down with GST or GST-Scc1 (281–420)-SA2 (80–1060), followed by immunoblotting with the antibody for GFP, and CBB staining. (H, I) U2OS-LacO cells transiently expressing the indicated proteins were stained with antibodies for the Flag-tag, Myc-tag, and DAPI. Example images are shown (H). The white arrows point to the LacO repeats. Scale bars, 10 μm. The fluorescence intensity ratio of SFB-CENP-U/EGFP at the LacO repeats was quantified in 30 cells for each condition, with statistics being performed using one-way ANOVA (I). Means and SDs are shown. NS no significance. Source data are available online for this figure.

the ADA mutant) in HeLa cells (Fig. 4A). In line with the FDF motif-dependent interaction of CENP-U with Scc1-SA2, CENP-U-GFP, but not the CENP-U-ADA-GFP mutant, co-immunoprecipitated endogenous cohesin subunits Scc1, SA2, SMC1 and SMC3 (Fig. 4B). Fluorescence microscopy showed that the FDF-to-ADA mutation did not noticeably affect the localization of CENP-U to the kinetochore (Fig. 4C). Importantly, knockdown of endogenous CENP-U by siRNA caused a strong defect in maintaining metaphase chromosome alignment in control HeLa cells and CENP-U-ADA-GFP-expressing cells, but not in CENP-U-GFP-expressing cells (Fig. 4D). Thus, CENP-U-ADA-GFP mutant is defective in supporting chromosome alignment on the metaphase plate.

We next examined chromosome spreads prepared from MG132-treated mitotic cells expressing CENP-U-GFP (WT and the ADA mutant). As shown in Fig. 4E,F, the defect in maintaining metaphase sister-chromatid cohesion caused by depletion of endogenous CENP-U was partly but significantly ($p = 0.0025$) rescued by CENP-U-GFP, whereas the CENP-U-ADA-GFP mutant did not show any rescue effect. Similar results were observed in other independent stable clones expressing siRNA-resistant CENP-U-GFP (WT and the ADA mutant) (Fig. EV4A–D).

We further used CRISPR/Cas9-mediated homology-directed repair to mutate the DVFDF motif of endogenous CENP-U to AVAAA in HeLa cells (Fig. 4G). Inspection of chromosome spreads from MG132-arrested metaphase cells demonstrated that cells with the CENP-U-AVAAA mutation were significantly less effective than control HeLa cells in maintaining sister-chromatid cohesion (Fig. 4H,I). Moreover, upon brief treatment with nocodazole, the inter-KT distance was around 10.5% further apart in CENP-U-AVAAA mutant cells than in control cells, which is indicative of weakened centromeric cohesion (Fig. 4J,K). These results indicate the importance of the FDF motif of CENP-U for strengthening centromeric cohesion and maintaining metaphase sister-chromatid cohesion.

We and others previously showed that CENP-U directly interacts with Plk1 and recruits Plk1 to the inner kinetochore (Chen et al, 2021; Kang et al, 2006; Nguyen et al, 2021; Singh et al, 2021). We, therefore, assessed whether the CENP-U-bound Plk1 is involved in regulating sister-chromatid cohesion. We established a HeLa-derived cell line stably expressing the CENP-U-T78A-GFP mutant (Fig. 4L), in which threonine-78 was mutated to alanine to prevent Plk1 binding (Chen et al, 2021; Kang et al, 2006; Nguyen et al, 2021; Singh et al, 2021). Examination of MG132-arrested metaphase chromosome spreads showed that the CENP-U-T78A-

GFP mutant was able to rescue the cohesion defect caused by knockdown of endogenous CENP-U (Fig. 4M,N), indicating that the interaction with Plk1 is dispensable for CENP-U to maintain metaphase sister-chromatid cohesion. Moreover, both CENP-U-GFP and the CENP-U-T78A-GFP mutant were able to rescue the chromosome misalignment defect caused by the depletion of endogenous CENP-U (Fig. 4O), which is in line with our previous study (Chen et al, 2021).

Taken together, these data led us to conclude that the FDF motif-mediated interaction of CENP-U with the Scc1-SA2 subcomplex strengthens centromeric cohesion, which is required to maintain sister-chromatid cohesion under the metaphase spindle pulling forces.

## Wapl depletion bypasses the requirement for CENP-U in protecting centromeric cohesion

During mitosis, Wapl releases the bulk of cohesin from chromosome arms, whereas its activity at centromeres is normally inhibited to prevent premature loss of centromeric cohesion. We confirmed that MG132 treatment for up to 8 h did not detectably affect the protein levels of Wapl, Scc1 and SMC3 in both control HeLa cells and CENP-U-depleted cells (Fig. 5A). We thus wondered whether aberrantly increased activity of Wapl at metaphase centromeres might account for the weakened centromeric cohesion in cells in which the CENP-U-cohesin interaction is disrupted. If this is the case, Wapl knockdown would be expected to rescue the centromeric cohesion defect in CENP-U-depleted cells.

We next examined metaphase sister-chromatid cohesion in HeLa cells depleted of Wapl and CENP-U, either individually or in combination. As expected, 16.1 and 50.8% of cells underwent PSCS upon transfection with control siRNA and CENP-U siRNA, respectively (Fig. 5B–D). Interestingly, only 14.3% of cells co-depletion of Wapl and CENP-U underwent PSCS, indicating that Wapl knockdown prevented the premature loss of sister-chromatid cohesion in cells depleted of CENP-U.

We further measured the inter-KT distance on chromosome spreads prepared from nocodazole-arrested mitotic cells depleted of Wapl and/or CENP-U. We found that the inter-KT distance in HeLa cells co-depleted of Wapl and CENP-U was 15.9% shorter than that in CENP-U depleted cells, and was comparable to that in Wapl depleted cells (Fig. 5E–H). Thus, Wapl knockdown strengthens centromeric cohesion regardless of the presence or absence of CENP-U.

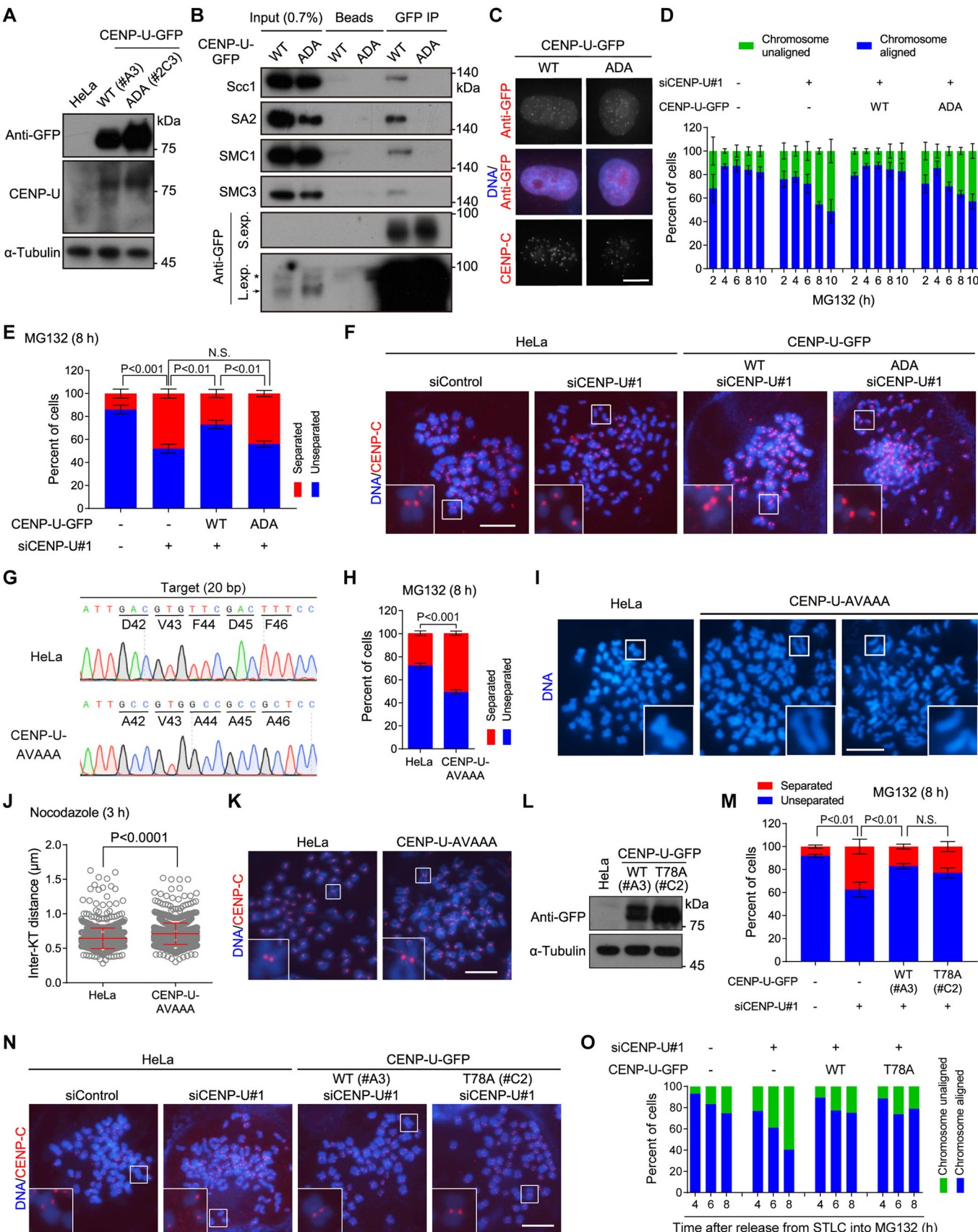

**Figure 4.   The FDF motif is required for CENP-U to maintain metaphase sister-chromatid cohesion.**

(A) Asynchronous HeLa cells stably expressing siRNA-resistant CENP-U-GFP (WT or the ADA mutant) were subjected to immunoblotting with antibodies for GFP, CENP-U, and α-Tubulin. (B) Asynchronous HeLa cells stably expressing the indicated proteins were subjected to immunoprecipitation with the GFP antibody or control protein A/G beads, followed by immunoblotting with antibodies for GFP, Scc1, SA2, SMC3, SMC1, and α-Tubulin. The asterisk points to background bands. The arrow points to the CENP-U-GFP protein. S. exp. short exposure, L. exp. long exposure. (C) Asynchronous HeLa cells stably expressing the indicated proteins were subjected to immunostaining with antibodies for GFP and CENP-C. Example images are shown. (D) HeLa cells stably expressing the indicated proteins were transfected with control siRNA or CENP-U siRNA. At 48 h post-transfection, cells were treated with MG132 and then fixed at the indicated time points for DNA staining. The percentage of mitotic cells in which chromosomes were aligned or unaligned was determined in over 300 cells for each condition from three independent experiments. (E, F) HeLa cells stably expressing the indicated proteins were transfected with control siRNA or CENP-U siRNA. At 48 h post-transfection, cells were treated with MG132 for 8 h, then mitotic cells were collected to prepare chromosome spreads, and then stained with the CENP-C antibody and DAPI. The percentage of cells in which the majority of sister chromatids was separated or unseparated was determined in over 300 cells for each condition from three independent experiments, with statistics being analyzed for cells with separated chromatids (E). Example images are shown (F). NS no significance. (G) Genomic DNA sequencing of control HeLa cells and the CENP-U-AVAAA mutant clone 1B6. The genomic DNA PCR fragments were subcloned and sequenced, which showed that all 20 individual bacterial colonies had the desired CENP-U-AVAAA mutation. (H, I) Control HeLa cells and the CENP-U-AVAAA mutant cells were treated with MG132 for 8 h, then mitotic chromosome spreads were prepared for DNA staining. The percentage of cells in which the majority of sister chromatids was separated or unseparated was determined in 300 cells for each condition from three independent experiments, with statistics being analyzed for cells with separated chromatids (H). Example images are shown (I). (J, K) Control HeLa cells and the CENP-U-AVAAA mutant cells were treated with nocodazole for 3 h, then mitotic chromosome spreads were stained with the CENP-C antibody and DAPI. The inter-KT distance was measured on over 1000 chromosomes in 20 cells (J). Example images are shown (K). (L) Asynchronous HeLa cells stably expressing siRNA-resistant CENP-U-GFP (WT or the T78A mutant) were subjected to immunoblotting with antibodies for GFP and α-Tubulin. (M, N) Cells were transfected with control siRNA or CENP-U siRNA. At 48 h post-transfection, cells were treated with MG132 for 8 h, then mitotic chromosome spreads were stained with the CENP-C antibody and DAPI. Cells in which the majority of sister chromatids was separated or unseparated were quantified in 300 cells for each condition from three independent experiments, with statistics being analyzed for cells with separated chromatids (M). Example images are shown (N). (O) HeLa cells stably expressing the indicated proteins were transfected with control siRNA or CENP-U siRNA. At 48 h post-transfection, cells were released from 5-h treatment with STLC into fresh medium containing MG132, then fixed at the indicated time points for DNA staining. The percentage of mitotic cells in which chromosomes were aligned or unaligned was determined in 100 cells for each condition. Data information: Statistics were performed using with one-way ANOVA (E) or unpaired Student's *t*-test (H, J, M). Means and SDs are shown (D, E, H, J, M). Scale bars, 10 μm (C, F, I, K, N). Source data are available online for this figure.

Taken together, these data indicate that the requirement for CENP-U in protecting centromeric cohesion can be bypassed by Wapl depletion.

## Wapl directly binds to the Scc1-SA2 sub-complex in an FGF motif-dependent manner

Given our observations that the FDF motif is required for CENP-U to bind Scc1-SA2 and to protect sister-chromatid cohesion, that Wapl knockdown rescues the centromeric cohesion defect in CENP-U depleted cells, and the fact that vertebrate Wapl contains three conserved FGF motifs which were reported to be involved in binding cohesin in vitro (Ouyang et al, 2013; Shintomi and Hirano, 2009), we next examined the contribution of FGF motifs to human Wapl binding to Scc1-SA2, particularly in cells.

In the pull-down assay with HeLa cell lysates, endogenous Wapl bound to the GST-Scc1-SA2 sub-complex, but not to GST-Scc1 (281–420) (Fig. 6A). Similarly, Wapl with a C-terminal Flag-tag (Wapl-Flag) transiently expressed in HEK-293T cells was specifically pulled down by GST-Scc1-SA2 (Fig. 6B). Remarkably, the Wapl-3xEGE-Flag mutant, in which all three FGF motifs ($F_{73}GF_{75}$, $F_{429}GF_{431}$, and $F_{453}GF_{455}$) were mutated to EGE, was much less efficiently pulled down by GST-Scc1-SA2 (Fig. 6B). Using purified recombinant proteins, we further found that GST-Scc1-SA2 specifically pulled down MBP-fused Wapl (1–630) containing all three FGF motifs, and that mutation of these motifs to EGE strongly reduced the interaction (Fig. 6C).

In U2OS-LacO cells transiently expressing Myc-SA2, we found that tethering EGFP-LacI-Scc1 (281–420) to the LacO repeats recruited Wapl-Flag, and that the FGF-to-EGE mutation significantly reduced the recruitment (Fig. 6D,E). As a comparison, co-expression of Myc-SA2 with EGFP-LacI-Scc1 (281–420) recruited SFB-CENP-U, but not the SFB-CENP-U-ADA mutant (Fig. 6F,G), which is in line with the results shown in Fig. 3H,I. These results indicate that Wapl directly

binds to the Scc1-SA2 complex, and that this interaction is largely dependent on the FGF motifs both in vitro and in cells.

## CENP-U competes with Wapl for binding to the Scc1-SA2 sub-complex

Based on the structural analysis of the CENP-U FDF motif interaction with the interface between Scc1 and SA2, we next examined whether mutations of Scc1 residues I337 and L341, as well as SA2 residues W334, F367, and F371, affect the binding of Scc1-SA2 to CENP-U.

Using the LacO/LacI-based recruitment assay in the U2OS-LacO cells, we found that the mutations of I337 A/L341A in Scc1 and W334A, F367A, or F371A in SA2 did not detectably affect the recruitment of Myc-SA2 by EGFP-LacI-Scc1 (Fig. 7A,B). Thus, these mutations do not disrupt the formation of the Scc1-SA2 sub-complex, which is in line with the observation that Scc1 binds SA2 via an extensive interface (Hara et al, 2014).

Pull-down assays with cell lysates showed that GST-CENP-U (1–60) bound to co-expressed Scc1-GFP and Myc-SA2, but not the mutants of Scc1-I337A/L341A-GFP, Myc-SA2-W334A, Myc-SA2-F367A, and Myc-SA2-F371A (Fig. 7C,D). Interestingly, when MBP-Wapl (1–630) was used to pull down co-expressed Scc1-GFP and Myc-SA2, these mutations strongly reduced the pull-down efficiency (Fig. 7E,F). Thus, the residues of Scc1-SA2 required for binding to the FDF motif of CENP-U are also important for interaction with Wapl.

These results suggest that Wapl binds to the composite interface formed between Scc1 and SA2, similar to CENP-U interaction with Scc1-SA2. We thus carried out GST-CENP-U and MBP-Wapl pull-down competition assays. The pull-down of Scc1-GFP and Myc-SA2 in cell lysates by GST-CENP-U (1–60) was substantially reduced upon the addition of MBP-Wapl (1–630) (Fig. 7G). In contrast, the addition of the same amount of the MBP-Wapl (1–630)-3xEGE mutant did not

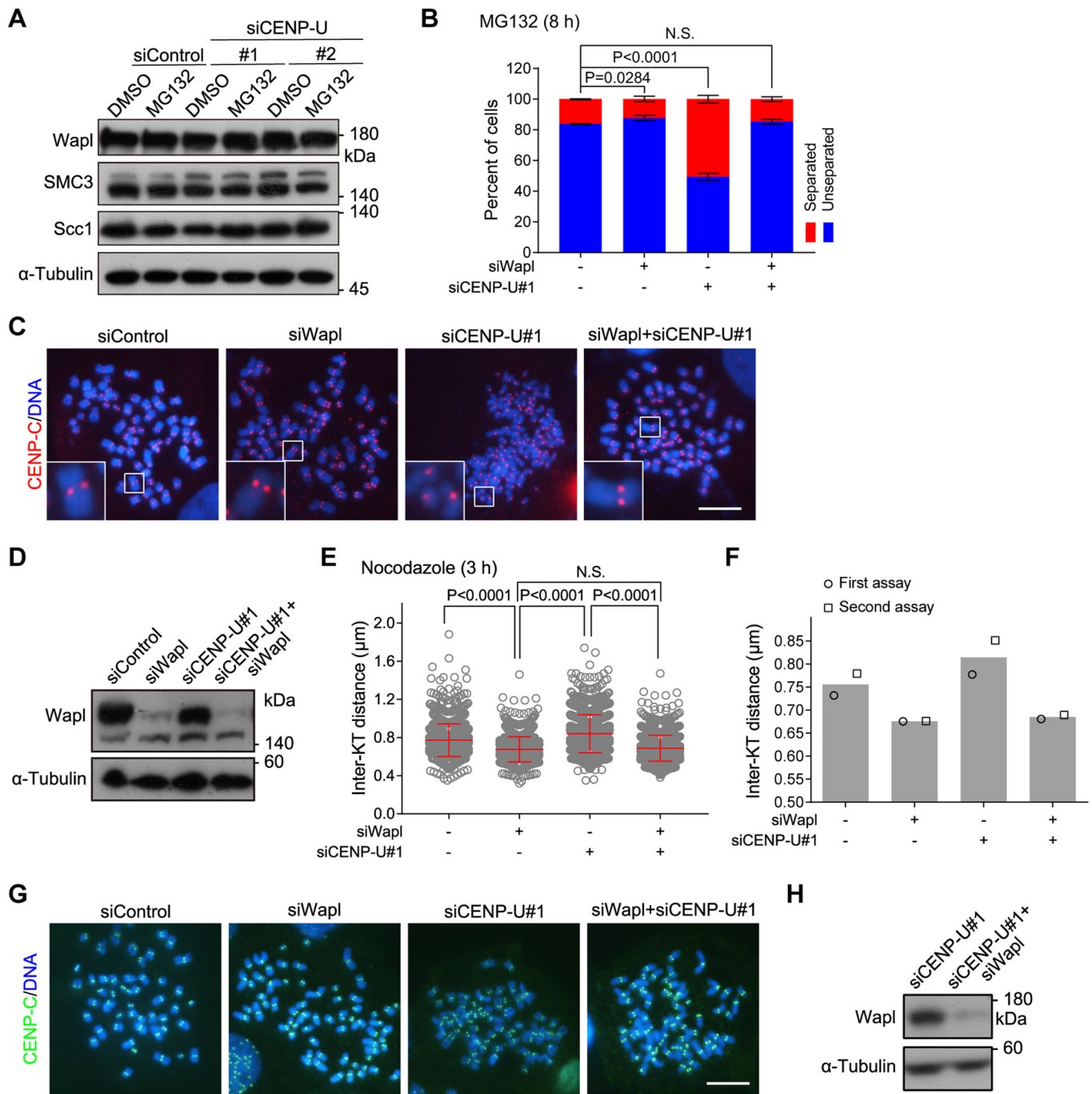

**Figure 5. Wapl depletion bypasses the requirement for CENP-U in protecting centromeric cohesion.**

(A) HeLa cells were transfected with control siRNA or CENP-U siRNA. At 48 h post-transfection, cells were treated with the solvent DMSO or MG132 for 8 h, then cell lysates were immunoblotted with antibodies for Wapl, SMC3, Scc1, and α-Tubulin. (B–D) HeLa cells were transfected with control siRNA, CENP-U siRNA, and/or Wapl siRNA. At 48 h post-transfection, cells were treated with MG132 for 8 h, then mitotic chromosome spreads were prepared and stained with the CENP-C antibody and DAPI. The percentage of cells in which the majority of sister chromatids was separated or unseparated was determined in 300 cells for each condition from three independent experiments, with statistics being analyzed for cells with separated chromatids (B). Example images are shown (C). Cell lysates were immunoblotted with antibodies for Wapl and α-Tubulin (D). (E–H) HeLa cells were transfected with the indicated siRNAs. At 48 h post-transfection, cells were treated with nocodazole for 3 h. Then mitotic chromosome spreads were prepared and stained with the CENP-C antibody and DAPI. The inter-KT distance was measured on over 1000 chromosomes in 20 cells (E). The means and individual data points from two independent experiments are plotted (F). Example images are shown (G). Cell lysates were immunoblotted with antibodies for Wapl and α-Tubulin (H). Data information: Statistics were performed using unpaired Student's t-test (B) and one-way ANOVA (E). Means and SDs are shown (B, E). Scale bars, 10 μm (C, G). NS no significance (B, E). Source data are available online for this figure.

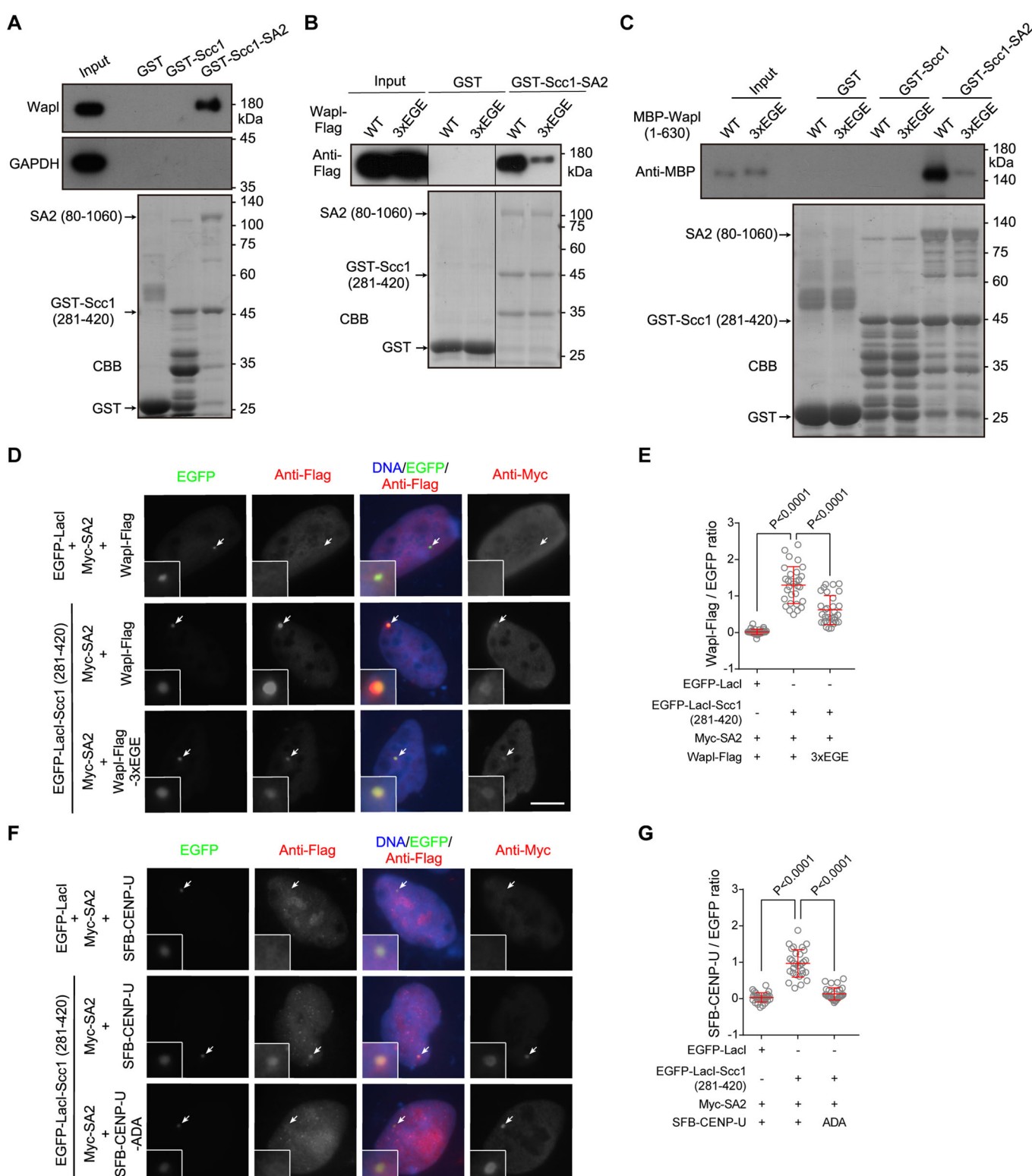

noticeably affect the pull-down (Fig. 7H). Moreover, the pull-down of Scc1-GFP and Myc-SA2 by MBP-Wapl (1–630) was reduced upon the addition of GST-CENP-U (1–60), but not the GST-CENP-U (1–60)-ADA mutant (Fig. 7I). We therefore conclude that CENP-U and Wapl bind to Scc1-SA2 in a similar and competitive manner.

## CENP-U and Sgo1 additively contribute to the strength of centromeric cohesion

Our data presented above indicate a role for CENP-U in protecting centromeric cohesion through binding the Scc1-SA2 sub-complex of

**Figure 6.  Wapl directly binds to the Scc1-SA2 sub-complex in an FGF motif-dependent manner.**

(A) Lysates prepared from asynchronous HeLa cells were subjected to pull-down with GST, GST-Scc1 (281–420), or GST-Scc1 (281–420)-SA2 (80–1060), followed by immunoblotting with antibodies for Wapl and GAPDH, and CBB staining. (B) Lysates prepared from HEK-293T cells transiently expressing Wapl-Flag in the forms of WT and the 3xEGE mutant were subjected to pull-down by GST or GST-Scc1 (281–420)-SA2 (80–1060), followed by immunoblotting with the antibody for the Flag-tag, and CBB staining. Irrelevant lanes were removed. (C) MBP-Wapl (1–630) in the forms of WT and the 3xEGE mutant were subjected to pull-down by GST, GST-Scc1 (281–420), or GST-Scc1 (281–420) -SA2 (80–1060), followed by immunoblotting with the antibody for MBP, and CBB staining. (D, E) U2OS-LacO cells transiently expressing the indicated proteins were stained with antibodies for the Flag-tag and Myc-tag, and DAPI. Example images are shown (D). The fluorescence intensity ratio of Wapl-Flag/ EGFP at the LacO repeats was quantified in 30 cells for each condition (E). (F, G) U2OS-LacO cells transiently expressing the indicated proteins were stained with antibodies for the Flag-tag, Myc-tag, and DAPI. Example images are shown (F). The fluorescence intensity ratio of SFB-CENP-U/EGFP at the LacO repeats was quantified in 30 cells for each condition (G). Data information: The white arrows point to the LacO repeats (D, F). Scale bars, 10 μm (D, F). Statistics were performed using unpaired Student's *t*-test (E, G). Means and SDs are shown (E, G). Source data are available online for this figure.

cohesin. A previous study reported that the Scc1-SA2 sub-complex can also bind Shugoshin-1 (Sgo1) (Hara et al, 2014), which plays a well-known role in protecting sister-chromatid cohesion (Kitajima et al, 2004; McGuinness et al, 2005; Salic et al, 2004; Tang et al, 2004). In early mitosis, Sgo1 predominantly localized to centromeres, whereas some Sgo1 was also distributed on the chromosome arm (Fig. EV5A). This is in line with previous studies (Kitajima et al, 2005; McGuinness et al, 2005; Nakajima et al, 2007), as well as the observations that small amounts of cohesin is present on mitotic chromosome arms (Chu et al, 2020; Gimenez-Abian et al, 2004), and that Sgo1 can bind cohesin (Hara et al, 2014; Liu et al, 2013b).

We next examined the effect of CENP-U knockdown on sister-chromatid cohesion in Sgo1-depleted cells. As expected, Sgo1 depletion by siRNA caused a dramatic loss of sister-chromatid cohesion (Fig. EV5B), reflecting an important role for Sgo1 in protecting cohesin both on chromosome arms and at centromeres (Nakajima et al, 2007). Interestingly, cells co-depleted of CENP-U and Sgo1 showed a further defect in sister-chromatid cohesion when compared to cells depleted of Sgo1 alone (Fig. EV5B), indicating that CENP-U and Sgo1 additively promote sister-chromatid cohesion.

As previously reported (Kawashima et al, 2010; Liu et al, 2015), when cells enter mitosis, the spindle checkpoint protein Bub1-mediated histone H2A threonine-120 phosphorylation (H2ApT120) at centromeres directly recruits Sgo1. Upon chromosome biorientation on the metaphase plate, the H2ApT120 signal is largely reduced following Bub1 release from kinetochores, resulting in Sgo1 delocalization from centromeres (Fig. EV5C). We then assessed the contribution of CENP-U to sister-chromatid cohesion when Sgo1 is uncapable of binding H2ApT120. Using CRISPR/Cas9 genome editing, we previously obtained HeLa-derived cell lines expressing the H2ApT120-binding-deficient Sgo1-K492A mutant (Liang et al, 2019), which cannot localize to mitotic centromeres to protect centromeric cohesion (Liu et al, 2013a; Liu et al, 2015). In line with our previous observation (Liang et al, 2019), cells expressing Sgo1-K492A were defective in maintaining sister-chromatid cohesion during metaphase arrest induced by MG132 treatment for 2–4 h (Fig. 8A,B). Importantly, the depletion of CENP-U resulted in a further increase in the percentage of Sgo1-K492A cells with PSCS, demonstrating a role for CENP-U in maintaining sister-chromatid cohesion when Sgo1 is delocalized from centromeres. Moreover, CENP-U knockdown caused a 20.9% increase in the inter-KT distance in nocodazole-arrested mitotic cells expressing the Sgo1-K492A mutant (Figs. 8C and EV5D,E). This indicates that CENP-U is required to maintain centromeric cohesion in cells lacking centromeric Sgo1.

The additive role of CENP-U and Sgo1 in protecting centromeric cohesion prompted us to determine whether Sgo1 binds to the cohesin complex in a way similar to CENP-U does. Human Sgo1 contains a $Y_{335}NF_{337}$ motif (Fig. EV5F), which is conserved in vertebrates as a Y/F-x-F motif and is similar to the FDF motif of CENP-U and the Y-x-F motif of CTCF (Li et al, 2020). Though the YNF motif-containing Sgo1 peptide SNDAYNFNLEE did not show apparent binding to Scc1-SA2 in the in vitro peptide array assays (Li et al, 2020), our pull-down assays showed that mutating the YNF motif to ANA prevented GST-Sgo1 (313–353) from binding endogenous Scc1, SA2, SMC1, and SMC3 in cell lysates (Fig. 8D). Conversely, GST-Scc1-SA2 specifically pulled down Sgo1-GFP transiently expressed in HEK-293T cells, but not the Sgo1-ANA-GFP mutant (Fig. 8E). The YNF-to-ANA mutation also blocked the direct binding of GST-Scc1-SA2 to recombinant MBP-Sgo1 (230–400) (Fig. 8F).

Moreover, pull-down assays with HEK-293T cell lysates showed that GST-Sgo1 (313–353) bound to Scc1-GFP and Myc-SA2 only when they were co-expressed (Fig. 8G), indicating that Sgo1 binds to a sub-complex formed between Scc1 and SA2. Importantly, GST-Sgo1 (313–353) failed to pull down the mutants of Scc1-I337A/L341A-GFP, Myc-SA2-W334A, Myc-SA2-F367A, and Myc-SA2-F371A (Fig. 8H), which were defective in binding CENP-U as shown above (Fig. 7C,D). Besides, in U2OS-LacO cells transiently expressing Myc-SA2, tethering EGFP-LacI-Scc1 (281–420) to the LacO repeats recruited Sgo1-Flag, but not the Sgo1-ANA-Flag mutant (Fig. 8I,J). Thus, the YNF motif mediates Sgo1 interaction with the binding interface between Scc1 and SA2 in vitro and in cells, as recently reported (Garcia-Nieto et al, 2023).

Taken together, these data indicate that, while Sgo1 plays a major role in protecting sister-chromatid cohesion along the whole chromosomes, CENP-U specifically promotes centromeric cohesion.

## Discussion

The CCAN-based inner kinetochore is well-known for linking centromeric chromatin to the outer kinetochore-bound spindle microtubules. Cohesin-mediated cohesion at sister centromeres resists the poleward forces exerted by spindle microtubules until anaphase onset. We find that the inner kinetochore exerts a non-canonical function in protecting centromeric cohesion, which is achieved through the interaction of the CENP-U subunit of the CENP-OPQUR complex with the Scc1-SA2 sub-complex of cohesin. Since the CENP-OPQUR complex localizes to the inner

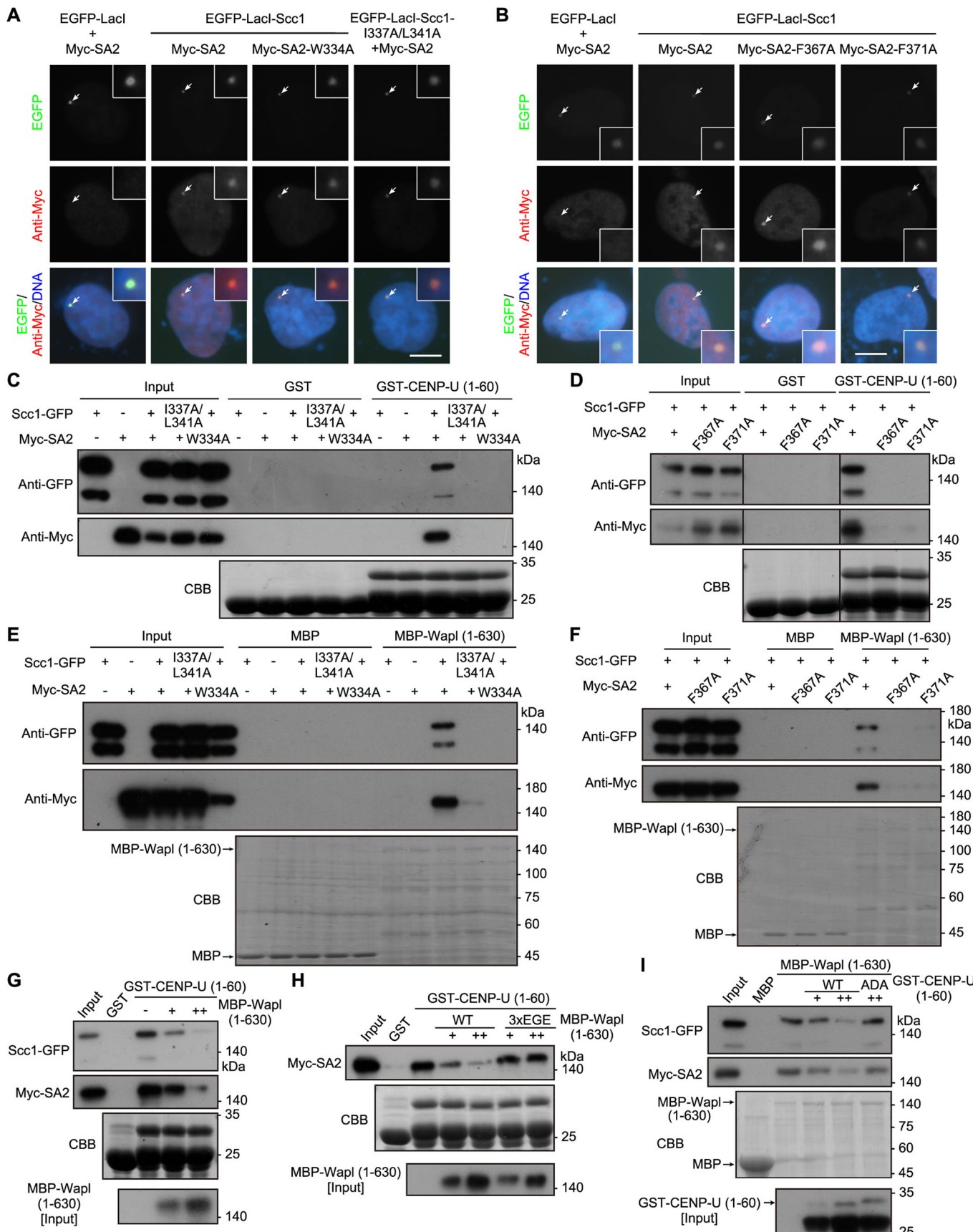

**Figure 7. CENP-U competes with Wapl for binding to the Scc1-SA2 sub-complex.**

(A) U2OS-LacO cells transiently expressing the indicated proteins and the mutants of EGFP-LacI-Scc1 (I337A/L341A) and Myc-SA2 (W334A) were stained with the antibody for the Myc-tag, and DAPI. Example images are shown. (B) U2OS-LacO cells transiently expressing the indicated proteins and the mutants of Myc-SA2 (F367A) and Myc-SA2 (F371A) were stained with the antibody for the Myc-tag and DAPI. Example images are shown. (C) Lysates prepared from HEK-293T cells transiently expressing Scc1-GFP and/or Myc-SA2 in the forms of WT and the indicated mutants were subjected to pull-down with GST or GST-CENP-U (1–60), followed by immunoblotting with antibodies for GFP and the Myc-tag, and CBB staining. (D) Lysates prepared from HEK-293T cells transiently co-expressing Scc1-GFP and Myc-SA2 (WT and the indicated mutants) were subjected to pull-down with GST or GST-CENP-U (1–60), followed by immunoblotting with antibodies for GFP and the Myc-tag, and CBB staining. Irrelevant lanes were removed. (E) Lysates prepared from HEK-293T cells transiently expressing Scc1-GFP and/or Myc-SA2 in the forms of WT and the indicated mutants were subjected to pull-down with MBP or MBP-Wapl (1–630), followed by immunoblotting with antibodies for GFP and the Myc-tag, and CBB staining. (F) Lysates prepared from HEK-293T cells transiently co-expressing Scc1-GFP and Myc-SA2 (WT and the indicated mutants) were subjected to pull-down with MBP or MBP-Wapl (1–630), followed by immunoblotting with antibodies for GFP and the Myc-tag, and CBB staining. (G) Lysates prepared from HEK-293T cells transiently expressing Scc1-GFP and Myc-SA2 were subjected to pull-down with GST-CENP-U (1–60) in the presence of increased amount of eluted MBP-Wapl (1–630) protein, followed by immunoblotting with antibodies for MBP, GFP and the Myc-tag, and CBB staining. (H) Lysates prepared from HEK-293T cells transiently expressing Scc1-GFP and Myc-SA2 were subjected to pull-down with GST-CENP-U (1–60) in the presence of increased amounts of eluted MBP-Wapl (1–630) protein (WT or the 3xEGE mutant), followed by immunoblotting with antibodies for MBP and the Myc-tag, and CBB staining. (I) Lysates prepared from HEK-293T cells transiently expressing Scc1-GFP and Myc-SA2 were subjected to pull-down with MBP-Wapl (1–630) in the presence of increased amounts of eluted GST-CENP-U (1–60) protein (WT or the ADA mutant), followed by immunoblotting with antibodies for GST, GFP, and the Myc-tag, and CBB staining. Data information: The white arrows point to the LacO repeats (A, B). Scale bars, 10 μm (A, B). Source data are available online for this figure.

kinetochore throughout the cell cycle, this study identifies a constitutive kinetochore receptor for cohesin, and reveals the existence of a CENP-U-bound pool of cohesin at the inner kinetochore. Our data pinpoint a model in which the interaction of CENP-U with the inner kinetochore pool of cohesin physically shields it from binding to the cohesin releaser Wapl, thereby strengthening centromeric cohesion (Fig. 9). Future studies are required to overcome the technical challenges to reconstitute the whole complexes of CENP-OPQUR and cohesin to the elucidate more interaction details in vitro.

The exact mechanism by which cohesin rings mediate sister chromatids remains elusive (Gligoris and Lowe, 2016; Haering et al, 2008; Huang et al, 2005; Matityahu and Onn, 2022; Murayama et al, 2018; Nasmyth, 2011; Xiang and Koshland, 2021; Zhang et al, 2008). For example, cohesin may embrace the replicated DNA molecules as a single ring; alternatively, two cohesin rings, in which one cohesin ring embraces a single DNA molecule, may also interact with each other to physically connect the sister chromatids. Moreover, cohesin can even assemble into larger oligomers on DNA. Since the distance between the inner kinetochore on metaphase sister chromatids is around 750 nm on average, which is far longer than the diameter of the cohesin ring (around 50 nm), our data imply that the CENP-U-bound cohesin links sister centromeres together through cooperation between two or more individual cohesin complexes.

The CENP-OPQUR complex-bound pool of cohesin at the inner kinetochore is distinct from the Sgo1-bound pool of cohesin at centromeres and on chromosome arms (Gimenez-Abian et al, 2004; Liang et al, 2019; Liu et al, 2013a; Liu et al, 2015; Liu et al, 2013b; Nakajima et al, 2007). While CENP-U protects cohesin specifically at the inner kinetochore region, Sgo1 plays a prominent role in protecting cohesin both on chromosomes' arms and at centromeres. Our finding that the role of CENP-U in maintaining sister-chromatid cohesion is better revealed in delayed metaphase is in line with the observations that a small amount of cohesin is detected on chromosome arms and that sister chromatids remain associated along the arms until anaphase onset (Chu et al, 2020; Gimenez-Abian et al, 2004; Hirano, 2015; Nakajima et al, 2007; Rieder and Cole, 1999).

The distinct pools of cohesin at the inner kinetochore and at centromeres are analogous to the outer centromere and inner centromere pools of chromosomal passenger complex (CPC) that bind to H2ApT120 and histone H3 phosphorylated at threonine-3 (H3pT3), respectively (Broad et al, 2020; Hadders et al, 2020; Kelly et al, 2010; Liang et al, 2020; Wang et al, 2010; Yamagishi et al, 2010). Learning from the observations that these two pools of the CPC act together to ensure the proper kinetochore-microtubule attachments (Broad et al, 2020; Hadders et al, 2020; Liang et al, 2020), future studies are required to explore the potential cross-talk between the CENP-U-bound cohesin and the Sgo1-bound cohesin.

In budding yeast, the COMA complex comprising the Ame1/Okp1[CENP-U/Q] and Ctf19/Mcm21[CENP-P/O] heterodimers is equivalent to the mammalian CENP-O/P/Q/U complex. Phosphorylation of the Ctf19 (CENP-P in mammals) subunit by the conserved kinase DDK creates a kinetochore binding site for a conserved, positively charged surface patch of the cohesin loader Scc4 (MAU2 in mammals) in complex with Scc2 (NIPBL in mammals) (Hinshaw et al, 2017). Since the DDK-phosphorylated residues of Ctf19 seem not conserved in higher eukaryotes, it remains to be determined whether there is an unknown protein at the kinetochore, whose phosphorylation by DDK (or other kinases) can be recognized by the conserved surface patch of MAU2 to recruit the cohesin loader to the kinetochore. Regardless, our study identifies CENP-U as a direct kinetochore anchor protein for the cohesin core, and reveals an unexpected, non-canonical role for the inner kinetochore in retaining cohesin and protecting centromeric sister-chromatid cohesion.

Interestingly, the FDF motif of CENP-U is not conserved in vertebrates. For example, we find that the FDF motif is not present in chicken CENP-U, and that chicken CENP-U does not interact with the Scc1-SA2 sub-complex. Using chicken DT-40 cells, it was previously shown that CENP-U depletion caused a defect in the recovery of mitosis progression from spindle damage (Hori et al, 2008). It is unknown whether this phenotype is due to cohesion defects, or to the impaired function of the CENP-O/P/Q/U/R subunits in modulating kinetochore-microtubule attachment (Amaro et al, 2010; Bancroft et al, 2015; Hua et al, 2011; Pesenti et al, 2018; Sedzro et al, 2022). If the former is the case, it will be interesting to examine whether the CENP-O, P, Q, or R subunit of the vertebrate CENP-OPQUR complex contains the F/Y-x-F motif that can bind to the Scc1-SA2 sub-complex. We envision that the divergent molecular mechanisms might carry out similar activities in various organisms. In other words, it is likely that the process is

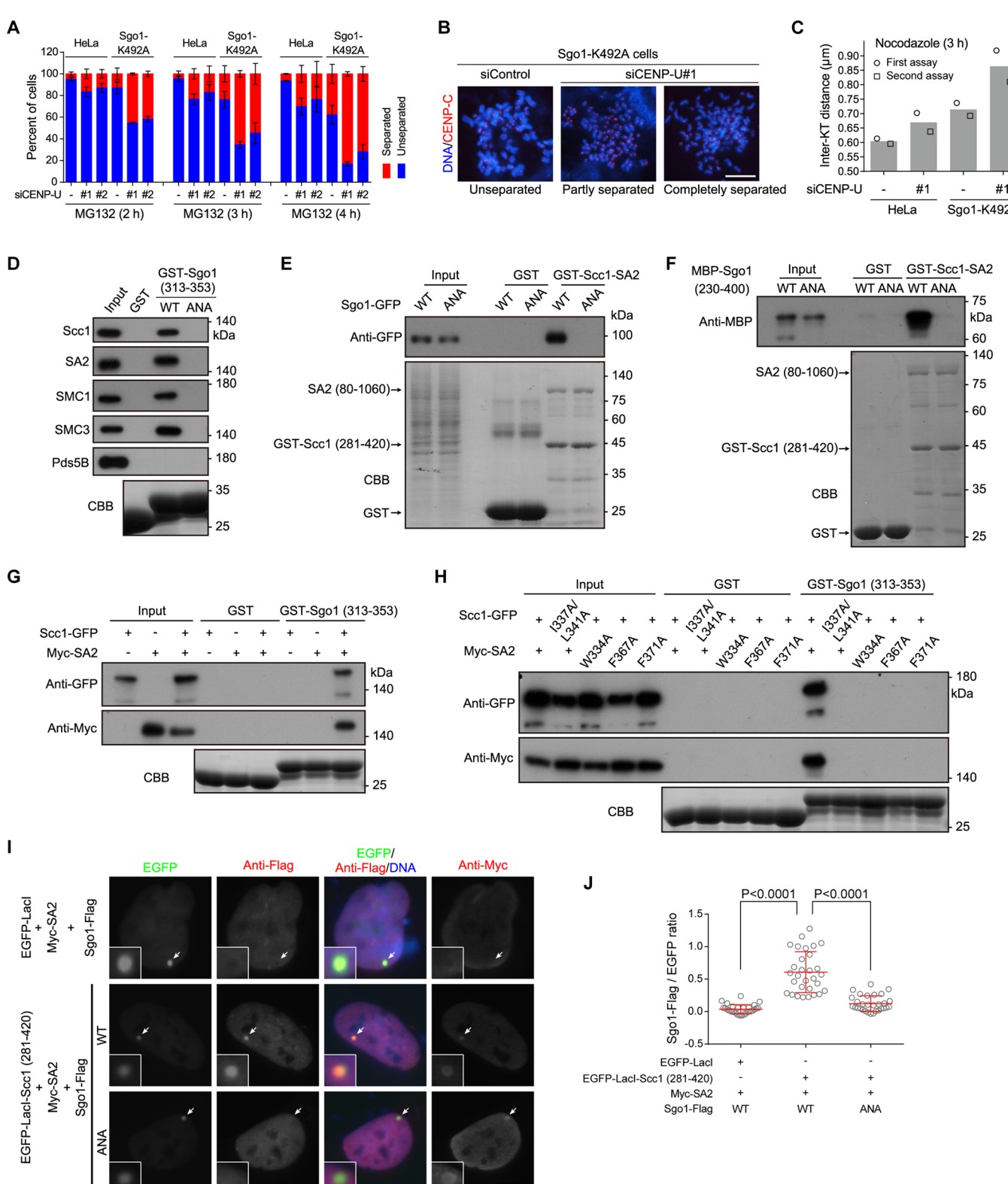

evolutionarily conserved, but not the specific molecular mechanisms. Indeed, it is increasingly recognized that centromeres (both the DNA and proteins) undergo rapid evolution—the centromere paradox (Saint-Leandre and Levine, 2020). Future studies are required to determine whether a certain component of the inner

kinetochore in vertebrates interacts with a certain subunit of the cohesin complex through a yet-to-be-identified motif.

We find that CENP-U uses the FDF motif to directly interact with the binding interface between Scc1 and SA2, which is similar to the Y-x-F motif-dependent binding of CTCF to Scc1-SA2 (Li et al, 2020).

**Figure 8.   CENP-U and Sgo1 additively contribute to the strength of centromeric cohesion.**

(A, B) Control HeLa cells and Sgo1-K492A mutant cells were transfected with control siRNA or CENP-U siRNA. At 48 h post-transfection, cells were treated with MG132, then mitotic cells were collected at the indicated time points to prepare chromosome spreads, and then stained with the CENP-C antibody and DAPI. The percentage of cells in which the majority of sister chromatids was separated or unseparated was determined in over 300 cells for each condition from three independent experiments (A). Example images are shown (B). (C) HeLa cells and Sgo1-K492A cells were transfected with control siRNA or CENP-U siRNA. At 48 h post-transfection, cells were subjected to nocodazole treatment for 3 h. Mitotic chromosome spreads were stained with the CENP-C antibody and DAPI. The inter-KT distance was measured on over 1000 chromosomes in 20 cells. The means and individual data points from two independent experiments are plotted. Data for each individual experiment and example images are shown in Fig. EV5D,E. (D) HeLa cell lysates were subjected to pull-down with GST, GST-Sgo1 (313–353) (WT or the ANA mutant), followed by immunoblotting with antibodies for Scc1, SA2, SMC1, SMC3, and Pds5B and CBB staining. (E) Lysates prepared from HEK-293T cells transiently expressing Sgo1-GFP in the forms of WT and ANA were subjected to pull-down with GST or GST-Scc1 (281–420)-SA2 (80–1060), followed by immunoblotting with the antibody for GFP and CBB staining. (F) The GST-Scc1 (281–420)-SA2 (80–1060) sub-complex was used to pull down MBP-Sgo1 (230–400) in the forms of WT and ANA, followed by immunoblotting with the antibody for MBP and CBB staining. (G) Lysates prepared from HEK-293T cells transiently expressing Scc1-GFP and/or Myc-SA2 were subjected to pull-down with GST or GST-Sgo1 (313–353), followed by immunoblotting with antibodies for GFP and the Myc-tag, and CBB staining. (H) Lysates prepared from HEK-293T cells transiently expressing Scc1-GFP and Myc-SA2 in the forms of WT and the indicated mutants were subjected to pull-down with GST or GST-Sgo1 (313–353), followed by immunoblotting with antibodies for GFP and the Myc-tag, and CBB staining. (I, J) U2OS-LacO cells transiently expressing the indicated proteins were stained with antibodies for the Flag-tag, Myc-tag, and DAPI. Example images are shown (I). The white arrows point to the LacO repeats. The fluorescence intensity ratio of Sgo1-Flag/ EGFP at the LacO repeats was quantified in 30 cells for each condition, with statistics being performed using unpaired Student's t-test (J). Data information: Means and SDs are shown (A, J). Scale bars, 10 μm (B, I). Source data are available online for this figure.

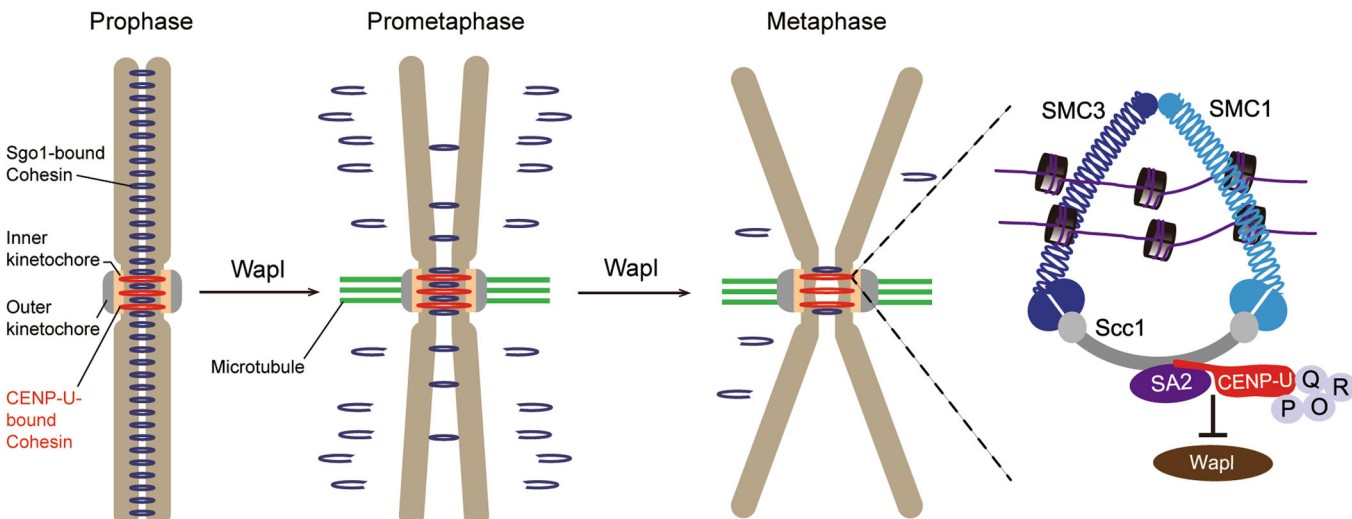

**Figure 9.   Model for the role of the inner kinetochore in protecting sister-chromatid cohesion at mitotic centromeres.**

The CENP-OPQUR complex anchors a subset of cohesin at the inner kinetochore through direct binding of its CENP-U subunit to the composite interface between Scc1 and SA2, which antagonizes the competitive binding of Wapl to Scc1-SA2. Sgo1-bound cohesin is both distributed on chromosome arms and enriched at centromeres.

CTCF enables the formation of chromatin loops by cohesin, which can be released by Wapl (Davidson and Peters, 2021; Haarhuis et al, 2017; Hoencamp and Rowland, 2023; Wutz et al, 2017). Whether the CENP-U-cohesin interaction helps organize the higher-order chromatin loops at centromeres is worth investigating in the future.

# Methods

## Cell lines, plasmids, siRNA, transfection, and drug treatments

All cells were cultured in DMEM supplemented with 1% penicillin/ streptomycin and 10% FBS (Gibco) and maintained at 37 °C with 5% $CO_2$. U2OS-LacO cells, kindly provided by Dr. David Spector (Cold Spring Harbor Laboratory, USA), were maintained in the presence of 100 μg/ml hygromycin (Sigma). HeLa cells stably expressing H2B-GFP or CENP-U-GFP were maintained in 2 μg/ml blasticidin (Sigma). HeLa cells stably expressing CENP-U-GFP (WT, the ADA mutant, and the T78A mutant) were isolated and maintained in 2 μg/ml blasticidin (Sigma).

To make the EGFP-LacI fusion constructs, the PCR products of full-length Scc1 or the fragment encoding residues 281–420 were inserted into the BamHI site of pSV2-EGFP-LacI. To make the CENP-U-EGFP-LacI constructs, the PCR fragments of CENP-U were inserted into the HindIII site of pCDNA3.1-3xMyc-LacI, and then the 3xMyc tag was replaced by EGFP. The pBos-CENP-U-GFP constructs were made by replacing the H2B fragment in pBos-H2B-GFP with the KpnI/BamHI digested PCR fragments encoding CENP-U (full-length or the fragments of 1–100 and 101–418). pBos-GgCENP-U-GFP was constructed similarly with the chicken CENP-U cDNA synthesized by ZENTA Life Sciences. The pBos-Scc1-GFP constructs were constructed using the PCR fragments encoding full-length Scc1 or fragments encoding residues 1–281,

281–420, and 420–631. pBos-Sgo1-GFP was constructed by replacing the H2B fragment in pBos-H2B-GFP with the KpnI/BamHI digested PCR fragments encoding Sgo1. SFB-CENP-U and SFB-CENP-Q were constructed by transferring the CENP-U or CENP-Q cDNA into a Gateway-compatible destination vector which harbors an N-terminal triple SFB-tag. Myc-SA2 was similarly constructed into the vector harboring an N-terminal Myc-tag. The GST-CENP-U constructs were made by subcloning the PCR fragments encoding CENP-U residues 101–401, 201–418, and 1–418 into the BamHI site of pGEX-4T1 (GE Healthcare), and the truncations encoding CENP-U residues 1–39, 1–50, 1–60, 1–100, and 1–200 were made by mutating related codon to the stop codon TAA. The GST-Sgo1 (313–353) construct was made by subcloning the PCR fragments encoding Sgo1 (313–353) into the BamHI site of pGEX-4T1 (GE Healthcare). MBP fusion constructs were made by subcloning the PCR fragments encoding H2A, Wapl (1–630), Sgo1 (230–400), and CENP-U (1–200) into the BamHI site of pMal-C2E (New England Biolabs). The Wapl-Flag, Sgo1-Flag, and SA2-Flag constructs were made by inserting the PCR fragments of Wapl, Sgo1, and SA2 into the NotI/SfaAI sites of pEF-IRES-P-EGFP-Flag-6xHis, respectively. The Wapl-Myc and Scc1-Myc constructs were made by inserting the PCR fragments of Wapl or Scc1 into the KpnI/EcoRV sites of pEF6-Myc-6xHis. To make the GST-Scc1 (281–420)-SA2 (80–1060) constructs, the PCR fragments of Scc1 was inserted into the BamHI/SalI sites of pGEX-6P-2rbs (Amersham Pharmacia Biotech), then the PCR fragments of SA2 was inserted into the BglII/XhoI sites. All point mutations were introduced with the QuikChange II XL site-directed mutagenesis kit (Agilent Technologies) or by the MultiF Seamless Assembly mix kit (ABclonal Biotechnology). All plasmids were sequenced to verify desired mutations and the absence of unintended mutations.

The following siRNA duplexes were selected from previous publications, and were ordered from Integrated DNA Technologies (IDT) or RiboBio were used: siCENP-U#1 (5′-GAAAGCCAUCUGCGAAAUAdTdT-3′); siCENP-U#2 (5′-GAAAAUAAGUACACAACGUdTdT-3′); siWapl (5′-CGGACUACCCUUAGCACAAdTdT-3′); siScc1 (5′-AUACCUUCUUGCAGACUGUdTdT-3′); siSA2 (5′-CCGAAUGAAUGGUCAUCACdTdT-3′); siSMC1 (5′-GGAAGAAAGUAGAGACAGAdTdT-3′); siSMC3 (5′-GGAGGGCAGUCAGUCUCAAGAUGAA-3′); siSgo1 (5′-GAGGGGACCCUUUUACAGAdTdT-3′); siCENP-Q (5′-GGUCUGGCAUUA-CUACAGGAAGAAA-3′).

Plasmid transfection was done with FuGENE 6 (Promega) or Lipofectamine 2000 (Thermo Fisher Scientific). siRNA transfection was done twice with Oligofectamine (Invitrogen) and Lipofectamine RNAiMAX (Thermo Fisher Scientific) in a 24 h interval. Cells were subjected to drug treatment or analysis at 48 and 24 h after transfection with siRNA and plasmids, respectively. Cells were arrested in S-phase by single thymidine (2 mM, Sigma) treatment. Cells were arrested in monopolar mitosis with STLC (5 μM, Tocris Bioscience), were arrested in metaphase with MG132 (10–20 μM, Selleckchem), or were arrested in mitosis with Apcin (250 μM, MCE) or nocodazole (100 ng/mL, Selleckchem). Mitotic cells were collected by selective detachment with "shake-off".

## Antibodies

Rabbit polyclonal antibodies used were GFP (A11122, Invitrogen), GAPDH (14C10, Cell Signaling Technology/CST), Scc1 (ab992, Abcam), SA2 (Rabbit mAb #5882, CST), SMC1 (A300-055A, Bethyl Laboratories), SMC3 (A300-060A, Bethyl Laboratories), Pds5B (A300-537A, A300-538A, Bethyl Laboratories), GST (G7781, Sigma), Wapl (A300-268A, Bethyl), Sororin (ab192237, Abcam), Flag (GenScript), H2ApT120 (Active motif). Rabbit anti-CENP-U polyclonal antibodies were produced by immunization with the synthetic peptide EPNVKETYDSSSLP (Chen et al, 2021). Mouse monoclonal antibodies used were to α-Tubulin (T-6074, Sigma), Myc-tag (4A6, Millipore), Flag-tag (M2, Sigma), MBP (E8032, New England Biolabs), SA2 (sc-81852, Santa Cruz Biotechnology), GFP (M20004, Abmart), Wapl (M221-3, MBL), Sgo1 (ab58023, Abcam or sc-393993, Santa Cruz Biotechnology), Plk1 (ab17057, Abcam). The anti-human centromere autoantibody (ACA) was from Immunovision. Guinea pig polyclonal antibodies against CENP-C were from MBL (PD030). Secondary antibodies for immunoblotting were goat anti-rabbit or horse anti-mouse IgG-HRP (CST). Secondary antibodies for immunostaining were donkey anti-rabbit IgG-Alexa Fluor 488 or Cy3 (Jackson ImmunoResearch); anti-mouse IgG-Alexa Fluor 488 or 546 (Invitrogen) or Cy5 (Jackson ImmunoResearch); Goat anti-guinea pig IgG-Alexa Fluor 488 or 647 (Invitrogen); anti-human IgG-Alexa Fluor 647 (Jackson ImmunoResearch).

## CRISPR/Cas9-mediated editing of CENP-U gene in HeLa cells

Single guide RNA (sgRNA) for the human CENP-U gene was ordered as oligonucleotides, annealed and cloned into the dual Cas9 and sgRNA expression vector pX330 (Dr. Feng Zhang laboratory, Addgene, #42230) with BbsI sites. The plasmids were transfected into HeLa cells using Fugene 6 (Promega) according to the manufacturer's protocol. To make the AVAAA mutations in endogenous CENP-U, plasmids encoding Cas9 and sgRNA targeting a sequence close to that encoding the DVFDF motif of CENP-U were co-transfected into HeLa cells with a single-stranded oligodeoxynucleotide (ssODN) as the HDR repair template. Transfected cells were briefly treated with the DNA ligase IV inhibitor Scr7 (5 μM) to increase the efficiency of HDR-mediated genome editing. After 48 h incubation, the cells were split individually to make a clonal cell line with brief selection using 1 μg/ml puromycin for 3 days. Individual clones were isolated, the genomic DNA was PCR-amplified, subcloned into pBluescriptII (-) (Agilent Technologies), and then 20 positive bacterial colonies were sequenced for CENP-U-AVAAA. CENP-U-AVAAA was obtained using sgRNAs targeting the sequences of 5'-ATTGACGTGTTC-GACTTTCC-3'. The sgRNA-resistant ssODN with the AVAAA mutations was ordered from IDT (5'-CACAGGTCTGAGGGCG-CAAGACGTTCAAAGAACACTTTAGAAAGAACACATTCCAT-GAAAGATAAAGCTGGTCAAAAGTGCAAGGCTATTGCCGT GGCCGCCGCTCCTGATAATTCTGATGTCTCAAGCATTGG-CAGGCTGGGTGAAAATGAGAAAGATGAAGAAACTTATGA-GACCTTTGATCCTCCT-3').

## Fluorescence microscopy and quantification

HeLa cells and U2OS-LacO cells were fixed with 2% PFA in PBS for 10 min and then extracted with 0.5% Triton X-100 for 5 min. Fixed cells were blocked with 3% BSA/PBS and then stained with primary antibodies for 1–2 h and secondary antibodies for 1 h, all with 3%

BSA in PBS at room temperature. DNA was stained for 5 min with DAPI. For chromosome spreads, HeLa cells were treated with MG132 (10 µM) for 2–8 h, Apcin (250 µM) for 6 h, or nocodazole (100 ng/mL) for 3 h. Mitotic cells were obtained by selective detachment and then incubated in hypotonic buffer (75 mM KCl or 0.25x PBS) at room temperature for 15 min. After attachment to glass coverslips by Cytospin at 1500 rpm for 5 min, chromosome spreads were fixed with 2% PFA/PBS for 15–20 min, extracted with 0.5% Triton X-100/PBS for another 10 min, and then subjected to blocking with 3% BSA/PBS and subsequent immunostaining with primary antibodies for 2 h. For the immunofluorescence of Sgo1 (Fig. EV5A), HeLa cells were treated with nocodazole (100 ng/ml) for 2 h after a single thymidine release. Mitotic cells were obtained by selective detachment and then incubated in hypotonic buffer (0.5x PBS) at room temperature for 5 min. Fluorescence microscopy was carried out at room temperature using a Nikon ECLIPSE Ni microscope with a Plan Apo Fluor 60X Oil (NA 1.4) objective lens and a Clara CCD (Andor Technology).

For quantification of fluorescent intensity, all images of similarly stained experiments were acquired with identical illumination settings, and cells expressing comparable levels of exogenous protein were selected and analyzed using ImageJ. To quantify the relative enrichment of proteins of interest at the LacO transgene array in U2OS-LacO cells, the average pixel intensity of antibody staining, within circles encompassing fluorescent signal of the EGFP-LacI fusion protein at LacO transgene array, and in the nearby nucleus, was determined. After background correction, the ratio of average immunostaining intensity at LacO repeats versus that in the nuclei was calculated. The inter-KT distance was measured with Nikon ECLIPSE Ni, using CENP-C immunofluorescence signal on over 25 kinetochores per cell in at least 20 cells per experiment. Sister chromatids that were obviously separated were not selected for the measurement.

## Time-lapse live cell imaging

Time-lapse live cell imaging was carried out with the GE DV Elite Applied Precision DeltaVision system (GE Healthcare) equipped with Olympus oil objectives of 60X (NA 1.42) Plan Apo N and an API Custom Scientific complementary metal-oxide semiconductor camera, and Resolve3D softWoRx imaging software. HeLa cells stably expressing H2B-GFP were plated in four-chamber glass-bottomed 35-mm dishes (Cellvis) coated with poly-D-lysine, and filmed in a climate-controlled and humidified environment (37 °C and 5% CO$_2$). Images were captured every 3 min. The acquired images were processed using Adobe Photoshop and Adobe Illustrator.

## Pull-down assays, immunoprecipitation, and immunoblotting

For GST or MBP fusion-protein pulldown with cell lysates, HeLa cells were lysed in P150 buffer (50 mM Tris-HCl, pH 7.5, 150 mM NaCl, 1% Triton X-100, 10 mM MgCl$_2$, 5 mM EDTA) in the presence of 1 mM dithiothreitol (DTT), protease inhibitor cocktail (P8340, Sigma), 1 mM PMSF, 0.1 µM okadaic acid (Calbiochem), 10 mM NaF, and 20 mM β-glycerophosphate and Benzonase (GenScript). After removal of insoluble materials by high-speed centrifugation, lysates were precleared with glutathione Sepharose

4B beads (GE Healthcare) or Amylose Resin (New England Biolabs), then incubated with beads-immobilized GST fusion proteins for 4 h. Beads were then washed 3 times with the lysis buffer, boiled in standard SDS sample buffer, and subject to immunoblotting. For competitive binding assays (Fig. 7G–I), the cell lysates were mixed with recombinant MBP-Wapl (1–630) and GST-CENP-U (1–60) for pull-down by GST-CENP-U (1–60) and MBP-Wapl (1–630), respectively. For MBP-CENP-U (1–200) pulldown of eluted GST-Scc1 (281–420)-SA2 (80–1060), the binding (2 h) and wash (three times, 5 min each) were carried out in P150 buffer (50 mM Tris-HCl, pH 7.5, 150 mM NaCl, 1% Triton X-100, 10 mM MgCl$_2$, 5 mM EDTA). SDS-PAGE and immunoblotting were carried out with standard procedures using samples prepared in a standard SDS sample buffer.

For the co-immunoprecipitation, cells were lysed in P50 buffer (50 mM Tris-HCl, pH 7.5, 50 mM NaCl, 1% Triton X-100, 10 mM MgCl2, 5 mM EDTA) in the presence of 1 mM DTT, protease inhibitor cocktail (P8340, Sigma), 1 mM PMSF, 0.1 µM okadaic acid (Calbiochem), 10 mM NaF, and 20 mM β-glycerophosphate and Benzonase (GenScript). After removal of insoluble materials by high-speed centrifugation, lysates were precleared with rProtein A/G beads (Smart-Lifesciences; Cat. No. SA032100). Lysates were then incubated with Anti-GFP Affinity beads (Smart-Lifesciences; Cat. No. SA070005) or control rProtein A/G beads for 4 h at 4 °C. Beads were washed three times with P50 buffer, boiled in standard SDS sample buffer, and subject to immunoblotting.

## Protein expression and purification

The plasmids encoding the GST or MBP fusion proteins were transformed into BL21 (DE3) competent cells (Stratagene). Cells were grown in LB broth under antibiotic selection at 37 °C until OD$_{600}$ at 0.6–0.8, and protein expression was induced with 0.4 mM IPTG at 16 °C for 16 h. Cells were lysed by sonication in buffer A (20 mM Tris-HCl, pH 8.0, 100 mM NaCl, 1 mM EDTA, 1% Triton X-100; for GST fusion proteins) or buffer B (50 mM Tris-HCl, pH 7.5, 300 mM NaCl, 1 mM EDTA, 0.5% Triton X-100; for MBP fusion proteins). The lysate was clarified by centrifugation and incubated with Glutathione Sepharose 4B beads (GE Healthcare) or Amylose Resin (New England Biolabs) in lysis buffer. The resin was washed with lysis buffer and eluted with 100 mM glutathione or 10 mM maltose.

To purify the GST-Scc1 (281–420)-SA2 (80–1060) complex for crystallization, recombination proteins were induced with 0.4 mM IPTG at 16 °C for 16 h in *E. coli* BL21 (DE3), following 5 h incubation at 37 °C. Cells were resuspended in lysis buffer (1xPBS, 0.02% Triton X-100, and 0.5 mM TCEP). After breaking by the high-pressure homogenizer and centrifugation, the supernatant was applied onto Glutathione Beads (GE Healthcare). The GST tag was cleaved by PreScission Protease in QA buffer (50 mM Tirs, pH 7.5, 100 mM NaCl, and 0.5 mM TCEP) during overnight incubation at 4 °C. Cleaved protein complex was applied to a HiTrap Q HP column (GE Healthcare) in QA buffer and eluted via a linear gradient of QB buffer (50 mM Tris, pH 7.5, 1 M NaCl, and 0.5 mM TCEP) and further purified using a Superdex 200 Increase 10/300 GL column (GE Healthcare) in the purification buffer (20 mM Tris, pH 7.7, 300 mM NaCl, and 5 mM TCEP). Purified GST-Scc1 (281–420) and SA2 (80–1060) at a ratio of 1:1 was concentrated to 4 mg/ml for crystallization.

## Crystallization and structure determination

The crystallization of the Scc1-SA2 complex was done as previously described (Hara et al, 2014; Li Y. et al, 2020). The crystals of the Scc1-SA2 complex were grown by hanging drop vapor diffusion at 18 °C and mixing equal volumes of protein and crystallization solution containing 0.03 M CaCl$_2$, 0.03 M MgCl$_2$, 0.1 M MOPS-HEPES, pH 7.5, 10% PEG 8000, 20% ethylene glycol. Crystals were soaked for 24 h with a 500 μM peptide (synthesized by GL Biochem) encompassing CENP-U amino acid residues 40–50 PIDVFDFPDNS. The X-ray diffraction data were collected at beamline BL18U of the Shanghai Synchrotron Radiation Facility (SSRF) at wavelengths of 0.9792 Å, using a PILATUS 6 M detector. Data were processed with the XDS Package as previously described (Kabsch W. Acta Crystallographica Section D, 2010, 66).

The structure of the Scc1-SA2-CENP-U complex (PDB: 8K4D) was determined by molecular replacement method with PHASER in CCP4 suite, using SA2-Scc1-CTCF (PDB code: 6QNX) as the search model. A final model was produced by iterative rounds of manual model-building in Coot and refinement using REFMAC5. The statistics of data collection and refinement are shown in Table EV1.

## Mass spectrometry

For data shown in Dataset EV1, after pull-down assays, GST and GST-CENP-U (1–200)-binding proteins were subjected to SDS-PAGE. Protein gel bands were cut into small pieces and de-stained with buffer (25 mM NH$_4$HCO$_3$/25% methanol, pH 8.0). Proteins were reduced with 10 mM DTT for 60 min at 56 °C and alkylated with 55 mM iodoacetamide for 45 min. Gel pieces were washed twice with digestion buffer (50 mM NH$_4$HCO$_3$, pH 8.0), dehydrated with acetonitrile, and then dried with speed-vac. Gel pieces were rehydrated with trypsin solution (10 ng/μl sequencing grade modified trypsin, 50 mM NH$_4$HCO$_3$, pH 8.0) and incubated overnight at 37 °C. Digested peptides were extracted from gel pieces sequentially with elution buffer 1 (50% acetonitrile, 5% formic acid) and elution buffer 2 (75% acetonitrile, 0.1% formic acid). Gel pieces were dehydrated twice with acetonitrile, and all of supernatant were combined. Peptides solution was dried with speed-vac, then digested peptides were resuspended with 5% formic acid, desalted with StageTip, and then loaded on analytical column (75 × 15 cm, 1.9 μm C18, 1 μm tip) with Easy-nLC 1200 system. Samples were analyzed with a 60 min gradient at a flow rate of 300 nl/min as follows: 3–6% B for 2 min, 6–26% B for 38 min, 26–34% B for 12 min, 34–90% B for 3 min, 900% B for 3 min. Orbitrap Exploris 480 mass spectrometer was operated in data-dependent mode with one full MS scan at R = 60,000 (m/z 200), followed by twenty HCD MS/MS scans at R = 15,000, NCE = 30, with an isolation width of 1.6 m/z. Precursors +2–+5 were included; exclusion of isotopes was enabled; dynamic exclusion was set to 30 s. Mass spectrometry data were searched by MaxQuant.

## Quantitative RT-PCR analysis

Total cellular RNAs were extracted with Trizol (Thermo Fisher Scientific), then incubated on ice for 10 min and centrifuged at 12,000 rpm for 10 min. Then, the supernatant was mixed with chloroform, followed by a vigorous shake and incubation at room temperature. After centrifugation, the upper aqueous phase was mixed with an equal volume of isopropanol and then centrifuged. The RNA pellet was washed twice with 75% ethanol and dissolved in DEPC water. Next, 1000 ng of RNA was reverse-transcribed using PrimeScript RT Master Mix (Takara, RR047A). Quantitative PCR was carried out using SYBR qPCR SuperMix (Vazyme, Q711-03) according to the manufacturer's instructions. The CENP-U primer sequences are pair #1 (5′-ACCCACCTAGAGCATCAA-CAA-3′ and 5′-ACTTCAATCATACGCTGCCTTT-3′), pair #2 (5′-ATGAACTGCTTCGGTTAGAGC-3′ and 5′-TATTTCGCAGAT GGCTTTCGG-3′). GAPDH mRNA was used as an internal control with the primer pair (5′-GGAGCGAGATCCCTCCAAAAT-3′ and 5′-GGCTGTTGTCATACTTCTCATGG-3′).

## Statistical analysis and sequence alignment

Statistical analyses were performed with a two-tailed unpaired Student's *t*-test or one-way ANOVA in GraphPad Prism 7. A *P* value of <0.05 was considered significant. The multiple alignment of CENP-U or Sgo1 sequences was performed online using CLUSTAW (https://www.genome.jp/tools-bin/clustalw), and the resulting figures were prepared online using ESPript 3.0 (https://espript.ibcp.fr/ESPript/ESPript/).

## Data availability

The atomic model is available in the PDB database under access code SA2-Scc1-CENP-U (PDB 8K4D).

The source data of this paper are collected in the following database record: biostudies:S-SCDT-10_1038-S44318-024-00104-6.

## Peer review information

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

## Acknowledgements

The authors would like to thank members of Haiyan Yan Laboratory and Fangwei Wang Laboratory for commenting on the manuscript. This work was supported by grants from National Natural Science Foundation of China

(32025011, 32270772, 32061160470, 31970672, 32270877, 31925013, 32100583, and 32200580), National Key Research and Development Program of China (2022YFA1303102, 2022YFA1105203, and 2017YFA0503602), Natural Science Foundation of Zhejiang Province (LZ24C070001, LZ19C070001, and LY17C070003), the China Postdoctoral Science Foundation (2020M681838, 2022M720126, and 2021M692829), and the Royal Society Newton Advanced Fellowship (NA140075). We additionally acknowledge the Life Sciences Institute Core Facility and Dr. Cheng Ma of the Core Facility of Zhejiang University School of Medicine for technical assistance.

## Author contributions

**Lu Yan**: Data curation; Software; Validation; Investigation; Visualization; Methodology; Writing—review and editing. **Xueying Yuan**: Data curation; Software; Validation; Investigation; Visualization; Methodology; Writing—review and editing. **Mingjie Liu**: Investigation; Methodology. **Qinfu Chen**: Data curation; Software; Funding acquisition; Validation; Investigation; Visualization; Methodology. **Miao Zhang**: Funding acquisition; Investigation; Methodology. **Junfen Xu**: Resources; Methodology. **Ling-Hui Zeng**: Resources; Methodology. **Long Zhang**: Resources; Funding acquisition; Methodology. **Jun Huang**: Resources; Software; Methodology. **Weiguo Lu**: Resources; Software; Methodology. **Xiaojing He**: Resources; Data curation; Software; Supervision; Funding acquisition; Validation; Investigation; Visualization; Methodology; Writing—review and editing. **Haiyan Yan**: Conceptualization; Resources; Data curation; Software; Formal analysis; Supervision; Funding acquisition; Validation; Investigation; Visualization; Methodology; Writing—original draft; Project administration; Writing—review and editing. **Fangwei Wang**: Conceptualization; Resources; Data curation; Software; Formal analysis; Supervision; Funding acquisition; Validation; Investigation; Visualization; Methodology; Writing—original draft; Project administration; Writing—review and editing.

Source data underlying figure panels in this paper may have individual authorship assigned. Where available, figure panel/source data authorship is listed in the following database record: biostudies:S-SCDT-10_1038-S44318-024-00104-6.

## Disclosure and competing interests statement

The authors declare no competing interests.

# Expanded View Figures

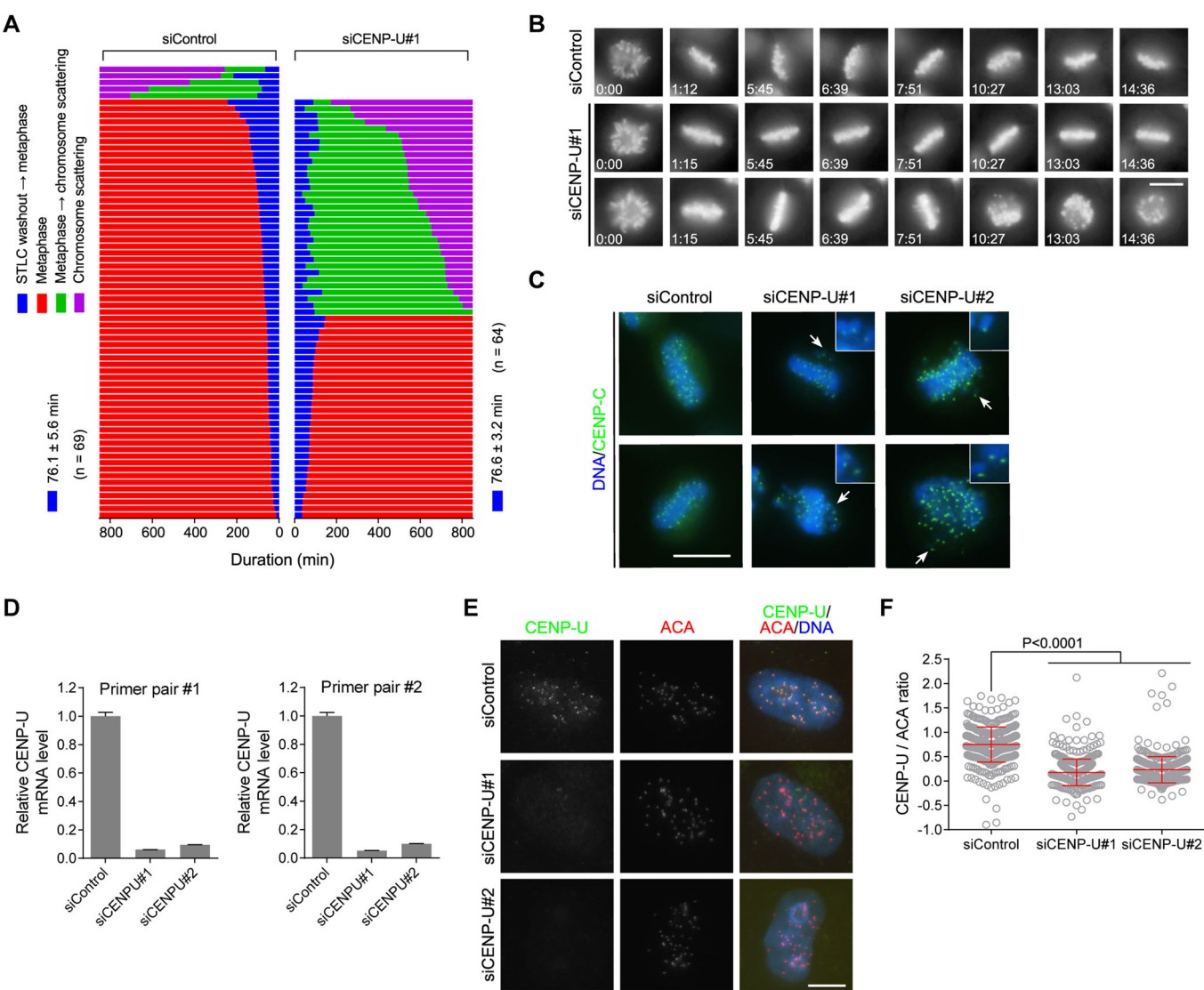

**Figure EV1.  CENP-U strengthens centromeric cohesion and promotes metaphase sister-chromatid cohesion.**

(A, B) HeLa cells stably expressing H2B-GFP were transfected with control siRNA or CENP-U siRNA, followed by synchronization in S-phase with thymidine treatment for 20 h, and then released into fresh medium. At 7 h after thymidine release, cells were treated for 5 h with STLC, then mitotic cells were collected and released into fresh medium containing MG132 followed by live imaging of mitosis progression for 879 min. The time from STLC washout to metaphase chromosome alignment, and from metaphase to chromosome scattering, was determined and profiled (A). The selected frames of the movies are shown (B). The time stated in hours: minutes. See Movies EV1, EV2. (C) HeLa cells were transfected with control siRNA or CENP-U siRNA. At 48 h after siRNA transfection, cells were treated with MG132 for 6 h and then stained with the CENP-C antibody and DAPI. Example images are shown. Arrows point to misaligned chromosomes with single CENP-C foci. (D) HeLa cells were transfected with control siRNA or CENP-U siRNA. At 48 h post-transfection, total RNA was extracted and subjected to quantitative RT-PCR analysis using two pairs of CENP-U primers. The level of CENP-U mRNA in CENP-U-depleted cells relative to that in control HeLa cells were determined in three independent experiments. (E, F) HeLa cells were transfected with control siRNA or CENP-U siRNA. At 48 h post-transfection, cells were stained with anti-human centromere autoantibody (ACA) and the CENP-U antibody. Example images are shown (E). The immunofluorescence intensity ratio of CENP-U/ACA was determined from ~400 centromere regions in 20 cells, with statistics being performed using unpaired Student's *t*-test (F). Data information: Means and SDs are shown (D, F). Scale bars, 10 μm (C, E).

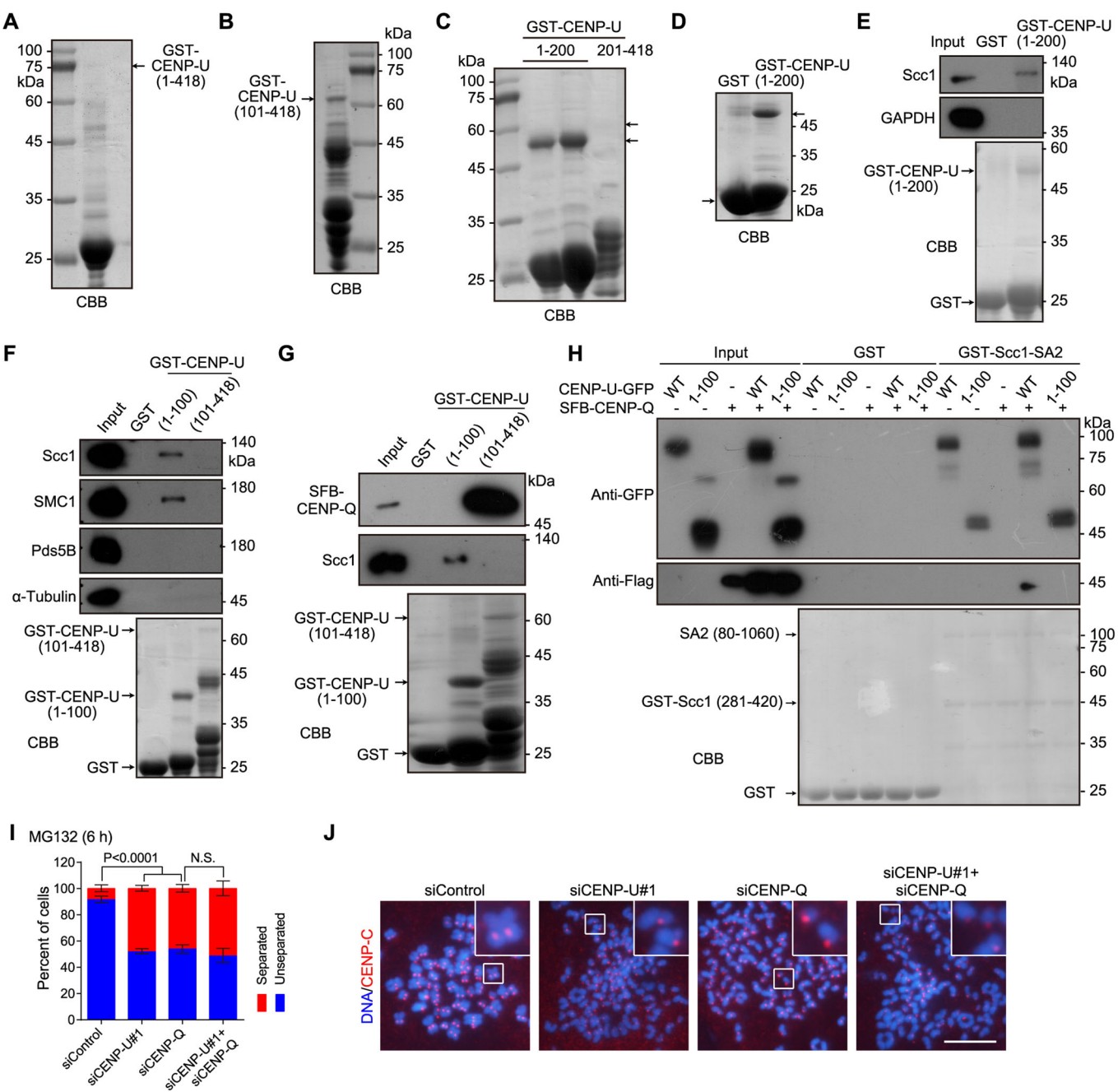

**Figure EV2. CENP-U directly interacts with the Scc1-SA2 sub-complex of cohesin.**

(A–C) CBB staining of GST-CENP-U (1–418) (A), GST-CENP-U (101–418) (B), GST-CENP-U (1–200), and GST-CENP-U (201–418) (C), which were expressed and purified in *E. coli*. The lower arrow points to the GST-CENP-U (1–200) protein. The upper arrow points to the theoretical size/position of the GST-CENP-U (201–418) protein which is undetectable. (D) CBB staining of GST and GST-CENP-U (1–200) was used for pull-down assay and MS analysis as shown in Dataset EV1. The arrow points to the GST-CENP-U (1–200) protein. (E) HeLa cell lysates were subjected to pull-down with GST or GST-CENP-U (1–200), followed by immunoblotting with antibodies for Scc1 and GAPDH and CBB staining. (F) HeLa cell lysates were subjected to pull-down with GST, GST-CENP-U (1–100), or GST-CENP-U (101–418), followed by immunoblotting with antibodies for Scc1, SMC1, Pds5B, and α-tubulin, and CBB staining. (G) Lysates prepared from HEK-293T cells transiently expressing SFB-CENP-Q were subjected to pull-down with GST, GST-CENP-U (1–100), or GST-CENP-U (101–418), followed by immunoblotting with antibodies for Scc1 and the Flag-tag, and CBB staining. (H) Lysates prepared from HEK-293T cells transiently expressing the indicated proteins of CENP-U-GFP (WT or the 1–100 fragment) and/or SFB-CENP-Q were subjected to pull-down with GST or GST-Scc1-SA2, followed by immunoblotting with the antibodies for GFP and the Flag-tag, and CBB staining. (I,J) HeLa cells were transfected with the indicated siRNAs. At 48 h post-transfection, cells were treated with MG132 for 6 h, then mitotic chromosome spreads were stained and counted in over 300 cells for each condition from three independent experiments, with statistics being analyzed for cells with separated chromatids using unpaired Student's *t*-test (I). NS no significance. Example images are shown (J). Scale bars, 10 μm.

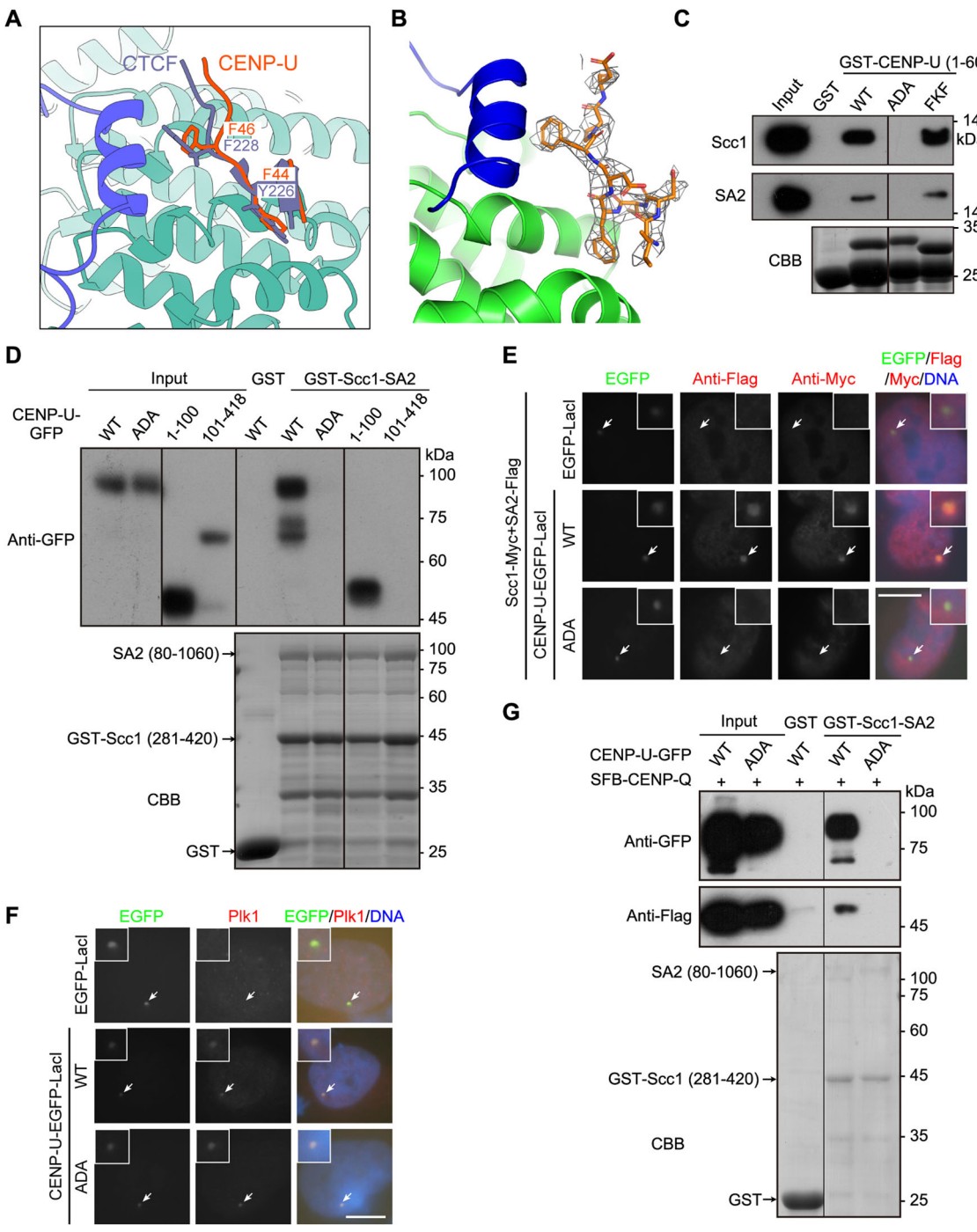

**Figure EV3. The FDF motif of CENP-U directly binds to the composite interface between Scc1 and SA2.**

(A) Structural superposition of Scc1-SA2 (purple and cyan) bound to CENP-U (orange) and CTCF (purple-blue). F44, F46 of CENP-U and Y226, F228 of CTCF are shown in stick. (B) Fo-Fc omit electron-density Fourier map contoured at 2.0 σ. Residues of CENP-U are shown in orange, and SA2 and Scc1 are in green and blue, respectively. (C) HeLa cell lysates were subjected to pull-down with GST or GST-CENP-U (1–60) in the forms of WT, ADA, and FKF, followed by immunoblotting with antibodies for Scc1 and SA2, and CBB staining. (D) Lysates prepared from HEK-293T cells transiently expressing CENP-U-GFP in the forms of WT, ADA, and the indicated fragments were subjected to pull-down with GST or GST-Scc1-SA2, followed by immunoblotting with the antibody for GFP, and CBB staining. (E) U2OS-LacO cells transiently expressing the indicated proteins were stained with antibodies for the Flag-tag, Myc-tag, and DAPI. Example images are shown. (F) U2OS-LacO cells transiently expressing the indicated proteins were stained with the antibody for Plk1, and DAPI. Example images are shown. (G) Lysates prepared from HEK-293T cells transiently expressing SFB-CENP-Q and CENP-U-GFP (WT or ADA) were subjected to pull-down with GST or GST-Scc1-SA2, followed by immunoblotting with antibodies for GFP and the Flag-tag, and CBB staining. Data information: The white arrows point to the LacO repeats (E, F). Scale bars, 10 μm (E, F). Irrelevant lanes were removed (C, D, G).

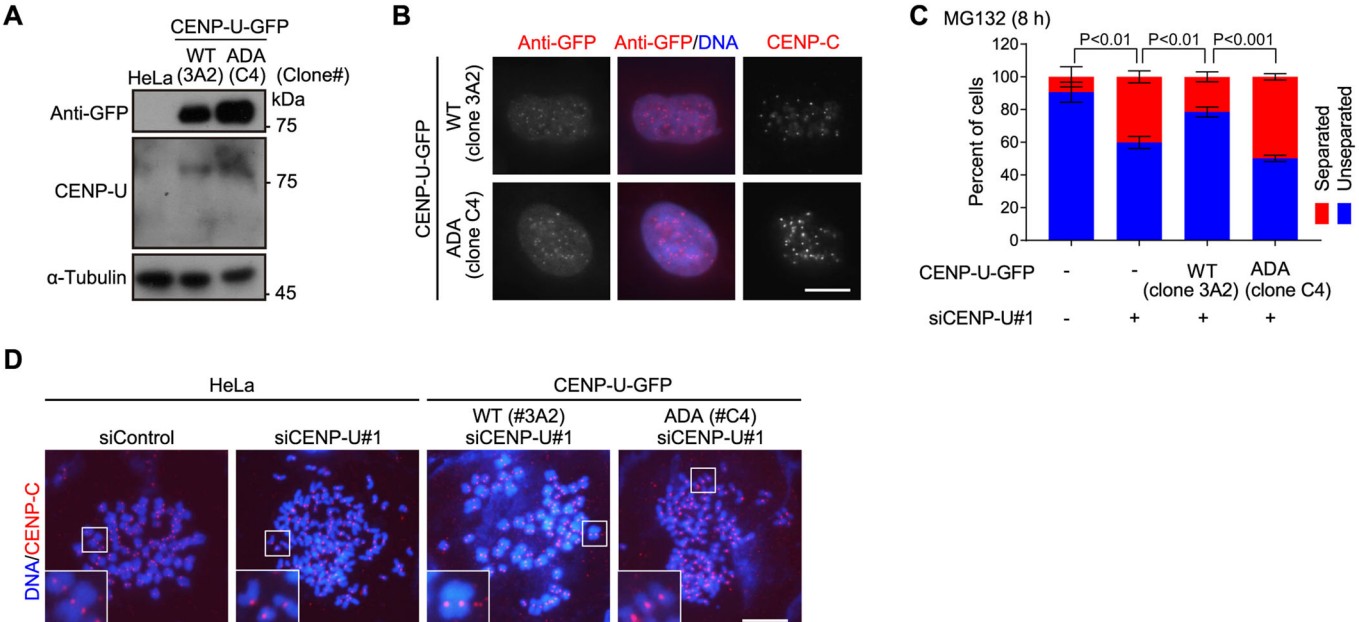

**Figure EV4.  The FDF motif is required for CENP-U to maintain metaphase sister-chromatid cohesion.**

(A) Asynchronous HeLa cells stably expressing siRNA-resistant CENP-U-GFP (WT or the ADA mutant) were subjected to immunoblotting with antibodies for GFP, CENP-U, and α-Tubulin. (B) The indicated stable cell lines were immunostained with antibodies for GFP, CENP-C, and DAPI. Example images are shown. (C, D) HeLa cells stably expressing the indicated proteins were transfected with control siRNA or CENP-U siRNA. At 48 h post-transfection, cells were treated with MG132 for 8 h, then mitotic chromosome spreads were stained with the CENP-C antibody and DAPI. The percentage of cells in which the majority of sister chromatids was separated or unseparated was determined in 300 cells for each condition from three independent experiments, with statistics being analyzed for cells with separated chromatids using unpaired Student's *t*-test. Means and SDs are shown (C). Example images are shown (D). Data information: Scale bars, 10 μm (B, D).

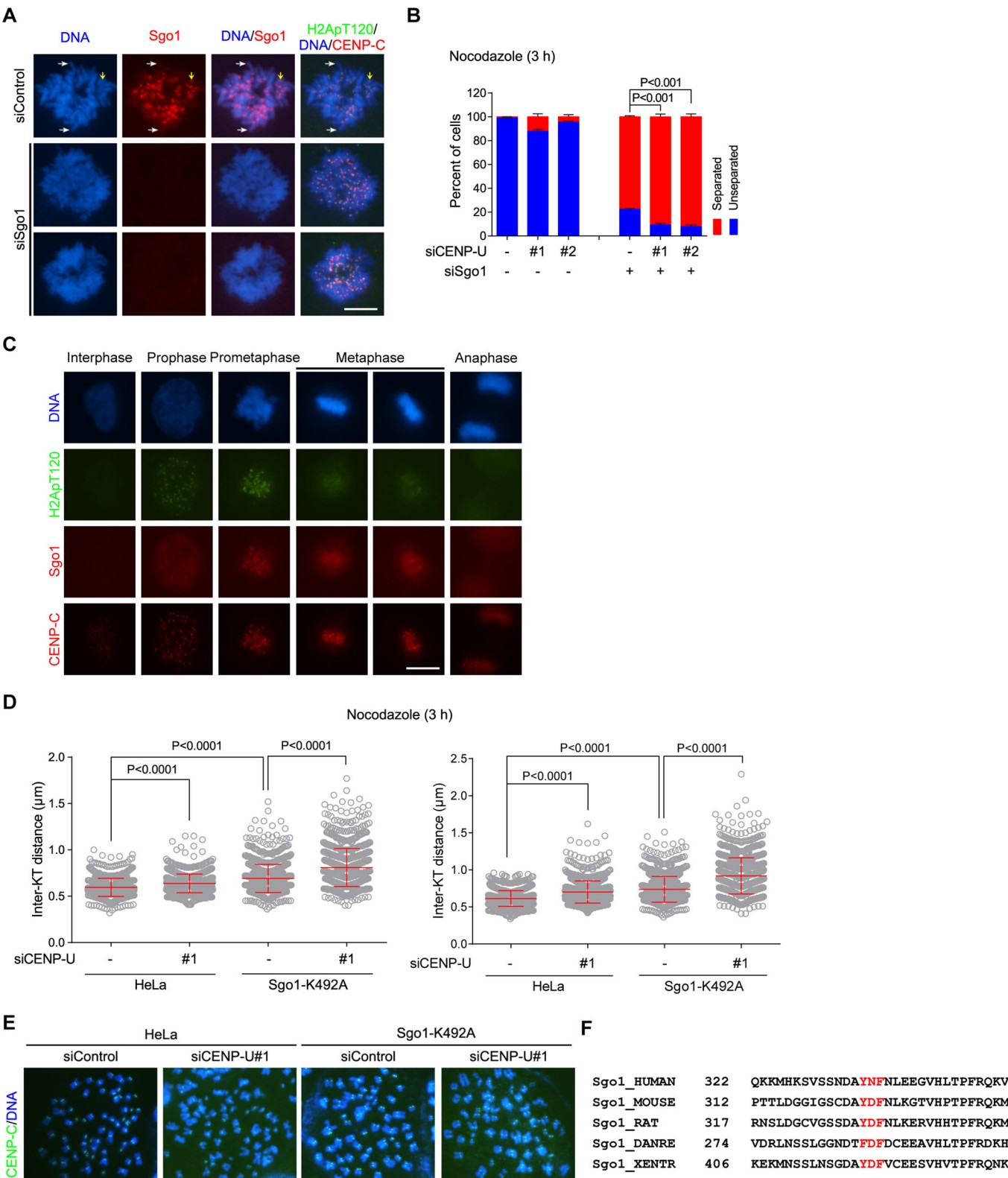

◀ **Figure EV5.   CENP-U and Sgo1 additively contribute to the strength of centromeric cohesion.**

(A) HeLa cells were transfected with control siRNA or Sgo1 siRNA. At 28 h post-transfection, cells were arrested in S-phase with thymidine treatment for 20 h, and then released into fresh medium. At 9 h post-release, cells were treated for 2 h with nocodazole, then mitotic cells were collected and then cytospun onto coverslips, fixed, and immunostained with antibodies for Sgo1, H2ApT120 and CENP-C, and DAPI. Example images are shown. White arrows point to Sgo1 distributed on chromosome arms. The yellow arrow points to Sgo1 enriched at mitotic centromeres. (B) HeLa cells were transfected with control siRNA, CENP-U siRNA, and/or Sgo1 siRNA. At 48h post-transfection, cells were treated with nocodazole for 3 h, then mitotic cells were collected to prepare chromosome spreads, and then stained with the CENP-C antibody and DAPI. The percentage of cells in which the majority of sister chromatids was separated or unseparated was determined in 300 cells for each condition from three independent experiments, with statistics being analyzed for cells with separated chromatids. (C) Asynchronous cells were fixed and immunostained with antibodies for Sgo1, H2ApT120, CENP-C, and DAPI. Example images for cells at the indicated stages of the cell cycle are shown. (D, E) Control HeLa cells and Sgo1-K492A mutant cells were transfected with control siRNA or CENP-U siRNA. At 48 h post-transfection, cells were subjected to nocodazole treatment for 3 h. Mitotic chromosome spreads were stained with the CENP-C antibody and DAPI. The inter-KT distance was measured on over 1000 chromosomes in 20 cells. Data from two individual experiments are shown (D). Example images are shown (E). Related to Fig. 8C. (F) Multiple sequence alignment for the Y/F-x-F motif-containing region of Sgo1 in the indicated vertebrates. Data information: Statistics were performed using unpaired Student's *t*-test (B, D). Means and SDs are shown (B, D). Scale bars, 10 μm (A, C, E).

