## [Peer Review File · The EMBO Journal]

A non-canonical role of the inner kinetochore in regulating sister-chromatid cohesion at centromeres

Fangwei Wang, Lu Yan, Xueying Yuan, Mingjie Liu, Qinfu Chen, Miao Zhang, Junfen Xu, Ling-Hui Zeng, Long Zhang, Jun Huang, Weiguo Lu, Xiaojing He, and Haiyan Yan

Corresponding author(s): Fangwei Wang (fwwang@zju.edu.cn), Xiaojing He (hexj@hust.edu.cn), Haiyan Yan (yanhy@hzcu.edu.cn)

Review Timeline:

Submission Date:	21st Sep 23
Editorial Decision:	26th Oct 23
Appeal:	22nd Nov 23
Editorial Decision:	22nd Dec 23
Revision Received:	23rd Jan 24
Editorial Decision:	21st Mar 24
Revision Received:	27th Mar 24
Accepted:	12th Apr 24

Editor: Hartmut Vodermaier

Transaction Report:

Prof. Fangwei Wang
Zhejiang University
Life Sciences Institute
866 Yuhangtang Rd
Nano Building Rm 577
Hangzhou, Zhejiang 310058
China

26th Oct 2023

Re: EMBOJ-2023-115677

The inner kinetochore binds Cohesin to protect centromeric sister chromatid cohesion

Dear Fangwei,

Thank you again for submitting your manuscript on cohesion regulation by CENP-U to The EMBO Journal. Three referees with expertise in structural, biochemical and cell biological studies of kinetochores and cohesins have now assessed it, and their reports are copied below for your information. I am afraid to say that in light of their comments, we had to conclude that we cannot offer publication in The EMBO Journal. The referees acknowledge that your principle finding of a CENP-U role in centromeric cohesion would be potentially interesting and helping to better understand how cohesin is differentially protected by different mechanisms. However, they also point out that the principle ability of CENP-U peptides to bind SCC1-SA2 interfaces has already been published (Li et al 2020), and that the structural data aimed at directly demonstrating, defining and altering this interaction are not sufficiently conclusive (according to the structural expert referee 1). The other key concern raised throughout the reports is the over-reliance on the use of protein fragments and overexpression of proteins that are normally parts of well-defined multimeric protein complexes, and the consequent uncertainty about their behavior in physiological context, when present in full-length and at endogenous levels. We feel that these combined issues have the potential to undermine the key conclusions of the study. Also, since the results of the extensive follow-up work required to clarify them is of uncertain outcome, I unfortunately do not see myself in the position to invite (and thus to some degree commit to) a revised version of this work for The EMBO Journal.

I am sorry that the reports do not allow me to be more positive on this occasion, but hope that you will nevertheless find the referees' comments and suggestions helpful when considering how to proceed further with this study. Thank you once more for having had the opportunity to consider this work for publication.

Yours sincerely,

Hartmut

Referee #1:

This manuscript reports the interaction between the inner kinetochore subunit CENP-U and the SA2-SCC1 components of the cohesin complex. The authors propose that this interaction protects sister chromatid cohesin because the CENP-U-binding site of SA2-SCC1 overlaps with the binding site of Wapl, a cohesin release factor. Specifically the authors suggest that the FDF (residues 44-46) motif of CENP-U interacts at the so-called CES-binding site of the composite interface of SA2-SCC1, that has been reported by the Panne and Rowland groups to be the binding site for CTCF and SGO1, and other proteins with an FxF motif.

The study reports cell biology data with protein crystallography and biochemical studies, mainly pull downs. The authors show that residues 1-200 of CENP-U pulls down SA2 and SCC1 from HeLa cell lysates and also recombinant fragments of SA2-SCC1 in vitro.

Concerning the protein crystallography, and in vitro characterization of CENP-U binding to SA2-SCC1, this reviewer has major

concerns.

1. The 2Fo-Fc simulated annealing omit map (missing from the submitted manuscript) requested by this reviewer does not show convincing density for a peptide bound to the CES site of SA2-SCC1. The building of the CENP-U peptide in this structure must have been based on the SA2-SCC1-CTCF and SA2-SCC1-SGO1 structures. Even the main-chain density for the CENP-U peptide from Phe44 to Phe46 is broken and fragmented, there is no convincing density for the Phe44 side chain. The electron density present at this site is fragmented and the same level as surrounding noise peaks. While electron density is missing for most of the peptide, other electron density peaks connected to that associated with the peptide is ignored. Thus, based on these crystallographic data, this reviewer is not convinced that CENP-U binds to the SA2-SCC1/cohesin complex. The absence of strong electron density at a significant contour level could be due to either weak or even no occupancy at the CES site, coupled to the low resolution of the structure. From this, the authors cannot claim they have determined the crystal structure of the SA2-SCC1-CENP-U peptide complex.
2. A further and related concern is that the authors provide no quantitative data on the affinity of the CENP-U peptide for SA2-SCC1. Thus the question of whether the lack of definitive electron density at the CES site being due to low occupancy cannot easily be addressed. The authors incubated SA2-SCC1 crystals with 500 μ M CENP-U peptide, but without knowing the K_d for the CENP-U peptide binding to SA2-SCC1, it is difficult to judge whether this concentration is in the correct range.
3. The authors need to determine quantitatively that the CENP-U peptide, and ideally the whole CENP-U subunit or the CENP-OPQUR sub-complex, binds to SA2-SCC1. The use of short peptides to infer binding between complexes is a starting point to testing the interactions between the larger complexes. It is possible that the CENP-U peptide that the authors claim binds to SA2-SCC1 is not accessible for such binding in the context of the fully assembled complexes.
4. The authors used AlphaFold to predict how SGO1 binds to SA2-SCC1. This is a surprising inclusion given that the crystal structure of the SA2-SCC1-SGO1 complex was recently published (Garcia-Nieto et al., 2023, ref 54). This latter paper also included an AlphaFold prediction of the SA2-SCC1-SGO1 complex.
5. All interaction data reported involves over-expression of either CENP-U or SA2-SCC1 in HeLa cells. This raises the question as to whether these proteins interact when expressed at endogenous levels.
6. The authors used siRNA to deplete CENP-U (Fig. 1). No control Western blot experiment to show that CENP-U protein levels were depleted is shown. The authors refer to an earlier paper (Chen et al., 2021, ref. 42) that does show CENP-U protein depletion with siRNA. This should be clearly stated.

Referee #2:

To ensure accurate chromosome segregation in cellular division, duplicated sister chromatids must remain tightly associated until all chromosomes are accurately aligned on the metaphase plate. To achieve this, the cohesin complex acts as a molecular glue, tethering sister chromatids together until anaphase onset. Once chromosome alignment is achieved, cohesive cohesin complexes are then cleaved by the protease Separase, allowing chromatid separation. During DNA replication, cohesive cohesin complexes are added along the length of chromatid arms, with the majority being stripped away in early mitosis by the protein WAPL in a process known as the prophase pathway. A small population of cohesin found at centromeric DNA is protected from WAPL removal by Shugoshin, allowing sister chromatids to remain tethered in mitosis. The precise localization of cohesin at centromeric DNA and the proteins it interacts with in this region remain poorly defined.

Here, the authors identify CENP-U, a component of the inner kinetochore, as a potential binding partner of cohesin. Similar to previous reports, the authors show that perturbation of CENP-U does not result in significant defects in chromosome segregation under normal mitotic division. However, under conditions of prolonged mitosis, driven by incubation in the proteasome inhibitor MG132, CENP-U knockdown cells display characteristic phenotypes of cohesin fatigue including misaligned chromosomes and prematurely separated sister chromatids (PSSC). Using pull down assays and GST tagged CENP-U constructs the authors show CENP-U can interact directly with two of the core cohesin subunits, RAD21 and STAG2. Further analysis using truncation constructs revealed residues 40-50 in CENP-U as significant for this interaction and the authors identified a mammalian conserved FDF motif in CENP-U that resembles the YxF motif found in cohesin interacting proteins. Importantly the authors show that point mutations in this domain are sufficient to prevent the CENP-U-cohesin interaction. The authors go on to purify a RAD21, STAG2, CENP-U complex, using fragments for each of the proteins. From these data, the authors predict that the binding of CENP-U to cohesin protects from premature removal of cohesin by WAPL. In support of this hypothesis, the authors show that the phenotypes observed in CENP-U depletion under prolonged mitosis can be rescued by depletion of WAPL.

Together, this work identifies an interaction between the inner kinetochore and cohesin that may help protect against cohesin fatigue in situations of prolonged mitosis. The identification of a specific binding partner of cohesin at centromeric DNA is of interest to both the cell division and genome organization fields. These findings will also have clear implications in understanding how cells maintain chromosome alignment under prolonged stress. If the authors are able to sufficiently address the comments below through appropriate revisions, I would support publication of this manuscript in EMBOJ.

1) Perhaps the most important experiment in this paper is the analysis of the CENP-U ADA mutant in Figure 4D and 4E. However, in looking at the quantification for this experiment, it is not particularly convincing that this mutation disrupts the role of

the CENP-U complex in helping protect cohesion during a prolonged arrest. In particular, the rescue for the wild type construct is modest, and thus the difference between the wild type control and this mutant is also quite small. Considering the previously described roles of this complex in binding to Plk1 or microtubules, it is also possible that these other interactions contribute to the chromosome alignment phenotype (Figure 4C). The authors should use caution with their conclusions here.

2) Due to the instability of purified full-length CENP-U the authors used GST fragments spanning the N terminus of CENP-U for their analysis. Additionally, in the crystallization experiments only a small fragment of CENP-U is purified along with fragments of RAD21 and STAG2. CENP-U functions as part of a larger complex comprised of CENP-O, CENP-P, CENP-Q, and CENP-R. This complex then assembles as part of the larger Constitutive Centromere associated network at centromeric DNA. Similarly, RAD21 and STAG2 also function as part of larger protein assemblies that include SMC1, SMC3. Although the authors provide evidence that the expressed CENP-U fragments can interact with fragments of RAD21 and STAG2, these experiments cannot recapitulate the behavior or orientation of these proteins when part of their larger interaction networks. Although I recognize that technical challenges limit the ability to test these full interactions, these caveats should be discussed and highlighted in both the results and discussion sections.

3) In Figure 1 and Supplementary Figure S1, the authors identify chromosome alignment defects when CENP-U is perturbed in the presence of MG132. The authors go on to state that "CENP-U is required to maintain chromosome alignment on the metaphase plate". This statement should be altered to indicate that this "during a prolonged mitosis" as CENP-U is not required for chromosome alignment in normal mitotic cells.

4) In Figure 1 and Supplementary Figure S1, the authors show that CENP-U knockdown cells have increased incidence of PSSC when incubated in MG132. From this data the authors then state that "CENP-U is necessary for the maintenance of sister chromatid cohesin at mitotic centromeres". This statement should be revised as this function is not shown in this manuscript, as there is no direct quantification of cohesin turnover or loss in CENP-U knockdown cells. Instead, the data shown indicate a phenotype of cohesin fatigue in situations of prolonged mitosis.

5) The authors show that, during a prolonged mitosis, CENP-U is required to protect against cohesin fatigue. To achieve prolonged mitosis, the authors rely on MG132, a proteasome inhibitor. As this will alter protein degradation globally, it is important to consider whether the observed behavior a consequence of aberrant protein turnover? For example, is the inability to properly turnover WAPL altered, resulting in an increase in WAPL activity? Or is this behavior observed in all situations of prolonged mitosis? The authors should test additional mitotic perturbations, for example incubation in nocodazole or KIF11-inhibited cells, and test if CENP-U depletion still results in PSSC.

6) The authors provide multiple experiments to support their hypothesis that CENP-U interacts directly with cohesin. These experiments are important to the manuscript as a whole, but in the main figures complicated the story. For example, I found the biochemical experiments in Figure 3 to be convincing particularly due to the mutant that eliminates the interaction providing a clear control. In contrast, the experiments in Figure 2 were much less convincing and are overwhelming and very hard to follow. To simplify the manuscript for the reader, I recommend moving many of supportive figures to the supplement or eliminating unnecessary and redundant experiments to focus on the critical pieces of data.

7) Throughout the manuscript and in all figures the authors need to make clear that the cells are incubated in the presence of the proteasome inhibitor MG132 and that the phenotypes are only observed in situations of prolonged mitosis - in both the figure itself, in the text, and in the legends

8) Table S1 was not included with manuscript

9) Figure 1C. The misaligned chromosomes are difficult to see. I recommend adjusting the images to be in greyscale rather than blue.

10) Figure 2G. No band is shown in the control GST-CENPU lane for STAG2. Why is this?

Referee #3:

This study reports the role of the inner kinetochore component CENP-U in regulation of centromeric cohesion through an interaction between the N-terminal region of CENP-U and an interface formed by SCC1-SA2 components of cohesin. This surface had been identified 10 years ago by the Yu lab as responsible for the competition of WAPL and SGO1 for binding cohesin and therefore critical for protection of a fraction of cohesin from the prophase pathway (Hara 2014 NSMB). More recently, the same surface was also identified by the Rowland-Panne labs as key for the interaction between cohesin and CTCF (Li 2020 Nature). In this last publication, the authors further proposed the existence of additional interactors for this surface using peptide arrays. CENP-U is one of them, consistent with findings described in the current manuscript. (By the way, this should be mentioned either in Introduction or in Discussion)

Here, SA2 and SCC1 are found in pull downs from human cell extracts using as bait a 100-aa fragment of CENP-U and the authors demonstrate that this interaction contributes to strengthen centromeric cohesion in mitosis. The manuscript is well written and identifies a new role for the CENP-OPQUR complex in regulation of cohesion. Overall, the biochemical and immunofluorescence data are solid and convincingly support the conclusions. My major criticism is that the relevance of this mechanism for mitotic cohesion is unclear since:

(1) most protein interaction data are obtained with overexpressed proteins or in vitro;

(2) proper comparison of the consequences of CENP-U knock down and Sgo1 knock down is missing (Sgo1 being currently considered the major mechanism ensuring mitotic cohesion through protection of centromeric cohesin from Wapl mediated release).

I have some suggestions to strengthen these aspects of the manuscript:

1. Most biochemical data in the paper come from incubations of a GST-bound CENP-U fragment and cell extracts. These data are in general solid and appropriately controlled, and provide important information about the physical interactions between the corresponding proteins. However, it would be important to detect some interactions with endogenous proteins using antibodies against CENP-U or even in the GFP-CENPU cell line.

Also, in Figure 2L, lysates prepared from HEK-293T cells transiently expressing Myc-SA2, Scc1-GFP or both, are subjected to pull-down with GST or GST-CENP-U (1-60), and only when both are present, they are able to bind to GST-CENPU beads. Since the extract has endogenous proteins, I assume that overexpression of both cohesin components is required to detect the interaction and that it happens in the absence of the rest of the complex. Is this correct? I would like to see both blots for SCC1 and SA2 in Figures 2E-H.

2. Does CENP-U bind cohesin in the context of a CENP-OPQUR complex? Does knock down of other component of the complex result in mitotic cohesion defects?

3. The authors assume that a Sgo1 mutant unable to interact with phosphoH2A (K492A) and therefore not properly targeted to centromeres, is equivalent to a loss of Sgo1. However, defects after Sgo1 knock down are likely much stronger than those observed after CENP-U knock down, which require prolonged metaphase arrest. The Yu lab showed that there is still some Sgo1 at centromeres, and quite a lot along the chromosome arms, in this mutant background. They also showed that ectopic expression of this mutant after Sgo1 knock down rescued mitotic cohesion to a large extent (Liu 2013 Curr Biol). The authors should include siSgo1 and double siSgo1 siCENP-U conditions in the experiments shown in Figure 8. Only in this way the relevance of a Sgo1-independent pathway can be assessed. Showing Sgo1 staining and, if possible, cohesin staining in all these conditions would provide important clues to understand regulation of mitotic cohesion and the contribution of CENP-U to this pathway.

4. The authors should include western blots showing the remaining levels of proteins after siRNA for all experiments with knock downs. In Figure 4, we should be able to compare endogenous levels and those of ectopically expressed CENP-U WT and mutant proteins.

*** As a service to authors, The EMBO Journal offers the possibility to directly transfer declined manuscripts to another EMBO Press title (EMBO Reports, EMBO Molecular Medicine, Molecular Systems Biology) or to the open access journal Life Science Alliance launched in partnership between EMBO Press, Rockefeller University Press and Cold Spring Harbor Laboratory Press. The full manuscript (including reviewer comments, where applicable and if chosen) will be automatically forwarded to the receiving journal, to allow for fast handling and a prompt decision on your manuscript. For more details of this service, and to transfer your manuscript to another EMBO title please follow this link:

Link Not Available

Thanks for letting me know the regretful outcome, and sorry for the delay in response. I appreciate the comments from reviewers, which are really helpful for improving the quality of our manuscript. Actually, all of the recommended experiments are doable. Importantly, we have done many experiments during the review/revision process of the manuscript, and have obtained lots of important new data which can address all the reviewers' comments. In addition, we have carefully reprocessed the structure data which unambiguously show that CENP-U binds to the Scc1-SA2 sub-complex in the crystal.

Here, I attach the point-to-point response file to appeal your decision on our manuscript. I believe that we can finish all the remaining experiments within the next two weeks or so. I will be very happy if revision of our manuscript is invited after you carefully read my response to reviewer comments. Many thanks in advance for your reply.

Reviewer comments

Referee #1:

This manuscript reports the interaction between the inner kinetochore subunit CENP-U and the SA2-SCC1 components of the cohesin complex. The authors propose that this interaction protects sister chromatid cohesin because the CENP-U-binding site of SA2-SCC1 overlaps with the binding site of Wapl, a cohesin release factor. Specifically the authors suggest that the FDF (residues 44-46) motif of CENP-U interacts at the so-called CES-binding site of the composite interface of SA2-SCC1, that has been reported by the Panne and Rowland groups to be the binding site for CTCF and SGO1, and other proteins with an FxF motif.

The study reports cell biology data with protein crystallography and biochemical studies, mainly pull downs. The authors show that residues 1-200 of CENP-U pulls down SA2 and SCC1 from HeLa cell lysates and also recombinant fragments of SA2-SCC1 in vitro.

Concerning the protein crystallography, and in vitro characterization of CENP-U binding to SA2-SCC1, this reviewer has major concerns.

1. The 2Fo-Fc simulated annealing omit map (missing from the submitted manuscript) requested by this reviewer does not show convincing density for a peptide bound to the CES site of SA2-SCC1. The building of the CENP-U peptide in this structure must have been based on the SA2-SCC1-CTCF and SA2-SCC1-SGO1 structures. Even the main-chain density for the CENP-U peptide from Phe44 to Phe46 is broken and fragmented, there is no convincing density for the Phe44 side chain. The electron density present at this site is fragmented and the same level as surrounding noise peaks. While electron density is missing for most of the peptide, other electron density peaks connected to that associated with the peptide is ignored. Thus, based on these crystallographic data, this reviewer is not convinced that CENP-U binds to the SA2-SCC1/cohesin complex. The absence of strong electron density at a significant contour level could be due to either weak or even no occupancy at the CES site, coupled to the low resolution of the structure. From this, the authors cannot claim they have determined the crystal structure of the SA2-SCC1-CENP-U peptide complex.

Response: Prompted by this reviewer's concern regarding whether CENP-U is present in the

crystal, we carefully reprocessed the data and re-refined the model. To avoid model bias, we did not include CENP-U in the model until the last round of the refinement. **The fo-fc map calculated prior to inclusion of CENP-U contoured at 2 sigma shows strong and continuous positive density at the location where CENP-U is bound (New Fig. 1).** The density covers residues 41-48 of CENP-U, with the side-chains of two aromatic residues (Phe-44 and Phe-46) clearly resolved. This map shows unambiguously that CENP-U is present in our crystal structure. We will put this figure in the supplemental data.

New Fig. 1. Fo-Fc omit electron-density Fourier map contoured at 2.0σ . Residues of CENP-U are shown in orange, and SA2 and Scc1 are in green and blue respectively.

As suggested by this reviewer, we calculated a new simulated-annealing composite omit map with the newly processed data and refined model in the attached file named “composite_omit_2.mtz”. This new composite omit map shows better quality than the previous one, and the density for CENP-U is continuous as expected.

Taken together, we believe that these new analyses support the presence of CENP-U in the structure. We would be happy to share the diffraction data if they are considered helpful for the reviewer to further evaluate this point.

2. A further and related concern is that the authors provide no quantitative data on the affinity of the CENP-U peptide for SA2-SCC1. Thus the question of whether the lack of definitive electron density at the CES site being due to low occupancy cannot easily be addressed. The authors incubated SA2-SCC1 crystals with 500 μ M CENP-U peptide, but without knowing the K_d for the CENP-U peptide binding to SA2-SCC1, it is difficult to judge whether this concentration is in the

correct range.

Response: Actually, quantification of peptide arrays performed by the Rowland-Panne labs have shown that the Scc1-SA2 complex binds to the CENP-U peptide (PIDVFDFPDNS) with Kd of $4.03 \pm 0.22 \mu\text{M}$ (Extended Data Table 2, Li et al., Nature, 2020, PMID: 31905366).

3. The authors need to determine quantitatively that the CENP-U peptide, and ideally the whole CENP-U subunit or the CENP-OPQUR sub-complex, binds to SA2-SCC1. The use of short peptides to infer binding between complexes is a starting point to testing the interactions between the larger complexes. It is possible that the CENP-U peptide that the authors claim binds to SA2-SCC1 is not accessible for such binding in the context of the fully assembled complexes.

Response: Actually, quantification of peptide arrays performed by the Rowland-Panne labs showed that the Scc1-SA2 complex binds to the CENP-U peptide (PIDVFDFPDNS) with Kd of $4.03 \pm 0.22 \mu\text{M}$ (Li et al., Nature, 2020, PMID: 31905366).

Regarding whether CENP-U is able to bind SA2-Scc1 in the context of the fully assembled complexes, the Musacchio lab previously reported that “CENP-O, -P, -Q, and -U were unstable when expressed individually in bacteria or insect cells and could not be recovered in soluble form (unpublished data)” (Pesenti M et al., Mol Cell, 2018, PMID: 30174292). In line with this, we found that bacterially expressed GST-fused human CENP-U protein in the forms of full-length (amino acid residues 1-418) and fragments encompassing residues 101-418 and 201-418 were unstable (Figure S2A-S2C). This technical obstacle prevented us from quantitatively determining the binding of whole CENP-U subunit to Scc1-SA2.

Regardless, we showed in the manuscript that:

1) Co-expression of Myc-fused SA2 with EGFP-LacI-fused Scc1 caused a strong recruitment of exogenously expressed CENP-U, which presumably interacted with endogenous proteins to form the CENP-OPQUR complex, to the transgenic LacO repeats in a euchromatic region of chromosome 1 in U2OS cells (Figure 2H-2I), in an FDF motif-dependent manner (Figure 3H-3I);

2) The GST-Scc1 (281-420)-SA2 (81-1060) sub-complex pulled down exogenously expressed CENP-U, which presumably interacted with endogenous CENP-OPQR to form a complex (Figure 2L), in an FDF motif-dependent manner (Figure 3G);

3) GST-CENP-U (1-100) pulled down exogenously expressed Scc1-GFP and Myc-SA2 (Figure 2J);

4) CENP-U-EGFP-LacI recruited Scc1-Myc and SA2-Flag to the LacO repeats in U2OS cells, in an FGF motif-dependent manner (Figure S3D-S3E).

It is known that CENP-Q and CENP-U form a sub-complex within the CENP-OPQUR complex in cells (Pesenti ME et al., Mol Cell, 2022, PMID: 35525244; Yatskevich S et al., Science, 2022, PMID: 35420891). **During the review/revision of the manuscript, we additionally found a line of new evidence indicating that the Scc1-SA2 sub-complex can bind CENP-U in complex with CENP-Q:**

1) GST-fused CENP-U (1-100) and CENP-U (101-418) bound to Scc1 and SFB-tagged

CENP-Q, respectively (New Fig. 2). This indicates that CENP-U uses its N-terminal region and C-terminal region to bind the Scc1-SA2 sub-complex and the CENP-Q subunit of the CENP-OPQUR complex, respectively. In other words, Scc1-SA2 binding to CENP-U may not interfere with CENP-Q binding to CENP-U, and vice versa.

New Fig. 2.

2) SFB-CENP-Q was pulled down by GST-Scc1-SA2 only when co-expressed with CENP-U-GFP, but not the Scc1-SA2-binding-deficient CENP-U-ADA-GFP mutant (New Fig. 3), or the CENP-Q-binding-deficient ENP-U (1-100)-GFP mutant (New Fig. 4). This indicates that CENP-Q indirectly associates with Scc1-SA2 through forming a complex with CENP-U;

New Fig. 3.

New Fig. 4.

4. The authors used AlphaFold to predict how SGO1 binds to SA2-SCC1. This is a surprising inclusion given that the crystal structure of the SA2-SCC1-SGO1 complex was recently published (Garcia-Nieto et al., 2023, ref 54). This latter paper also included an AlphaFold prediction of the SA2-SCC1-SGO1 complex.

Response: Actually, we cited the Garcia-Nieto et al., 2023 paper in the Results, and stating that “Thus, the Y-x-F motif of Sgo1 directly interacts with the binding interface between Sccl1 and SA2 in vitro and in cells, as recently reported (Garcia-Nieto et al., 2023). The similarity between the F/Y-x-F motif-dependent binding of CENP-U and Sgo1 to the Sccl1-SA2 sub-complex further supports our model that CENP-U protects centromeric cohesion independent of Sgo1”. Of course, we can also omit the AlphaFold prediction model from the manuscript and simply cite the Garcia-Nieto paper.

5. All interaction data reported involves over-expression of either CENP-U or SA2-SCC1 in HeLa cells. This raises the question as to whether these proteins interact when expressed at endogenous levels.

Response: Due to low abundance of endogenous CENP-U protein in cells, we cannot find an antibody for immunoblotting or immunoprecipitation of endogenous CENP-U, which prevented us from determining whether endogenous CENP-U interacts with endogenous Sccl1-SA2 in cells.

Importantly, during the review/revision of the manuscript, we found that endogenous Sccl1 and SA2 were co-immunoprecipitated with stably expressed CENP-U-GFP, but not the CENP-U-ADA-GFP mutant in which the FDF motif was mutated to ADA (New Fig. 5).

New Fig. 5.

6. The authors used siRNA to deplete CENP-U (Fig. 1). No control Western blot experiment to show that CENP-U protein levels were depleted is shown. The authors refer to an earlier paper (Chen et al., 2021, ref. 42) that does show CENP-U protein depletion with siRNA. This should be clearly stated.

Response: As mentioned above, due to low abundance of endogenous CENP-U protein in cells, we cannot find an antibody for immunoblotting of endogenous CENP-U. In our previous study (Chen et al., Cell Reports, 2021, PMID: 34551298), we showed siRNA-mediated CENP-U protein depletion by immunofluorescence staining. **Actually, we have carried out RT-PCR assays and immunofluorescence staining assays to confirm the knockdown of CENP-U at the mRNA level (New Fig. 6) and protein level (New Fig. 7), respectively. These data will be included in the revised manuscript.**

New Fig. 6.

New Fig. 7.

Referee #2:

To ensure accurate chromosome segregation in cellular division, duplicated sister chromatids must remain tightly associated until all chromosomes are accurately aligned on the metaphase plate. To achieve this, the cohesin complex acts as a molecular glue, tethering sister chromatids together until anaphase onset. Once chromosome alignment is achieved, cohesive cohesin complexes are then cleaved by the protease Separase, allowing chromatid separation. During DNA replication, cohesive cohesin complexes are added along the length of chromatid arms, with the majority being stripped away in early mitosis by the protein WAPL in a process known as the prophase pathway. A small population of cohesin found at centromeric DNA is protected from WAPL removal by Shugoshin, allowing sister chromatids to remain tethered in mitosis. The precise localization of cohesin at centromeric DNA and the proteins it interacts with in this region remain poorly defined.

Here, the authors identify CENP-U, a component of the inner kinetochore, as a potential binding partner of cohesin. Similar to previous reports, the authors show that perturbation of CENP-U does not result in significant defects in chromosome segregation under normal mitotic division. However, under conditions of prolonged mitosis, driven by incubation in the proteasome inhibitor MG132, CENP-U knockdown cells display characteristic phenotypes of cohesin fatigue including misaligned chromosomes and prematurely separated sister chromatids (PSSC). Using pull down assays and GST tagged CENP-U constructs the authors show CENP-U can interact directly with two of the core cohesin subunits, RAD21 and STAG2. Further analysis using truncation constructs revealed residues 40-50 in CENP-U as significant for this interaction and the authors identified a mammalian conserved FDF motif in CENP-U that resembles the YxF motif found in cohesin interacting proteins. Importantly the authors show that point mutations in this domain are sufficient to prevent the CENP-U-cohesin interaction. The authors go on to purify a RAD21, STAG2, CENP-U complex, using fragments for each of the proteins. From these data, the authors predict that the binding of CENP-U to cohesin protects from premature removal of cohesin by WAPL. In support of this hypothesis, the authors show that the phenotypes observed in CENP-U depletion under prolonged mitosis can be rescued by depletion of WAPL.

Together, this work identifies an interaction between the inner kinetochore and cohesin that may help protect against cohesin fatigue in situations of prolonged mitosis. The identification of a specific binding partner of cohesin at centromeric DNA is of interest to both the cell division and

genome organization fields. These findings will also will have clear implications in understanding how cells maintain chromosome alignment under prolonged stress. If the authors are able to sufficiently address the comments below through appropriate revisions, I would support publication of this manuscript in EMBOJ.

1) Perhaps the most important experiment in this paper is the analysis of the CENP-U ADA mutant in Figure 4D and 4E. However, in looking at the quantification for this experiment, it is not particularly convincing that this mutation disrupts the role of the CENP-U complex in helping protect cohesion during a prolonged arrest. In particular, the rescue for the wild type construct is modest, and thus the difference between the wild type control and this mutant is also quite small. Considering the previously described roles of this complex in binding to Plk1 or microtubules, it is also possible that these other interactions contribute to the chromosome alignment phenotype (Figure 4C). The authors should use caution with their conclusions here.

Response: As this reviewer pointed out, wild-type CENP-U-GFP did not fully rescue the sister chromatid cohesion defect caused by knockdown of endogenous CENP-U (Figures 4D and S4C), possibly due to overexpression of the exogenous protein. Importantly, statistical analysis of the percentage of mitotic cells with cohesion loss showed that, while wild-type CENP-U-GFP significantly restored sister chromatid cohesion, the CENP-U-ADA-GFP did not. The statistical analysis information will be included in the revised figures.

Importantly, during the review/revision of the manuscript, we succeeded in knocking in the DVFDF motif-to-AVAAA mutation into endogenous CENP-U of HeLa cells (New Fig. 8). Functional analysis showed that HeLa cells containing the CENP-U-AVAAA mutation are largely defective in maintaining sister chromatid cohesion during the metaphase arrest (New Fig. 9).

New Fig. 8.

New Fig. 9.

We and others previously reported that CENP-U interacts with Plk1 in a manner dependent on phosphorylation of T78 of CENP-U (Kang YH et al., Mol Cell, 2006, PMID: 17081991; Singh P et al., Mol Cell, 2021, PMID: 33248027; Chen et al., Cell Reports, 2021, PMID: 34551298). We showed in the manuscript that the CENP-U-T78A-GFP mutant significantly restored sister chromatid cohesion in the absence of endogenous CENP-U (Figure 4F-4H), indicating that CENP-U binding to Plk1 does not contribute to sister chromatid cohesion. **In addition, during the revision of the manuscript, we obtained new data indicating that the CENP-U-T78A-GFP mutant efficiently rescues the chromosome misalignment defect caused by knockdown of endogenous CENP-U (New Fig. 10).**

New Fig. 10.

2) Due to the instability of purified full-length CENP-U the authors used GST fragments spanning the N terminus of CENP-U for their analysis. Additionally, in the crystallization experiments only a small fragment of CENP-U is purified along with fragments of RAD21 and STAG2. CENP-U functions as part of a larger complex comprised of CENP-O, CENP-P, CENP-Q, and CENP-R. This complex then assembles as part of the larger Constitutive Centromere associated network at centromeric DNA. Similarly, RAD21 and STAG2 also function as part of larger protein assemblies that include SMC1, SMC3. Although the authors provide evidence that the expressed CENP-U fragments can interact with fragments of RAD21 and STAG2, these experiments cannot recapitulate the behavior or orientation of these proteins when part of their larger interaction networks. Although I recognize that technical challenges limit the ability to test these full interactions, these caveats should be discussed and highlighted in both the results and discussion sections.

Response: As mentioned above in the response to the first reviewer, during the review/revision of the manuscript, **we additionally found a line of new evidence indicating that the Scc1-SA2 sub-complex can bind CENP-U in complex with CENP-Q (New Figs. 2-4), and that endogenous Scc1 and SA2 were co-immunoprecipitated with stably expressed CENP-U-GFP, but not the CENP-U-ADA-GFP mutant in which the FDF motif was mutated to ADA (New Fig. 5).** We will also highlight and discuss in the Results and Discussion sections of the revised manuscript as recommended by this reviewer.

3) In Figure 1 and Supplementary Figure S1, the authors identify chromosome alignment defects when CENP-U is perturbed in the presence of MG132. The authors go on to state that "CENP-U is required to maintain chromosome alignment on the metaphase plate". This statement should be altered to indicate that this "during a prolonged mitosis" as CENP-U is not required for chromosome alignment in normal mitotic cells.

Response: We will alter the statement as suggested.

4) In Figure 1 and Supplementary Figure S1, the authors show that CENP-U knockdown cells have increased incidence of PSSC when incubated in MG132. From this data the authors then state that "CENP-U is necessary for the maintenance of sister chromatid cohesin at mitotic centromeres". This statement should be revised as this function is not shown in this manuscript, as there is no direct quantification of cohesin turnover or loss in CENP-U knockdown cells. Instead, the data shown indicate a phenotype of cohesin fatigue in situations of prolonged mitosis.

Response: We will revise the statement as suggested.

5) The authors show that, during a prolonged mitosis, CENP-U is required to protect against cohesin fatigue. To achieve prolonged mitosis, the authors rely on MG132, a proteasome inhibitor. As this will alter protein degradation globally, it is important to consider whether the observed behavior a consequence of aberrant protein turnover? For example, is the inability to properly

turnover WAPL altered, resulting in an increase in WAPL activity? Or is this behavior observed in all situations of prolonged mitosis? The authors should test additional mitotic perturbations, for example incubation in nocodazole or KIF11-inhibited cells, and test if CENP-U depletion still results in PSSC.

Response: In the mitosis research field, the proteasome inhibitor MG132 is widely used to induce metaphase arrest by inhibiting the anaphase-promoting complex/Cyclosome (APC/C)-dependent degradation of the Cdk1 activator Cyclin B and the Separase inhibitor Securin.

Regarding whether the observed behavior is a potential consequence of aberrant turnover of proteins such as Wapl, **we obtained new data indicating that Wapl protein levels are comparable in control siRNA and CENP-U siRNA transfected cells after treatment with MG132 for 8 h (New Fig. 11)**. This indicates that, at least during MG132 treatment for up to 8 h, CENP-U knockdown does not obviously affect Wapl protein level.

New Fig. 11.

Importantly, as suggested by this reviewer, we treated cells with Apcin, a small-molecule APC/C inhibitor that can arrest cells in metaphase by blocking the interaction between APC/C and its activator Cdc20 (Sackton K et al., Nature, 2014, PMID: 25156254). **We found that CENP-U knockdown caused an obvious loss of sister chromatid cohesion when cells were arrested in metaphase by Apcin treatment for 4.5 h (New Fig. 12)**.

New Fig. 12.

Besides, we showed in the manuscript that, upon arrest in a prometaphase-like stage by the spindle destabilizer nocodazole, the inter-kinetochore distance between sister kinetochores of chromosome spreads was around 12% apart in CENP-U depleted cells than in control cells (Figure 4D).

6) The authors provide multiple experiments to support their hypothesis that CENP-U interacts directly with cohesin. These experiments are important to the manuscript as a whole, but in the main figures complicated the story. For example, I found the biochemical experiments in Figure 3 to be convincing particularly due to the mutant that eliminates the interaction providing a clear control. In contrast, the experiments in Figure 2 were much less convincing and are overwhelming and very hard to follow. To simplify the manuscript for the reader, I recommend moving many of supportive figures to the supplement or eliminating unnecessary and redundant experiments to focus on the critical pieces of data.

Response: We will re-organize the Figure 2 and Figure S2 as recommended.

7) Throughout the manuscript and in all figures the authors need to make clear that the cells are incubated in the presence of the proteasome inhibitor MG132 and that the phenotypes are only observed in situations of prolonged mitosis - in both the figure itself, in the text, and in the legends

Response: We will include this information as recommended.

8) Table S1 was not included with manuscript

Response: Table S1 was actually included in the uploaded files, but probably not in the way that is obvious for the reviewer to download. We will include this table as recommended.

9) Figure 1C. The misaligned chromosomes are difficult to see. I recommend adjusting the images to be in greyscale rather than blue.

Response: We will do the adjustment as recommended.

10) Figure 2G. No band is shown in the control GST-CENPU lane for STAG2. Why is this?

Response: We have repeated this experiment including the GST control in the SA2 siRNA-transfected sample (New Fig. 13).

New Fig. 13.

Referee #3:

This study reports the role of the inner kinetochore component CENP-U in regulation of centromeric cohesion through an interaction between the N-terminal region of CENP-U and an interface formed by SCC1-SA2 components of cohesin. This surface had been identified 10 years ago by the Yu lab as responsible for the competition of WAPL and SGO1 for binding cohesin and therefore critical for protection of a fraction of cohesin from the prophase pathway (Hara 2014 NSMB). More recently, the same surface was also identified by the Rowland-Panne labs as key for the interaction between cohesin and CTCF (Li 2020 Nature). In this last publication, the authors further proposed the existence of additional interactors for this surface using peptide arrays. CENP-U is one of them, consistent with findings described in the current manuscript. (By the way, this should be mentioned either in Introduction or in Discussion)

Here, SA2 and SCC1 are found in pull downs from human cell extracts using as bait a 100-aa fragment of CENP-U and the authors demonstrate that this interaction contributes to strengthen centromeric cohesion in mitosis. The manuscript is well written and identifies a new role for the CENP-OPQUR complex in regulation of cohesion. Overall, the biochemical and immunofluorescence data are solid and convincingly support the conclusions. My major criticism is that the relevance of this mechanism for mitotic cohesion is unclear since:

- (1) most protein interaction data are obtained with overexpressed proteins or in vitro;
- (2) proper comparison of the consequences of CENP-U knock down and Sgo1 knock down is missing (Sgo1 being currently considered the major mechanism ensuring mitotic cohesion through protection of centromeric cohesin from Wapl mediated release).

I have some suggestions to strengthen these aspects of the manuscript:

1. Most biochemical data in the paper come from incubations of a GST-bound CENP-U fragment and cell extracts. These data are in general solid and appropriately controlled, and provide important information about the physical interactions between the corresponding proteins. However, it would be important to detect some interactions with endogenous proteins using antibodies against CENP-U or even in the GFP-CENPU cell line.

Response: As mentioned above, due to low abundance of endogenous CENP-U protein in

cells, we cannot find an antibody for immunoblotting or immunoprecipitation of endogenous CENP-U, which prevented us from determining whether endogenous CENP-U interacts with endogenous Scc1-SA2 in cells. **Importantly, during the review/revision of the manuscript, we found that endogenous Scc1 and SA2 were co-immunoprecipitated with stably expressed CENP-U-GFP, but not the CENP-U-ADA-GFP mutant in which the FDF motif was mutated to ADA (New Fig. 5).**

Also, in Figure 2L, lysates prepared from HEK-293T cells transiently expressing Myc-SA2, Scc1-GFP or both, are subjected to pull-down with GST or GST-CENP-U (1-60), and only when both are present, they are able to bind to GST-CENPU beads. Since the extract has endogenous proteins, I assume that overexpression of both cohesin components is required to detect the interaction and that it happens in the absence of the rest of the complex. Is this correct? I would like to see both blots for SCC1 and SA2 in Figures 2E-H.

Response: I assume that this comment points to Figure 2J, which shows that co-expression of Myc-SA2 strongly promotes Scc1-GFP binding to CENP-U in vitro, as we observed in cells (Figure 2H and 2I). In other words, these results indicate that overexpression of both Cohesin components (Scc1 and SA2) is required to detect the interaction between GST-CENP-U (1-100) and the Scc1-GFP/Myc-SA2 sub-complex. We reason that endogenous Scc1 and SA2 form a sub-complex with a 1:1 ratio, leaving free endogenous Scc1 and SA2 unavailable for binding exogenous Myc-SA2 and Scc1-GFP, respectively.

Regarding blots for SCC1 and SA2 in Figures 2E-H (actually Figures 2D-2G), we have repeated the experiments and blotted Scc1, SA2, SMC1 and SMC3 upon depletion of these individual components. The data are shown in New Fig. 14-16.

New Fig. 14.

New Fig. 15.

New Fig. 16.

2. Does CENP-U bind cohesin in the context of a CENP-OPQUR complex? Does knock down of other component of the complex result in mitotic cohesion defects?

Response: Regarding whether CENP-U binds Cohesin in the context of a CENP-OPQUR complex, as mentioned above, it is known that CENP-Q and CENP-U form a sub-complex within the CENP-OPQUR complex in cells (Pesenti ME et al., Mol Cell, 2022, PMID: 35525244; Yatskevich S et al., Science, 2022, PMID: 35420891). **During the review/revision of the manuscript, we additionally found a line of new evidence indicating that the Scc1-SA2 sub-complex can bind CENP-U in complex with CENP-Q:**

1) GST-fused CENP-U (1-100) and CENP-U (101-418) bound to Scc1 and SFB-tagged CENP-Q, respectively (**New Fig. 2**). This indicates that CENP-U uses its N-terminal region and C-terminal region to bind the Scc1-SA2 sub-complex and the CENP-Q subunit of the CENP-OPQUR complex, respectively.

2) SFB-CENP-Q was pulled down by GST-Scc1-SA2 only when co-expressed with CENP-U-GFP, but not the Scc1-SA2-binding-deficient CENP-U-ADA-GFP mutant (**New Fig. 3**), or the CENP-Q-binding-deficient ENP-U (1-100)-GFP mutant (**New Fig. 4**). This indicates that CENP-Q indirectly associates with Scc1-SA2 through forming a complex with CENP-

U;

Regarding whether knock down of other component of the complex result in mitotic cohesion defects, we obtained new data showing that:

1) Knockdown of CENP-P or CENP-Q caused a defect in maintaining metaphase sister chromatid cohesion as that observed in CENP-U depleted cells (New Fig. 17), which is in line the CENP-P and CENP-Q-dependent localization of CENP-U at kinetochores (Chen Q et al., Cell Reports, 2021, PMID: 34551298; Hori T et al., Mol Biol Cell, 2008, PMID: 18094054);

New Fig. 17.

2) Co-depletion of CENP-Q and CENP-U did not cause a further defect in maintaining metaphase sister chromatid cohesion, as compared to cells depleted of CENP-Q or CENP-U alone (New Fig. 18), implying that CENP-Q strengthens centromeric cohesion through interaction with CENP-U.

New Fig. 18.

3. The authors assume that a Sgo1 mutant unable to interact with phosphoH2A (K492A) and therefore not properly targeted to centromeres, is equivalent to a loss of Sgo1. However, defects after Sgo1 knock down are likely much stronger than those observed after CENP-U knock down, which require prolonged metaphase arrest. The Yu lab showed that there is still some Sgo1 at centromeres, and quite a lot along the chromosome arms, in this mutant background. They also

showed that ectopic expression of this mutant after Sgo1 knock down rescued mitotic cohesion to a large extent (Liu 2013 Curr Biol).

The authors should include siSgo1 and double siSgo1 siCENP-U conditions in the experiments shown in Figure 8. Only in this way the relevance of a Sgo1-independent pathway can be assessed. Showing Sgo1 staining and, if possible, cohesin staining in all these conditions would provide important clues to understand regulation of mitotic cohesion and the contribution of CENP-U to this pathway.

Response: Sgo1 is an important Cohesin protector that predominantly localizes to centromeres in mitosis. Previous studies from the Yu lab (and other labs including the Watanabe lab) showed that, during prometaphase, Sgo1 localizes to inner centromeres in a stepwise manner. First, through binding histone H2A which is phosphorylated at threonine 120 (H2ApT120) by Bub1, Sgo1 is recruited to two kinetochore-proximal outer centromere regions under the inner layer of kinetochores. In a second step, Sgo1 moves to inner centromeres, where it binds Cohesin in a manner that is strongly enhanced by Cyclin-dependent kinase 1 (Cdk1) phosphorylation of Sgo1. During metaphase, kinetochore tension generated by spindle pulling force triggers Sgo1 dephosphorylation and redistributes Sgo1 from the inner centromere to the outer centromere regions where it binds to histone H2ApT120. After bi-polar kinetochore-microtubule attachment, the H2ApT120 signal is largely reduced in metaphase due to the delocalization of Bub1 from kinetochore. Thus, Sgo1 plays a major role in protecting centromeric cohesion during prometaphase, which is line with the well-known observation that Sgo1 depletion by siRNA causes strong loss of sister chromatid cohesion during prometaphase. Our data indicate that CENP-U plays a role in maintaining centromeric cohesion during metaphase prior to anaphase onset, whereas Sgo1 is mainly required for the protection of centromeric cohesion during prometaphase.

We did the experiments as suggested by this reviewer, and found that double knockdown of Sgo1 and CENP-U by siRNA causes additive loss of sister chromatid cohesion in nocodazole-arrested mitotic HeLa cells (New Fig. 19).

New Fig. 19.

In addition, we found that CENP-U knockdown by siRNA causes a further increase in the inter-kinetochore distance in nocodazole-arrested cells expressing the Sgo1-K492A mutant which is defective in binding H2ApT120 (New Fig. 20). This indicates that CENP-U is still required to maintain centromeric cohesion in cells lacking centromeric Sgo1.

New Fig. 20.

4. The authors should include western blots showing the remaining levels of proteins after siRNA for all experiments with knock downs. In Figure 4, we should be able to compare endogenous levels and those of ectopically expressed CENP-U WT and mutant proteins.

Response: As mentioned above, due to low abundance of endogenous CENP-U protein in cells, we cannot find an antibody for immunoblotting of endogenous CENP-U. **Actually, we have carried out RT-PCR assays and immunofluorescence staining assays to confirm the knockdown of CENP-U at the mRNA level (New Fig. 6) and protein level (New Fig. 7), respectively.** These data will be included in the revised manuscript. As suggested, we will repeat Western blotting of Figure 4A and Figure S4A using the anti-CENP-U antibody.

Dear Fangwei,

Once more, apologies for the delay in getting back to you with editorial feedback on the detailed letter you had sent in response to the points the referees had raised on your recent EMBO Journal submission. As you hopefully understand, we have to give precedence to newly submitted manuscripts that are yet awaiting a first decision, over those that have already received a decision. Moreover, given the technical nature of many of the issues raised in this case, I wanted to once more hear the opinions of referees 1 and 3, to see if they would find your responses overall satisfying or not. Both of them have now taken the extra time and provided feedback, which I am copying below for your information.

As you will see, referee 1 feels that his/her points would now be largely addressed, especially from the structural side, and only a few more textual modifications would be required.

Referee 3 is now also more convinced about the principle findings, but still considers the data to some degree overinterpreted, especially in the absence of more compelling insights into the relative importance of the CENP-U mechanism over Sgo1. A key point would be repeating and quantifying the data in New Fig. 19, and establishing their significance in more than one siRNA condition! Other major points would be Sgo1 staining in chromosome spreads and in mitotic cells, and showing input controls in co-IPs, etc. - only editing the endogenous CENP-U locus to add a tag would probably exceed the scope of revision here.

In conclusion, I would in light of your responses and the referees' feedback now be open to a formal resubmission of a revised manuscript - as long as you should also incorporate referee 1's remaining minor points and, importantly, strengthen the data mentioned by referee 3 below. When preparing such a revised versions, please also pay close attention to the revision guidelines detailed in the Guide to Authors on our website, as this would greatly facilitate the editorial procession at revision stage. Please do not hesitate to contact me in case you should have any questions regarding this decision.

With kind regards,
Hartmut

Hartmut Vodermaier, PhD
Senior Editor | The EMBO Journal
h.vodermaier@embojournal.org

REFEREE 1

They have responded to most concerns I raised:

Point 1

The maps are significantly improved. From inspection of the composite omit map the authors provided, the fitting of Leu42 to Pro47 looks correct.

Point 2

This reference should be cited.

Point 3

The authors need to determine quantitatively that the CENP-U peptide, and ideally the whole CENP-U subunit or the CENP-OPQUR sub-complex, binds to SA2-SCC1.

>Author Response: Actually, quantification of peptide arrays performed by the Rowland-Panne labs showed that the Scc1-SA2 complex binds to the CENP-U peptide (PIDVDFDPNS) with K_d of $4.03 \pm 0.22 \mu\text{M}$ (Li et al., Nature, 2020, PMID: 31905366)

This is an inadequate response.

>In line with this, we found that bacterially expressed GST-fused human CENP-U protein in the forms of full-length (amino acid residues 1-418) and fragments encompassing residues 101-418 and 201-418 were unstable (Figure S2A-S2C). This technical obstacle prevented us from quantitatively determining the binding of whole CENP-U subunit to Scc1-SA2.

The OPQUR complex can be expressed and purified - see:

(Pesenti ME et al., Mol Cell, 2022, PMID: 35525244; Yatskevich S et al., Science, 2022, PMID: 35420891).

However the authors do show data that would indicate CENP-U in the context of CENP-OQUR would bind Scc1-SA2.

Point 4

>Of course, we can also omit the AlphaFold prediction model from the manuscript and simply cite the Garcia-Nieto paper.

Yes that would be more appropriate.

Point 5

Ok

Point 6

Ok

REFEREE 3

I have had a look at the response by the authors. Some of my queries have been addressed and I think overall the manuscript has improved with the new data.

My feeling is that the interaction is real and probably meaningful to prevent cohesion fatigue but the way the manuscript is written places the role of CENP-U at the level of Sgo1 and it is clearly not right. Defects with CENP-U KD only appear after mitotic arrest. Data in new Figure 19 show a small effect of CENP-U KD over Sgo1 KD. This is probably one single replica, and it is unclear to me if the difference shown is biologically relevant. The difference between the two siRNA against CENP-U is larger than the increase they provide over Sgo1 KD. (By the way, I do not understand why this experiment is not done as the rest in the paper, using MG132 instead of nocodazole). In any case, Sgo1 staining (and if possible cohesin staining although I know this is complicated) in spreads and in mitotic cells would shed some light on the mechanisms to support what they now write:

"Our data indicate that CENP-U plays a role in maintaining centromeric cohesion during metaphase prior to anaphase onset, whereas Sgo1 is mainly required for the protection of centromeric cohesion during prometaphase"

Why does the cell need two different proteins using the same SA2/Scc1 surface to compete out Wapl? Is this because during prolonged metaphase arrest the amount of Sgo1 at centromeres decreases significantly?

The data in new Figure 5 are promising but they should be improved by showing input (what %?) next to IP fractions, so we can have an idea of the efficiency, and also adding blots for SMC1 or SMC3 to support an interaction with the whole cohesin complex, and with some other component of the CENP-OPQUR complex.

Also, I understand that the authors do not have an antibody that allows them to work with endogenous proteins, but maybe they could edit the endogenous CENP-U locus to add a tag.

Point-by-point response to the reviewers' comments on the originally submitted manuscript (#EMBOJ-2023-115677)

We deeply appreciate the reviewers for careful reading of our manuscript and thoughtful suggestions. Obviously, the reviewers made excellent comments which we have all addressed now. We think the novelty and rigorousness of the manuscript are further improved as a result. The new data and new figures are highlighted in dark red, and the new/revised statements are underlined as shown below.

Please find below the detailed point-by-point response to the reviewers' comments. We very much hope that you now find the manuscript suitable for publication in The EMBO Journal.

Reviewer comments

Referee #1:

This manuscript reports the interaction between the inner kinetochore subunit CENP-U and the SA2-SCC1 components of the cohesin complex. The authors propose that this interaction protects sister chromatid cohesin because the CENP-U-binding site of SA2-SCC1 overlaps with the binding site of Wapl, a cohesin release factor. Specifically the authors suggest that the FDF (residues 44-46) motif of CENP-U interacts at the so-called CES-binding site of the composite interface of SA2-SCC1, that has been reported by the Panne and Rowland groups to be the binding site for CTCF and SGO1, and other proteins with an FxF motif.

The study reports cell biology data with protein crystallography and biochemical studies, mainly pull downs. The authors show that residues 1-200 of CENP-U pulls down SA2 and SCC1 from HeLa cell lysates and also recombinant fragments of SA2-SCC1 in vitro.

Concerning the protein crystallography, and in vitro characterization of CENP-U binding to SA2-SCC1, this reviewer has major concerns.

1. The 2Fo-Fc simulated annealing omit map (missing from the submitted manuscript) requested by this reviewer does not show convincing density for a peptide bound to the CES site of SA2-SCC1. The building of the CENP-U peptide in this structure must have been based on the SA2-SCC1-CTCF and SA2-SCC1-SGO1 structures. Even the main-chain density for the CENP-U peptide from Phe44 to Phe46 is broken and fragmented, there is no convincing density for the Phe44 side chain. The electron density present at this site is fragmented and the same level as surrounding noise peaks. While electron density is missing for most of the peptide, other electron density peaks connected to that associated with the peptide is ignored. Thus, based on these crystallographic data, this reviewer is not convinced that CENP-U binds to the SA2-SCC1/cohesin complex. The absence of strong electron density at a significant contour level could be due to either

weak or even no occupancy at the CES site, coupled to the low resolution of the structure. From this, the authors cannot claim they have determined the crystal structure of the SA2-SCC1-CENP-U peptide complex.

Response: Prompted by this reviewer's concern regarding whether CENP-U is present in the crystal, we carefully reprocessed the data and re-refined the model. To avoid model bias, we did not include CENP-U in the model until the last round of the refinement. The fo-fc map calculated prior to inclusion of CENP-U contoured at 2 sigma shows strong and continuous positive density at the location where CENP-U is bound (see Fig EV3B; new data). The density covers residues 41-48 of CENP-U, with the side-chains of two aromatic residues (Phe-44 and Phe-46) clearly resolved. This map shows unambiguously that CENP-U is present in our crystal structure.

As suggested by this reviewer, we calculated a new simulated-annealing composite omit map with the newly processed data and refined model. This new composite omit map shows better quality than the previous one, and the density for CENP-U is continuous as expected.

Taken together, we believe that these new analyses support the presence of CENP-U in the structure. We would be happy to share the diffraction data if they are considered helpful for the reviewer to further evaluate this point.

2. A further and related concern is that the authors provide no quantitative data on the affinity of the CENP-U peptide for SA2-SCC1. Thus the question of whether the lack of definitive electron density at the CES site being due to low occupancy cannot easily be addressed. The authors incubated SA2-SCC1 crystals with 500 μ M CENP-U peptide, but without knowing the K_d for the CENP-U peptide binding to SA2-SCC1, it is difficult to judge whether this concentration is in the correct range.

Response: As mentioned above, the fo-fc map shows unambiguously that CENP-U is present in our crystal structure of the Scc1 (281-420)-SA2 (80-1060) complex. In line with this, the Rowland-Panne labs used peptide arrays to show that the Scc1-SA2 complex binds to the CENP-U peptide (PIDVDFDFDNS) with K_d of $4.03 \pm 0.22 \mu$ M (Extended Data Table 2, Li et al., Nature, 2020, PMID: 31905366). This reference is now cited five times in the "Results" section and once in the "Discussion" section. As an example, we clearly stated in the "Results" section that "Thus, the DVFDF motif enables chicken CENP-U to bind the Scc1-SA2 sub-complex. This is in line with a previous study which used peptide arrays to show that a CENP-U peptide containing the DVFDF motif bound to Scc1-SA2 in vitro with a K_d of around 4.0μ M (Li et al., 2020)."

3. The authors need to determine quantitatively that the CENP-U peptide, and ideally the whole CENP-U subunit or the CENP-OPQUR sub-complex, binds to SA2-SCC1. The use of short peptides to infer binding between complexes is a starting point to testing the interactions between the larger complexes. It is possible that the CENP-U peptide that the authors claim binds to SA2-SCC1 is not accessible for such binding in the context of

the fully assembled complexes.

Response: These are good comments in principle. Regarding CENP-U binding to SA2-Scc1, the Rowland-Panne labs showed that the Scc1-SA2 complex binds to the CENP-U peptide (PIDVDFDPNS) with K_d of $4.03 \pm 0.22 \mu\text{M}$ (Li et al., Nature, 2020, PMID: 31905366).

Regarding whether CENP-U is able to bind SA2-Scc1 in the context of the fully assembled complexes, the Andrea Musacchio lab previously reported that “CENP-O, -P, -Q, and -U were unstable when expressed individually in bacteria or insect cells and could not be recovered in soluble form (unpublished data)” (Pesenti M et al., Mol Cell, 2018, PMID: 30174292). In line with this, we found that the bacterially expressed GST-fused human CENP-U protein in the forms of full-length (amino acid residues 1-418) and fragments encompassing residues 101-418 and 201-418 were unstable (Fig EV2A-C). Though the Andrea Musacchio lab and the David Barford lab later succeeded in expressing and purifying the CENP-OPQR complex (Pesenti M et al., Mol Cell, 2022, PMID: 35525244; Yatskevich S et al., Science, 2022, PMID: 35420891), technically we were not able to express the CENP-OPQR complex which prevented us from quantitatively determining the binding of whole CENP-U subunit to Scc1-SA2.

In the revised manuscript, we showed that stably expressed CENP-U-GFP, but not the CENP-U-ADA-GFP mutant, co-immunoprecipitated endogenous Scc1, SA2, SMC1 and SMC3 (Fig 4B; new data). This strongly implies the interaction between the CENP-OPQR complex and the Cohesin complex in cells.

Since CENP-Q and CENP-U form a sub-complex within the CENP-OPQR complex in cells, we provided a line of strong evidence in the revised manuscript indicating that the CENP-U and CENP-Q sub-complex binds to the Scc1-SA2 sub-complex:

1) Pull-down assays showed that GST-CENP-U (1-100) bound to Scc1 but not SFB-CENP-Q in HEK-293T cell lysates. In sharp contrast, GST-CENP-U (101-418) pulled down SFB-CENP-Q but not Scc1. Thus, CENP-U uses N-terminus and C-terminal region to bind the Scc1-SA2 sub-complex and CENP-Q, respectively. These new data are now shown in Fig EV2G.

2) Moreover, SFB-CENP-Q did not bind to the GST-Scc1-SA2 sub-complex when expressed alone. Interestingly, co-expression with CENP-U-GFP, but not CENP-U (1-100)-GFP, enabled the interaction. We confirmed that both CENP-U-GFP and CENP-U (1-100)-GFP were efficiently pulled down by GST-Scc1-SA2. These results indicate that CENP-Q indirectly binds Scc1-SA2 through interacting with the C-terminal region of CENP-U. These new data are now shown in Fig EV2H.

3) In line with the CENP-Q-dependent localization of CENP-U at kinetochores (Chen et al., 2021; Hori et al, 2008), knockdown of CENP-Q in CENP-U-depleted cells did not cause a further defect in maintaining metaphase sister-chromatid cohesion, implying that CENP-Q indirectly strengthens sister-chromatid cohesion through binding CENP-U that interacts with Scc1-SA2. These new data are now shown in Fig EV2I and J.

4) SFB-CENP-Q was pulled down by GST-Scc1-SA2 when co-expressed with CENP-

U-GFP, but not the CENP-U-ADA-GFP mutant, further conforming the indirect association of CENP-Q with the Scc1-SA2 sub-complex. **These new data are now shown in Fig EV3G.**

5) Co-expression of Myc-SA2 with EGFP-LacI-Scc1 caused a strong recruitment of exogenously expressed SFB-tagged CENP-U, which presumably interacted with endogenous proteins to form the CENP-OPQUR complex, to the transgenic LacO repeats in a euchromatic region of chromosome 1 in U2OS cells (Fig 2F and G), in an FDF motif-dependent manner (Fig 3H and I);

6) The GST-Scc1 (281-420)-SA2 (80-1060) sub-complex pulled down exogenously expressed SFB-tagged CENP-U, which presumably interacted with endogenous proteins to form the CENP-OPQUR (Fig 2J), in an FDF motif-dependent manner (Fig 3G);

7) GST-CENP-U (1-60) pulled down exogenously expressed Scc1-GFP and Myc-SA2 (Fig 2H and I);

8) CENP-U-EGFP-LacI recruited Scc1-Myc and SA2-Flag to the LacO repeats in U2OS cells, in an FGF motif-dependent manner (Fig EV3E).

4. The authors used AlphaFold to predict how SGO1 binds to SA2-SCC1. This is a surprising inclusion given that the crystal structure of the SA2-SCC1-SGO1 complex was recently published (Garcia-Nieto et al., 2023, ref 54). This latter paper also included an AlphaFold prediction of the SA2-SCC1-SGO1 complex.

Response: We cited the Garcia-Nieto et al., 2023 paper in the Results section, and stating that “Thus, the YNF motif mediates Sgo1 interaction with the binding interface between Scc1 and SA2 in vitro and in cells, as recently reported (Garcia-Nieto et al, 2023)”. Following this reviewer’s suggestion, we omitted the AlphaFold prediction model from the manuscript and simply cite the Garcia-Nieto et al. paper.

5. All interaction data reported involves over-expression of either CENP-U or SA2-SCC1 in HeLa cells. This raises the question as to whether these proteins interact when expressed at endogenous levels.

Response: CENP-U, which is exclusively localized at the inner kinetochore, is a low abundant protein in cells. We cannot find an antibody for immunoblotting or immunoprecipitation of endogenous CENP-U, which prevented us from determining whether endogenous CENP-U interacts with endogenous Scc1-SA2 in cells. Importantly, we stably expressed CENP-U-GFP (wild-type/WT and the ADA mutant) in HeLa cells, and found that CENP-U-GFP, but not the CENP-U-ADA-GFP mutant, co-immunoprecipitated endogenous Scc1, SA2, SMC1 and SMC3 (Fig 4B; **new data**). This strongly implies the interaction between the CENP-OPQUR complex and the Cohesin complex in cells.

6. The authors used siRNA to deplete CENP-U (Fig. 1). No control Western blot experiment to show that CENP-U protein levels were depleted is shown. The authors

refer to an earlier paper (Chen et al., 2021, ref. 42) that does show CENP-U protein depletion with siRNA. This should be clearly stated.

Response: As mentioned above, due to low abundance of endogenous CENP-U protein in cells, we cannot find an antibody for immunoblotting of endogenous CENP-U. In our previous study (Chen et al., Cell Reports, 2021, PMID: 34551298), we showed siRNA-mediated knockdown of CENP-U protein by immunofluorescence staining. Actually, we have carried out quantitative RT-PCR assays and immunofluorescence staining assays to confirm the knockdown of CENP-U at the mRNA level and protein level, respectively. These new data are now shown in Fig EV1D-F.

Referee #2:

To ensure accurate chromosome segregation in cellular division, duplicated sister chromatids must remain tightly associated until all chromosomes are accurately aligned on the metaphase plate. To achieve this, the cohesin complex acts as a molecular glue, tethering sister chromatids together until anaphase onset. Once chromosome alignment is achieved, cohesive cohesin complexes are then cleaved by the protease Separase, allowing chromatid separation. During DNA replication, cohesive cohesin complexes are added along the length of chromatid arms, with the majority being stripped away in early mitosis by the protein WAPL in a process known as the prophase pathway. A small population of cohesin found at centromeric DNA is protected from WAPL removal by Shugoshin, allowing sister chromatids to remain tethered in mitosis. The precise localization of cohesin at centromeric DNA and the proteins it interacts with in this region remain poorly defined.

Here, the authors identify CENP-U, a component of the inner kinetochore, as a potential binding partner of cohesin. Similar to previous reports, the authors show that perturbation of CENP-U does not result in significant defects in chromosome segregation under normal mitotic division. However, under conditions of prolonged mitosis, driven by incubation in the proteasome inhibitor MG132, CENP-U knockdown cells display characteristic phenotypes of cohesin fatigue including misaligned chromosomes and prematurely separated sister chromatids (PSSC). Using pull down assays and GST tagged CENP-U constructs the authors show CENP-U can interact directly with two of the core cohesin subunits, RAD21 and STAG2. Further analysis using truncation constructs revealed residues 40-50 in CENP-U as significant for this interaction and the authors identified a mammalian conserved FDF motif in CENP-U that resembles the YxF motif found in cohesin interacting proteins. Importantly the authors show that point mutations in this domain are sufficient to prevent the CENP-U-cohesin interaction. The authors go on to purify a RAD21, STAG2, CENP-U complex, using fragments for each of the proteins. From these data, the authors predict that the binding of CENP-U to cohesin protects from premature removal of cohesin by WAPL. In support of this hypothesis, the authors show that the phenotypes observed in CENP-U depletion under prolonged mitosis can be rescued by depletion of WAPL.

Together, this work identifies an interaction between the inner kinetochore and cohesin that may help protect against cohesin fatigue in situations of prolonged mitosis. The identification of a specific binding partner of cohesin at centromeric DNA is of interest to both the cell division and genome organization fields. These findings will also have clear implications in understanding how cells maintain chromosome alignment under prolonged stress. If the authors are able to sufficiently address the comments below through appropriate revisions, I would support publication of this manuscript in EMBOJ.

1) Perhaps the most important experiment in this paper is the analysis of the CENP-U ADA mutant in Figure 4D and 4E. However, in looking at the quantification for this experiment, it is not particularly convincing that this mutation disrupts the role of the CENP-U complex in helping protect cohesion during a prolonged arrest. In particular, the rescue for the wild type construct is modest, and thus the difference between the wild type control and this mutant is also quite small. Considering the previously described roles of this complex in binding to Plk1 or microtubules, it is also possible that these other interactions contribute to the chromosome alignment phenotype (Figure 4C). The authors should use caution with their conclusions here.

Response: As suggested, we performed statistics analysis which confirmed that the defect in maintaining metaphase sister-chromatid cohesion caused by depletion of endogenous CENP-U was partly but significantly ($p=0.0025$) rescued by exogenously expressed siRNA-resistant CENP-U-GFP, whereas the Scc1-SA2-binding-deficient mutant CENP-U-ADA-GFP did not show any rescue effect (Fig 4E and F). Similar results were observed in other independent stable clones expressing CENP-U-GFP (WT and the ADA mutant) (Fig EV4A-D).

Importantly, we further used CRISPR/Cas9-mediated homology-directed repair to mutate the DVDF motif of endogenous CENP-U to AVAAA in HeLa cells (Fig 4G; new data). Inspection of chromosome spreads prepared from MG132-induced metaphase cells demonstrated that cells with the CENP-U-AVAAA mutation were significantly less effective than control HeLa cells in maintaining sister-chromatid cohesion (Fig 4H and I; new data). Moreover, upon brief treatment with nocodazole, the inter-KT distance was around 10.5% further apart in CENP-U-AVAAA mutant cells than in control cells, which is indicative of weakened centromeric cohesion (Fig 4J and K; new data). These results indicate that the FDF motif of CENP-U is important for strengthening centromeric cohesion and maintaining metaphase sister-chromatid cohesion.

In addition, we showed that CENP-U-EGFP-LacI, but not the CENP-U-ADA-EGFP-LacI mutant, recruited co-expressed Scc1-Myc and SA2-Flag to the Lac operator repeats on chromosome 1 of U2OS cells (Fig EV3E), confirming the FDF motif-dependent interaction of CENP-U with Scc1-SA2. In contrast, both CENP-U-EGFP-LacI and CENP-U-ADA-EGFP-LacI efficiently recruited Plk1 (Fig EV3F), indicating that mutating the FDF motif does not affect CENP-U interaction with Plk1.

We then examined whether the CENP-U-bound Plk1 is involved in regulating sister-chromatid cohesion. We established a HeLa-derived cell line stably expressing the CENP-U-T78A-GFP mutant (Fig 4L), in which threonine-78 was mutated to alanine to prevent Plk1 binding. Examination of MG132-arrested metaphase chromosome spreads

showed that the CENP-U-T78A-GFP mutant was able to rescue the cohesion defect caused by knockdown of endogenous CENP-U (Fig 4M and N), indicating that the interaction with Plk1 is dispensable for CENP-U to maintain metaphase sister-chromatid cohesion. Moreover, both CENP-U-GFP and the CENP-U-T78A-GFP mutant were able to rescue the chromosome misalignment defect caused by depletion of endogenous CENP-U (Fig 4O; new data), which is in line with our previous study (Chen et al., 2021).

2) Due to the instability of purified full-length CENP-U the authors used GST fragments spanning the N terminus of CENP-U for their analysis. Additionally, in the crystallization experiments only a small fragment of CENP-U is purified along with fragments of RAD21 and STAG2. CENP-U functions as part of a larger complex comprised of CENP-O, CENP-P, CENP-Q, and CENP-R. This complex then assembles as part of the larger Constitutive Centromere associated network at centromeric DNA. Similarly, RAD21 and STAG2 also function as part of larger protein assemblies that include SMC1, SMC3. Although the authors provide evidence that the expressed CENP-U fragments can interact with fragments of RAD21 and STAG2, these experiments cannot recapitulate the behavior or orientation of these proteins when part of their larger interaction networks. Although I recognize that technical challenges limit the ability to test these full interactions, these caveats should be discussed and highlighted in both the results and discussion sections.

Response: As shown above in the response to the first reviewer, in the revised manuscript, we provide a line of strong evidence indicating that the CENP-U and CENP-Q sub-complex binds to the Scc1-SA2 sub-complex. These new data are now shown in Fig EV2G, Fig EV2H, Fig EV2I and J, Fig EV3G. Importantly, we further show that stably expressed CENP-U-GFP, but not the CENP-U-ADA-GFP mutant, co-immunoprecipitated endogenous Cohesin subunits Scc1, SA2, SMC1 and SMC3 (Fig 4B; new data). This strongly implies the interaction between the CENP-OPQUR complex and the Cohesin complex in cells.

As suggested by this reviewer, we stated in the Results section that “We failed to bacterially purify full-length proteins of CENP-O, CENP-P, CENP-Q, CENP-R, and CENP-U, which prevented us from determining whether the whole CENP-OPQUR complex directly interacts with the Scc1-SA2 sub-complex *in vitro*.”, as well as in the Discussion section that “Future studies are required to overcome the technical challenges to reconstitute the whole complexes of CENP-OPQUR and Cohesin to elucidate more interaction details *in vitro*.”

3) In Figure 1 and Supplementary Figure S1, the authors identify chromosome alignment defects when CENP-U is perturbed in the presence of MG132. The authors go on to state that "CENP-U is required to maintain chromosome alignment on the metaphase plate". This statement should be altered to indicate that this "during a prolonged mitosis" as CENP-U is not required for chromosome alignment in normal mitotic cells.

Response: Good suggestion. We state in the revised manuscript that “Thus, CENP-U is required to maintain chromosome alignment on the metaphase plate during prolonged

metaphase arrest.”

4) In Figure 1 and Supplementary Figure S1, the authors show that CENP-U knockdown cells have increased incidence of PSSC when incubated in MG132. From this data the authors then state that "CENP-U is necessary for the maintenance of sister chromatid cohesin at mitotic centromeres". This statement should be revised as this function is not shown in this manuscript, as there is no direct quantification of cohesin turnover or loss in CENP-U knockdown cells. Instead, the data shown indicate a phenotype of cohesin fatigue in situations of prolonged mitosis.

Response: Good suggestion. In the revised manuscript, we concluded for Fig 1 and EV1 that “Therefore, CENP-U promotes the strength of centromeric cohesion, which is important to counteract the metaphase spindle pulling forces to prevent premature loss of sister-chromatid cohesion.”

5) The authors show that, during a prolonged mitosis, CENP-U is required to protect against cohesin fatigue. To achieve prolonged mitosis, the authors rely on MG132, a proteasome inhibitor. As this will alter protein degradation globally, it is important to consider whether the observed behavior a consequence of aberrant protein turnover? For example, is the inability to properly turnover WAPL altered, resulting in an increase in WAPL activity? Or is this behavior observed in all situations of prolonged mitosis? The authors should test additional mitotic perturbations, for example incubation in nocodazole or KIF11-inhibited cells, and test if CENP-U depletion still results in PSSC.

Response: Good suggestions. In the mitosis research field, the proteasome inhibitor MG132 is widely used to induce metaphase arrest by inhibiting the anaphase-promoting complex/Cyclosome (APC/C)-dependent degradation of the Cdk1 activator Cyclin B and the Separase inhibitor Securin.

In the revised manuscript, we confirmed that MG132 treatment for up to 8 h did not affect the protein levels of Wapl, Scc1 and SMC3 in both control HeLa cells and CENP-U-depleted cells (Fig 5A; new data).

As suggested by this reviewer, we further assessed the effect of CENP-U depletion on sister-chromatid cohesion when cells were delayed in metaphase by treatment with Apcin, a small molecule which binds to the APC/C activator Cdc20 and competitively inhibits the E3 ligase activity of APC/C (Sackton et al, 2014). After Apcin treatment for 6 h, 26.6-29.5% and 75.9-83.5% mitotic cells underwent sister chromatid separation in control HeLa cells and CENP-U-depleted cells, respectively (Fig 1F-I; new data). Thus, CENP-U depletion causes a defect in maintaining sister-chromatid cohesion during metaphase delay which is induced by inhibition of either proteasome or the APC/C activity.

Given that CENP-U is a constitutive inner kinetochore protein, we wondered whether CENP-U depletion would loosen centromeric cohesion. For this purpose, we measured the inter-kinetochore (inter-KT) distance between sister kinetochores of chromosome spreads prepared from HeLa cells which were briefly arrested in mitosis with the

microtubule destabilizer nocodazole, a condition where the PSCS was less obvious in both control cells and CENP-U depleted cells when compared to MG132 treatment. As shown in Fig 1J-L, the inter-KT distance was around 12% further apart in CENP-U depleted cells than in control cells, indicative of weakened centromeric cohesion.

6) The authors provide multiple experiments to support their hypothesis that CENP-U interacts directly with cohesin. These experiments are important to the manuscript as a whole, but in the main figures complicated the story. For example, I found the biochemical experiments in Figure 3 to be convincing particularly due to the mutant that eliminates the interaction providing a clear control. In contrast, the experiments in Figure 2 were much less convincing and are overwhelming and very hard to follow. To simplify the manuscript for the reader, I recommend moving many of supportive figures to the supplement or eliminating unnecessary and redundant experiments to focus on the critical pieces of data.

Response: Thanks for the suggestions. In the revised manuscript, we have re-organized Fig 2 and EV2. More specifically, we moved original figures of Fig 2A and C to the current Fig EV2E and F. Moreover, we repeated some experiments to confirm that, while knockdown of SMC3 or SMC1 moderately reduced the binding of Scc1 and SA2 to GST-CENP-U (1-60) (Fig 2B and C; new figure), knockdown of Scc1 or SA2 prevented GST-CENP-U (1-60) from binding SMC1 and SMC3 (Fig 2D and E; new figure). These results indicate that CENP-U associates with the core Cohesin complex in a manner dependent on Scc1 and SA2, but not on SMC1 and SMC3. As a result, the data in Fig 2 become more concise and convincing.

7) Throughout the manuscript and in all figures the authors need to make clear that the cells are incubated in the presence of the proteasome inhibitor MG132 and that the phenotypes are only observed in situations of prolonged mitosis - in both the figure itself, in the text, and in the legends

Response: Good suggestion. We have included all such information in the text, in the figure, and in the figure legends throughout the revised manuscript.

8) Table S1 was not included with manuscript

Response: Table EV1 is now included in the submitted materials.

9) Figure 1C. The misaligned chromosomes are difficult to see. I recommend adjusting the images to be in greyscale rather than blue.

Response: We have now adjusted the images as recommended in the revised manuscript (Fig 1C).

10) Figure 2G. No band is shown in the control GST-CENPU lane for STAG2. Why is this?

Response: Actually, as shown below, in the control GST-CENP-U (1-100) in original Fig 2G, there is a weak band for SA2 which is unfortunately very close to the bottom of the excised membrane.

Regardless, we repeated this experiment to confirm that knockdown of SA2 prevented GST-CENP-U (1-60) from binding SMC1 and SMC3 (Fig 2D and E; new figure).

Referee #3:

This study reports the role of the inner kinetochore component CENP-U in regulation of centromeric cohesion through an interaction between the N-terminal region of CENP-U and an interface formed by SCC1-SA2 components of cohesin. This surface had been identified 10 years ago by the Yu lab as responsible for the competition of WAPL and SGO1 for binding cohesin and therefore critical for protection of a fraction of cohesin from the prophase pathway (Hara 2014 NSMB). More recently, the same surface was also identified by the Rowland-Panne labs as key for the interaction between cohesin and CTCF (Li 2020 Nature). In this last publication, the authors further proposed the existence of additional interactors for this surface using peptide arrays. CENP-U is one of them, consistent with findings described in the current manuscript. (By the way, this should be mentioned either in Introduction or in Discussion)

Here, SA2 and SCC1 are found in pull downs from human cell extracts using as bait a 100-aa fragment of CENP-U and the authors demonstrate that this interaction contributes to strengthen centromeric cohesion in mitosis. The manuscript is well written and identifies a new role for the CENP-OPQUR complex in regulation of cohesion. Overall, the biochemical and immunofluorescence data are solid and convincingly support the conclusions. My major criticism is that the relevance of this mechanism for mitotic cohesion is unclear since:

- (1) most protein interaction data are obtained with overexpressed proteins or in vitro;
- (2) proper comparison of the consequences of CENP-U knock down and Sgo1 knock down is missing (Sgo1 being currently considered the major mechanism ensuring mitotic cohesion through protection of centromeric cohesin from Wapl mediated release).

I have some suggestions to strengthen these aspects of the manuscript:

1. Most biochemical data in the paper come from incubations of a GST-bound CENP-U fragment and cell extracts. These data are in general solid and appropriately controlled, and provide important information about the physical interactions between the corresponding proteins. However, it would be important to detect some interactions with endogenous proteins using antibodies against CENP-U or even in the GFP-CENPU cell line.

Response: Good comments. As mentioned above, due to low abundance of endogenous CENP-U protein in cells, we cannot find an antibody for immunoblotting or immunoprecipitation of endogenous CENP-U, which prevented us from determining whether endogenous CENP-U interacts with endogenous Scc1-SA2 in cells. Importantly, in the revised manuscript, we showed that, upon stable expression in HeLa cells, CENP-U-GFP, but not the CENP-U-ADA-GFP mutant, co-immunoprecipitated endogenous Cohesin subunits Scc1, SA2, SMC1 and SMC3 (Fig 4B; new data). This strongly implies the interaction between the CENP-OPQR complex and the Cohesin complex in cells.

Also, in Figure 2L, lysates prepared from HEK-293T cells transiently expressing Myc-SA2, Scc1-GFP or both, are subjected to pull-down with GST or GST-CENP-U (1-60), and only when both are present, they are able to bind to GST-CENPU beads. Since the extract has endogenous proteins, I assume that overexpression of both cohesin components is required to detect the interaction and that it happens in the absence of the rest of the complex. Is this correct? I would like to see both blots for SCC1 and SA2 in Figures 2E-H.

Response: Good point. First of all, I assume that, in this reviewer's comments, "Figure 2L" is actually Fig 2J, and "Figures 2E-H" are actually Fig 2D-G in the original manuscript.

We have repeated the experiment for original Fig 2J for three times. Data in Fig 2H (New Figure) and 2C clearly showed that GST-CENP-U (1-60) pulled down Scc1-GFP and Myc-SA2 only when they were co-expressed, indicating that overexpression of both Cohesin components (Scc1 and SA2) is required to detect the binding of the Scc1-GFP and Myc-SA2 sub-complex to GST-CENP-U (1-60). These data seem a little odd at a glance. However, we obtained very similar results from three independent experiments. Moreover, Consistent with these *in vitro* biochemical data, we further showed that, while tethering EGFP-LacI-Scc1 to the LacO repeats moderately recruited SFB-tagged CENP-U, co-expression of Myc-SA2 with EGFP-LacI-Scc1 caused a strong recruitment of SFB-CENP-U to the LacO repeats (Fig 2F and G). We therefore added a new sentence in the revised manuscript that "The failure of GST-CENP-U (1-60) in binding Scc1-GFP or Myc-SA2 when exogenously expressed alone suggest that endogenous SA2 and Scc1 in the cell lysates exist as a sub-complex with a molecular ratio of 1:1, leaving little endogenous SA2 and Scc1 available for binding Scc1-GFP and Myc-SA2, respectively."

In addition, as recommended by this reviewer we showed both blots for Scc1 and SA2 in the revised manuscript (Fig 2B-E; new figure).

2. Does CENP-U bind cohesin in the context of a CENP-OPQR complex? Does knock down of other component of the complex result in mitotic cohesion defects?

Response: Good points. As shown above in the response to the other reviewers, in the revised manuscript, we provide a line of strong evidence indicating that the CENP-U and CENP-Q sub-complex binds to the Scc1-SA2 sub-complex. **These new data are now shown in Fig EV2G, Fig EV2H, Fig EV2I and J, Fig EV3G.** Importantly, we further show that stably expressed CENP-U-GFP, but not the CENP-U-ADA-GFP mutant, co-immunoprecipitated endogenous Cohesin subunits Scc1, SA2, SMC1 and SMC3 (**Fig 4B; new data**). This strongly implies the interaction between the CENP-OPQR complex and the Cohesin complex in cells.

Moreover, in line with the fact that CENP-Q and CENP-U form a sub-complex within the CENP-OPQR complex in cells (Pesenti et al., 2022; Tian et al., 2022; Yatskevich et al., 2022), as well as the CENP-Q-dependent localization of CENP-U at kinetochores (Chen et al., 2021; Hori et al., 2008), we showed in the revised manuscript that knockdown of CENP-Q in CENP-U-depleted cells did not cause a further defect in maintaining metaphase sister-chromatid cohesion (**Fig EV2I and J; new data**). These data imply that CENP-Q indirectly strengthens sister-chromatid cohesion through binding CENP-U that interacts with Scc1-SA2.

3. The authors assume that a Sgo1 mutant unable to interact with phosphoH2A (K492A) and therefore not properly targeted to centromeres, is equivalent to a loss of Sgo1. However, defects after Sgo1 knock down are likely much stronger than those observed after CENP-U knock down, which require prolonged metaphase arrest. The Yu lab showed that there is still some Sgo1 at centromeres, and quite a lot along the chromosome arms, in this mutant background. They also showed that ectopic expression of this mutant after Sgo1 knock down rescued mitotic cohesion to a large extent (Liu 2013 Curr Biol).

The authors should include siSgo1 and double siSgo1 siCENP-U conditions in the experiments shown in Figure 8. Only in this way the relevance of a Sgo1-independent pathway can be assessed. Showing Sgo1 staining and, if possible, cohesin staining in all these conditions would provide important clues to understand regulation of mitotic cohesion and the contribution of CENP-U to this pathway.

Response: As suggested by this reviewer, we performed additional experiments and stated in the Results section as shown below:

“In early mitosis, Sgo1 predominantly localized to centromeres, whereas some Sgo1 was also distributed on the chromosome arm (Fig EV5A; new data). This is in line with previous studies (Kitajima et al., 2005; McGuinness et al., 2005; Nakajima et al., 2007), as well as the observations that small amounts of Cohesin is present on mitotic chromosome arms (Chu et al., 2020; Gimenez-Abian et al., 2004), and that Sgo1 can bind Cohesin (Hara et al., 2014; Liu et al., 2013b).

We next examined the effect of CENP-U knockdown on sister chromatid cohesion in

Sgo1-depleted cells. As expected, Sgo1 depletion by siRNA caused a dramatic loss of sister-chromatid cohesion (Fig EV5B; new data), reflecting an important role for Sgo1 in protecting Cohesin both on chromosome arms and at centromeres (Nakajima et al., 2007). Interestingly, cells co-depleted of CENP-U and Sgo1 showed a further defect in sister-chromatid cohesion when compared to cells depleted of Sgo1 alone (Fig EV5B; new data), indicating that CENP-U and Sgo1 additively promote sister-chromatid cohesion.

As previously reported (Kawashima et al, 2010; Liu et al, 2015), when cells enter mitosis, the spindle checkpoint protein Bub1-mediated histone H2A threonine-120 phosphorylation (H2ApT120) at centromeres directly recruits Sgo1. Upon chromosome biorientation on the metaphase plate, the H2ApT120 signal is largely reduced following Bub1 release from kinetochores, resulting in Sgo1 delocalization from centromeres (Fig EV5C; new data). We then assessed the contribution of CENP-U to sister-chromatid cohesion when Sgo1 is incapable of binding H2ApT120. Using CRISPR/Cas9 genome editing, we previously obtained HeLa-derived cell lines expressing the H2ApT120-binding-deficient Sgo1-K492A mutant (Liang et al, 2019), which cannot localize to mitotic centromeres to protect centromeric cohesion (Liu et al, 2013a; Liu et al., 2015). In line with our previous observation (Liang et al., 2019), cells expressing Sgo1-K492A were defective in maintaining sister-chromatid cohesion during metaphase arrest induced by MG132 treatment for 2-4 h (Fig 8A and B). Importantly, depletion of CENP-U resulted in a further increase in the percentage of Sgo1-K492A cells with PSCS, demonstrating a role for CENP-U in maintaining sister-chromatid cohesion when Sgo1 is delocalized from centromeres. Moreover, CENP-U knockdown caused a 20.9% increase in the inter-KT distance in nocodazole-arrested mitotic cells expressing the Sgo1-K492A mutant (Figs 8C, EV5D and E; new data). This indicates that CENP-U is required to maintain centromeric cohesion in cells lacking centromeric Sgo1.”

We concluded in the last paragraph of the Results section that “Taken together, these data indicate that, while Sgo1 plays a major role in protecting sister-chromatid cohesion along the whole chromosomes, CENP-U specifically promotes centromeric cohesion.”

Moreover, we stated in the Discussion section that “The CENP-OPQUR complex-bound pool of Cohesin at the inner kinetochore is distinct from the Sgo1-bound pool of Cohesin at centromeres and on chromosome arms (Gimenez-Abian et al., 2004; Liang et al., 2019; Liu et al., 2013a; Liu et al., 2015; Liu et al., 2013b; Nakajima et al., 2007). While CENP-U protects Cohesin specifically at the inner kinetochore region, Sgo1 plays a prominent role in protecting Cohesin both on chromosome arms and at centromeres. Our finding that the role for CENP-U in maintaining sister-chromatid cohesion is better revealed in delayed metaphase is in line with the observations that a small amount of Cohesin is detected on chromosome arms and that sister chromatids remain associated along the arms until anaphase onset (Chu et al., 2020; Gimenez-Abian et al., 2004; Hirano, 2015; Nakajima et al., 2007; Rieder & Cole, 1999).”

4. The authors should include western blots showing the remaining levels of proteins after siRNA for all experiments with knock downs. In Figure 4, we should be able to compare endogenous levels and those of ectopically expressed CENP-U WT and mutant proteins.

Response: These good suggestions in principle. As mentioned above, CENP-U, which is exclusively localized at the inner kinetochore, is a low abundant protein in cells. Unfortunately, we cannot find an antibody for immunoblotting of endogenous CENP-U, which prevented us from showing the remaining levels of proteins after siRNA. Actually, we carried out quantitative RT-PCR assays and immunofluorescence staining assays to confirm the knockdown of CENP-U at the mRNA level and protein level, respectively. **These new data are now shown in Fig EV1D-F.**

Besides, we prepared rabbit anti-CENP-U polyclonal antibodies by immunization with the synthetic peptide EPNVKETYDSSSLP. This antibody does not work well in immunoblotting assay, and can only weakly detected CENP-U-GFP stably expressed in HeLa cells. The blots for CENP-U-GFP using this antibody are now shown in Figs 4A and EV4A.

A brief overview of how/where we have incorporated the various clarifications in the previous response letter last November

Referee #2:

2) Due to the instability of purified full-length CENP-U the authors used GST fragments spanning the N terminus of CENP-U for their analysis. Additionally, in the crystallization experiments only a small fragment of CENP-U is purified along with fragments of RAD21 and STAG2. CENP-U functions as part of a larger complex comprised of CENP-O, CENP-P, CENP-Q, and CENP-R. This complex then assembles as part of the larger Constitutive Centromere associated network at centromeric DNA. Similarly, RAD21 and STAG2 also function as part of larger protein assemblies that include SMC1, SMC3. Although the authors provide evidence that the expressed CENP-U fragments can interact with fragments of RAD21 and STAG2, these experiments cannot recapitulate the behavior or orientation of these proteins when part of their larger interaction networks. Although I recognize that technical challenges limit the ability to test these full interactions, these caveats should be discussed and highlighted in both the results and discussion sections.

Response: We now state in the Discussion section that “Future studies are required to overcome the technical challenges to reconstitute the whole complexes of CENP-OPQUR and Cohesin to elucidate more interaction details *in vitro*.”

3) In Figure 1 and Supplementary Figure S1, the authors identify chromosome alignment defects when CENP-U is perturbed in the presence of MG132. The authors go on to state that "CENP-U is required to maintain chromosome alignment on the metaphase plate". This statement should be altered to indicate that this "during a prolonged mitosis" as CENP-U is not required for chromosome alignment in normal mitotic cells.

Response: We now state in the second paragraph of the first section of Results that “Thus, CENP-U is required to maintain chromosome alignment on the metaphase plate during prolonged metaphase arrest”.

4) In Figure 1 and Supplementary Figure S1, the authors show that CENP-U knockdown cells have increased incidence of PSSC when incubated in MG132. From this data the authors then state that "CENP-U is necessary for the maintenance of sister chromatid cohesin at mitotic centromeres". This statement should be revised as this function is not shown in this manuscript, as there is no direct quantification of cohesin turnover or loss in CENP-U knockdown cells. Instead, the data shown indicate a phenotype of cohesin fatigue in situations of prolonged mitosis.

Response: We now state in the fourth paragraph of the first section of Results that “Thus, CENP-U depletion causes a defect in maintaining sister-chromatid cohesion during metaphase delay which is induced by inhibition of either proteasome or the APC/C activity”.

Moreover, we state in the end of the first section of Results that “Therefore, CENP-U promotes the strength of centromeric cohesion, which is important to counteract the metaphase spindle pulling forces to prevent premature loss of sister-chromatid cohesion”.

8) Table S1 was not included with manuscript

Response: Table S1 was actually included in the uploaded files, but probably not in the way that is obvious for the reviewer to download. This table is now included as recommended.

9) Figure 1C. The misaligned chromosomes are difficult to see. I recommend adjusting the images to be in greyscale rather than blue.

Response: We have now adjusted the images as recommended (Figure 1C).

Referee #3:

Also, in Figure 2L, lysates prepared from HEK-293T cells transiently expressing Myc-SA2, Scc1-GFP or both, are subjected to pull-down with GST or GST-CENP-U (1-60), and only when both are present, they are able to bind to GST-CENPU beads. Since the extract has endogenous proteins, I assume that overexpression of both cohesin components is required to detect the interaction and that it happens in the absence of the rest of the complex. Is this correct? I would like to see both blots for SCC1 and SA2 in Figures 2E-H.

Response: We note in the Results section of the revised manuscript that “The failure of GST-CENP-U (1-60) in binding Scc1-GFP or Myc-SA2 when exogenously expressed alone suggest that endogenous SA2 and Scc1 in the cell lysates exist as a sub-complex with a molecular ratio of 1:1, leaving little endogenous SA2 and Scc1 available for binding Scc1-GFP and Myc-SA2, respectively”.

3. The authors assume that a Sgo1 mutant unable to interact with phosphoH2A (K492A) and therefore not properly targeted to centromeres, is equivalent to a loss of Sgo1. However, defects after Sgo1 knock down are likely much stronger than those observed after CENP-U knock down, which require prolonged metaphase arrest. The Yu lab showed that there is still some Sgo1 at centromeres, and quite a lot along the chromosome arms, in this mutant background. They also showed that ectopic expression of this mutant after Sgo1 knock down rescued mitotic cohesion to a large extent (Liu 2013 Curr Biol).

The authors should include siSgo1 and double siSgo1 siCENP-U conditions in the experiments shown in Figure 8. Only in this way the relevance of a Sgo1-independent pathway can be assessed. Showing Sgo1 staining and, if possible, cohesin staining in all these conditions would provide important clues to understand regulation of mitotic cohesion and the contribution of CENP-U to this pathway.

Response: As suggested by this reviewer, we performed additional experiments. Based on the

data, we conclude in the last paragraph of the Results section that “Taken together, these data indicate that, while Sgo1 plays a major role in protecting sister-chromatid cohesion along the whole chromosomes, CENP-U specifically promotes centromeric cohesion”.

Moreover, we state in the Discussion section that “The CENP-OPQUR complex-bound pool of Cohesin at the inner kinetochore is distinct from the Sgo1-bound pool of Cohesin at centromeres and on chromosome arms (Gimenez-Abian et al., 2004; Liang et al., 2019; Liu et al., 2013a; Liu et al., 2015; Liu et al., 2013b; Nakajima et al., 2007). While CENP-U protects Cohesin specifically at the inner kinetochore region, Sgo1 plays a prominent role in protecting Cohesin both on chromosome arms and at centromeres. Our finding that the role for CENP-U in maintaining sister-chromatid cohesion is better revealed in delayed metaphase is in line with the observations that a small amount of Cohesin is detected on chromosome arms and that sister chromatids remain associated along the arms until anaphase onset (Chu et al., 2020; Gimenez-Abian et al., 2004; Hirano, 2015; Nakajima et al., 2007; Rieder & Cole, 1999)”.

A brief overview of how/where we have incorporated the various “new figures” in the previous response letter last November

- ✓ New Figure 1 is now incorporated as Figure EV3B.
- ✓ New Figure 2 is now incorporated as Figure EV2G.
- ✓ New Figure 3 is now incorporated as Figure EV3G.
- ✓ New Figure 4 is now incorporated as Figure EV2H.
- ✓ New Figure 5 is now incorporated as Figure 4B from an independent experiment.
- ✓ New Figure 6 is now incorporated as Figure EV1D.
- ✓ New Figure 7 is now incorporated as Figure EV1E-1F.
- ✓ New Figure 8 is now incorporated as Figure 4G.
- ✓ New Figure 9 is now incorporated as Figure 4H-4K.
- ✓ New Figure 10 is now incorporated as Figure 4O.
- ✓ New Figure 11 is now incorporated as Figure 5A.
- ✓ New Figure 12 is now incorporated as Figure 1F-1I.
- ✓ New Figure 13 is now incorporated as Figure 2E.
- ✓ New Figure 14 is now incorporated as Figure 2C.
- ✓ New Figure 15 is now incorporated as Figure 2B.
- ✓ New Figure 16 is now incorporated as Figure 2D.
- ✓ New Figures 17 and 18 are now incorporated as Figure EV2J-2J.
- ✓ New Figure 19 is now incorporated as Figure EV5B.
- ✓ New Figure 20 is now incorporated as Figures 8C, and EV5D-E.

Additional Point-by-point response for EMBOJ-2023-115677R

REFEREE 1

They have responded to most concerns I raised:

Point 1

The maps are significantly improved. From inspection of the composite omit map the authors provided, the fitting of Leu42 to Pro47 looks correct.

Response: Many thanks.

Point 2

This reference should be cited.

Response: Good suggestion. This reference (Li et al., Nature, 2020, PMID: 31905366) is now cited six times in the revised manuscript, 5 times in the Results sections and 1 time in the Discussion section.

Point 3

The authors need to determine quantitatively that the CENP-U peptide, and ideally the whole CENP-U subunit or the CENP-OPQUR sub-complex, binds to SA2-SCC1.

>Author Response: Actually, quantification of peptide arrays performed by the Rowland-Panne labs showed that the Scc1-SA2 complex binds to the CENP-U peptide (PIDVDFDFDNS) with Kd of $4.03 \pm 0.22 \mu\text{M}$ (Li et al., Nature, 2020, PMID: 31905366)

This is an inadequate response.

>In line with this, we found that bacterially expressed GST-fused human CENP-U protein in the forms of full-length (amino acid residues 1-418) and fragments encompassing residues 101-418 and 201-418 were unstable (Figure S2A-S2C). This technical obstacle prevented us from quantitatively determining the binding of whole CENP-U subunit to Scc1-SA2.

The OPQUR complex can be expressed and purified - see:

(Pesenti ME et al., Mol Cell, 2022, PMID: 35525244; Yatskevich S et al., Science, 2022, PMID: 35420891).

However the authors do show data that would indicate CENP-U in the context of CENP-OQUR would bind Scc1-SA2.

Response: The fo-fc map shows unambiguously that the CENP-U peptide

(PIDVFDFPDNS) is present in our crystal structure of the Scc1 (281-420)-SA2 (80-1060) complex. In line with this, the Rowland-Panne labs used peptide arrays to show that the Scc1-SA2 complex binds to the CENP-U peptide (PIDVFDFPDNS) with K_d of $4.03 \pm 0.22 \mu\text{M}$ (Extended Data Table 2, Li et al., Nature, 2020, PMID: 31905366). In the revised manuscript, we state in the “Results” section that “This is in line with a previous study which used peptide arrays to show that a CENP-U peptide containing the DVFDF motif bound to Scc1-SA2 in vitro with a K_d of around $4.0 \mu\text{M}$ (Li et al., 2020).”

We agree that it would be ideal to quantitatively determine the whole CENP-U subunit, or the CENP-OPQR sub-complex, binds to SA2-Scc1. However, given the intrinsic property of the subunits of the constitutive centromere-associated network (CCAN) at the inner centromere, it is technically difficult to purify the whole CENP-U subunit and to reconstitute the CENP-OPQR sub-complex in vitro, as noted by the Andrea Musacchio lab that “CENP-O, -P, -Q, and -U were unstable when expressed individually in bacteria or insect cells and could not be recovered in soluble form (unpublished data)” (Pesenti M et al., Mol Cell, 2018, PMID: 30174292).

We thank this reviewer for appreciating that we do show data that would indicate CENP-U in the context of CENP-OPQR and CENP would bind Scc1-SA2. In the revised manuscript, we provide important new data to further support our conclusion that CENP-U in the context of CENP-OPQR or CENP-QU binds Scc1-SA2.

1) Stably expressed CENP-U-GFP, but not the CENP-U-ADA-GFP mutant, co-immunoprecipitated endogenous Scc1, SA2, SMC1 and SMC3 (Fig 4B; new data). This strongly implies the interaction between the CENP-OPQR complex and the Cohesin complex in cells.

2) Pull-down assays showed that GST-CENP-U (1-100) bound to Scc1 but not SFB-CENP-Q in HEK-293T cell lysates. In sharp contrast, GST-CENP-U (101-418) pulled down SFB-CENP-Q but not Scc1. Thus, CENP-U uses N-terminus and C-terminal region to bind the Scc1-SA2 sub-complex and CENP-Q, respectively. These new data are now shown in Fig EV2G.

3) Moreover, SFB-CENP-Q did not bind to the GST-Scc1-SA2 sub-complex when expressed alone. Interestingly, co-expression with CENP-U-GFP, but not CENP-U (1-100)-GFP, enabled the interaction. We confirmed that both CENP-U-GFP and CENP-U (1-100)-GFP were efficiently pulled down by GST-Scc1-SA2. These results indicate that CENP-Q indirectly binds Scc1-SA2 through interacting with the C-terminal region of CENP-U. These new data are now shown in Fig EV2H.

4) In line with the CENP-Q-dependent localization of CENP-U at kinetochores (Chen et al., 2021; Hori et al, 2008), knockdown of CENP-Q in CENP-U-depleted cells did not cause a further defect in maintaining metaphase sister-chromatid cohesion, implying that CENP-Q indirectly strengthens sister-chromatid cohesion through binding CENP-U that interacts with Scc1-SA2. These new data are now shown in Fig EV2I and J.

5) SFB-CENP-Q was pulled down by GST-Scc1-SA2 when co-expressed with CENP-U-GFP, but not the CENP-U-ADA-GFP mutant, further conforming the indirect association of CENP-Q with the Scc1-SA2 sub-complex. **These new data are now shown in Fig EV3G.**

Point 4

>Of course, we can also omit the AlphaFold prediction model from the manuscript and simply cite the Garcia-Nieto paper.

Yes that would be more appropriate.

Response: As suggested, we have now omitted the AlphaFold prediction model from the revised manuscript, and cited the Garcia-Nieto paper in the Result section that “Thus, the YNF motif mediates Sgo1 interaction with the binding interface between Scc1 and SA2 in vitro and in cells, as recently reported (Garcia-Nieto et al, 2023).”

Point 5

Ok

Response: Many thanks.

Point 6

Ok

Response: Many thanks.

REFEREE 3

I have had a look at the response by the authors. Some of my queries have been addressed and I think overall the manuscript has improved with the new data.

Response: Many thanks.

My feeling is that the interaction is real and probably meaningful to prevent cohesion fatigue but the way the manuscript is written places the role of CENP-U at the level of Sgo1 and it is clearly not right.

Response: Based on data shown in the revised manuscript, including new data which will be mentioned later in this response, we conclude in the last paragraph of the Results section of the revised manuscript that **“Taken together, these data indicate**

that, while Sgo1 plays a major role in protecting sister-chromatid cohesion along the whole chromosomes, CENP-U specifically promotes centromeric cohesion.”

We further state in the Discussion section that “While CENP-U protects Cohesin specifically at the inner kinetochore region, Sgo1 plays a prominent role in protecting Cohesin both on chromosome arms and at centromeres.”

Defects with CENP-U KD only appear after mitotic arrest. Data in new Figure 19 show a small effect of CENP-U KD over Sgo1 KD. This is probably one single replica, and it is unclear to me if the difference shown is biologically relevant. The difference between the two siRNA against CENP-U is larger than the increase they provide over Sgo1 KD. (By the way, I do not understand why this experiment is not done as the rest in the paper, using MG132 instead of nocodazole).

Response: In our understanding, the relationship the relative contributions of CENP-U and Sgo1 to sister chromatid cohesion is only a minor point of this manuscript, which actually is not even mentioned in the Abstract of the manuscript. More specifically, the comparison of the cohesion defects after siRNA-mediated depletion of CENP-U and Sgo1 to sister chromatid cohesion is less meaningful, given that CENP-U exclusively localizes to the inner kinetochore whereas Sgo1 localizes to the whole chromosome as well as accumulates at centromeres in mitosis. Instead, it is more meaningful to compare the cohesion defects after depletion of CENP-U and delocalization of Sgo1 at mitotic centromeres by inhibiting the Bub1-H2ApT120 signaling axis.

Regardless, we have performed additional experiments, and show new data in the revised manuscript to address these points:

1) We show that, in early mitosis, Sgo1 predominantly localizes to centromeres, whereas some Sgo1 is also distributed on the chromosome arm (Fig EV5A; new data). This is in line with previous studies (Kitajima et al, 2005; McGuinness et al., 2005; Nakajima et al., 2007), as well as the observations that small amounts of Cohesin is present on mitotic chromosome arms (Chu et al, 2020; Gimenez-Abian et al., 2004), and that Sgo1 can bind Cohesin (Hara et al., 2014; Liu et al, 2013b).

2) We also examined the effect of CENP-U knockdown on sister chromatid cohesion in Sgo1-depleted cells. As expected, Sgo1 depletion by siRNA caused a dramatic loss of sister-chromatid cohesion (Fig EV5B; new data; Means and standard deviations from three independent experiments are shown), reflecting an important role for Sgo1 in protecting Cohesin both on chromosome arms and at centromeres (Nakajima et al., 2007). Interestingly, cells co-depleted of CENP-U and Sgo1 showed a further defect in sister-chromatid cohesion when compared to cells depleted of Sgo1 alone, indicating that CENP-U and Sgo1 additively promote sister-chromatid cohesion.

3) As previously reported (Kawashima et al, 2010; Liu et al, 2015), when cells enter mitosis, the spindle checkpoint protein Bub1-mediated histone H2A

thereonine-120 phosphorylation (H2ApT120) at centromeres directly recruits Sgo1. Upon chromosome biorientation on the metaphase plate, the H2ApT120 signal is largely reduced following Bub1 release from kinetochores, resulting in Sgo1 delocalization from centromeres (Fig EV5C; new data). We then assessed the contribution of CENP-U to sister-chromatid cohesion when Sgo1 is incapable of binding H2ApT120. Using CRISPR/Cas9 genome editing, we previously obtained HeLa-derived cell lines expressing the H2ApT120-binding-deficient Sgo1-K492A mutant (Liang et al, 2019), which cannot localize to mitotic centromeres to protect centromeric cohesion (Liu et al, 2013a; Liu et al., 2015). In line with our previous observation (Liang et al., 2019), cells expressing Sgo1-K492A were defective in maintaining sister-chromatid cohesion during metaphase arrest induced by MG132 treatment for 2-4 h (Fig 8A and B). Importantly, depletion of CENP-U resulted in a further increase in the percentage of Sgo1-K492A cells with PSCS, demonstrating a role for CENP-U in maintaining sister-chromatid cohesion when Sgo1 is delocalized from centromeres.

4) Moreover, CENP-U knockdown caused a 20.9% increase in the inter-KT distance in nocodazole-arrested mitotic cells expressing the Sgo1-K492A mutant (Figs 8C, EV5D and E; new data). This indicates that CENP-U is required to maintain centromeric cohesion in cells lacking centromeric Sgo1.

Finally, we treated Sgo1-depleted cells with the spindle destabilizer nocodazole instead of the proteasome MG132 (Figure EV5B) to avoid the influence on cohesion of the spindle microtubule pulling forces executed at kinetochores, given that Sgo1 depletion reduces the centromeric localization of Aurora B kinase that may consequently alter the kinetochore-microtubule attachment status.

In any case, Sgo1 staining (and if possible cohesin staining although I know this is complicated) in spreads and in mitotic cells would shed some light on the mechanisms to support what they now write: "Our data indicate that CENP-U plays a role in maintaining centromeric cohesion during metaphase prior to anaphase onset, whereas Sgo1 is mainly required for the protection of centromeric cohesion during prometaphase".

Response: As suggested, we performed Sgo1 staining for chromosome spreads (Fig EV5A; new data) and mitotic cells (Fig EV5C; new data), which show that Sgo1 predominantly localized to centromeres, whereas some Sgo1 was also distributed on the chromosome arm. This is in line with previous studies (Kitajima et al, 2005; McGuinness et al., 2005; Nakajima et al., 2007), as well as the observations that small amounts of Cohesin is present on mitotic chromosome arms (Chu et al, 2020; Gimenez-Abian et al., 2004), and that Sgo1 can bind Cohesin (Hara et al., 2014; Liu et al, 2013b).

Again, as mentioned above, we conclude in the last paragraph of the Results section of the revised manuscript that "Taken together, these data indicate that, while

Sgo1 plays a major role in protecting sister-chromatid cohesion along the whole chromosomes, CENP-U specifically promotes centromeric cohesion.”

Why does the cell need two different proteins using the same SA2/Scc1 surface to compete out Wapl?

Response: Good point. Learning from our data, cells need two different proteins, i.e. Sgo1 and CENP-U, using the same SA2-Scc1 interface to compete out Wapl because Sgo1 localizes to the whole chromosome and enriches at centromeres during mitosis, whereas CENP-U exclusively localizes to the inner kinetochore throughout the cell cycle.

We therefore conclude in the last paragraph of the Results section of the revised manuscript that “Taken together, these data indicate that, while Sgo1 plays a major role in protecting sister-chromatid cohesion along the whole chromosomes, CENP-U specifically promotes centromeric cohesion.”

Moreover, we state in the Discussion section that “While CENP-U protects Cohesin specifically at the inner kinetochore region, Sgo1 plays a prominent role in protecting Cohesin both on chromosome arms and at centromeres.”

We also state in the Discussion section that “The CENP-OPQR complex-bound pool of Cohesin at the inner kinetochore is distinct from the Sgo1-bound pool of Cohesin at centromeres and on chromosome arms (Gimenez-Abian et al., 2004; Liang et al., 2019; Liu et al., 2013a; Liu et al., 2015; Liu et al., 2013b; Nakajima et al., 2007)”.

Indeed, a small amount of Cohesin is detected on chromosome arms and that sister chromatids remain associated along the arms until anaphase onset (Chu et al., 2020; Gimenez-Abian et al., 2004; Hirano, 2015; Nakajima et al., 2007; Rieder & Cole, 1999).

Is this because during prolonged metaphase arrest the amount of Sgo1 at centromeres decreases significantly?

Response: I think this is indeed the case. As previously reported (Kawashima et al, 2010; Liu et al, 2015), when cells enter mitosis, the spindle checkpoint protein Bub1-mediated histone H2A threonine-120 phosphorylation (H2ApT120) at centromeres directly recruits Sgo1. Upon chromosome biorientation on the metaphase plate, the H2ApT120 signal is largely reduced following Bub1 release from kinetochores, resulting in Sgo1 delocalization from centromeres (Fig EV5C; new data). Therefore, during prolonged metaphase arrest the amount of Sgo1 at centromeres does decrease significantly, which renders the contribution of CENP-U to sister chromatid cohesion better revealed in this situation.

The data in new Figure 5 are promising but they should be improved by showing input (what %?) next to IP fractions, so we can have an idea of the efficiency, and also adding blots for SMC1 or SMC3 to support an interaction with the whole cohesin complex, and with some other component of the CENP-OPQUR complex.

Response: Good suggestions. We stably expressed CENP-U-GFP (wild-type and the ADA mutant) in HeLa cells, and found that CENP-U-GFP, but not the CENP-U-ADA-GFP mutant, co-immunoprecipitated endogenous Scc1, SA2, SMC1 and SMC3 (Fig 4B; new data). This strongly implies the interaction between the CENP-OPQUR complex and the Cohesin complex in cells. As suggested, we show input (%) next to the IP fractions, as well as the blots for Smc1 and Smc3, in the revised manuscript. The results imply that only a small fraction of Cohesin is associated with CENP-U, which is very reasonable given that Cohesin is abundant in cells and associates with the chromosome genome-wide whereas CENP-U exclusively localizes to the inner kinetochore.

Also, I understand that the authors do not have an antibody that allows them to work with endogenous proteins, but maybe they could edit the endogenous CENP-U locus to add a tag.

Response: Thanks for understanding the unavailability of antibodies that work for proteins such as CENP-U. Regarding the suggestion to edit endogenous CENP-U to add a tag by CRISPR-Cas9, this requires the existence of appropriate Cas9-specific PAM sequence 5'-NGG-3' (where "N" can be any nucleotide base) at the genomic DNA region in proximity to the stop codon. This actually is one of the biggest technical obstacles that prevent the universal application of the CRISPR-Cas9.

Prof. Fangwei Wang
Zhejiang University
Life Sciences Institute
866 Yuhangtang Rd
Nano Building Rm 577
Hangzhou, Zhejiang 310058
China

21st Mar 2024

Re: EMBOJ-2023-115677R

A non-canonical role for the inner kinetochore in regulating sister-chromatid cohesion at centromere

Dear Fangwei,

Thank you again for resubmitting a revised version of your manuscript. I have now carefully checked the incorporation of the new data and clarifications presented to the referees as part of your appeal, as well as the additional changes made in response to the referees' feedback at that stage. I am happy to say that we should now be able to accept the paper, as soon as several remaining editorial points have also been addressed:

- First of all, my colleague Hannah Sonntag, in charge of source data curation, will still contact you with a detailed list of original data that would need to be provided in the present case.
- On the abstract page of the manuscript, please include 4-5 general keyword terms to enhance searchability.
- Please rename the Conflict of Interest section into "Disclosure and Competing Interests Statement", in accordance with our updated Guide to Authors (<https://www.embopress.org/competing-interests/>)
- As we are switching from a free-text author contribution statement towards a more formal statement based on Contributor Role Taxonomy (CRediT) terms, please remove the present Author Contribution section and instead specify each author's contribution(s) directly in the Author Information page of our submission system during upload of the final manuscript. See <https://casrai.org/credit/> for more information.
- Please convert Table EV1 into an Expanded View Dataset - naming "Dataset EV1", and move its legend from the main text into a separate tab of the XLSX spreadsheet file.
- Please rename Table EV2 (the X-ray statistics) into Table EV1. Again, its legend should be removed from the main text and only be present in the Table file.
- Please also cut the EV movie legends from the main text, instead placing each one into one separate legend text file per EV movie; then move each legend file together with the respective movie file into a separate ZIP archive before re-uploading as "Movie EV1/2/3..."
- Please provide suggestions for a short 'blurb' text prefacing and summing up the conceptual aspect of the study in two sentences (max. 250 characters), followed by 3-5 one-sentence 'bullet points' with brief factual statements of key results of the paper; they will form the basis of an editor-written 'Synopsis' accompanying the online version of the article. Please also provide an altered synopsis image, making sure that the aspect ratio conforms to our website's format - it should be exactly 550 pixels wide and between 300-600 pixels high.
- Finally, during routine pre-acceptance checks, our data editors have raised the following queries regarding figures, data, and legends, which I would ask you to address (ideally using the Track Changes option):
 - * Please note that $n=2$ in figures 1d, k; 5f; 8c: in these cases, no statistical tests can be applied and not error bars can be drawn, but individual data points have to be plotted
 - * Although 'n' is provided, please describe the nature of entity for 'n' in the legends of figures 1b, f, 4d-e, h, m; 5b, 8a; EV 2i; EV 4c; EV 5b.
 - * Please note that a separate 'Data Information' section is required at the end of the legends for figures 1b-c, e-g, i-j, l; 4c-f, h-i, k, m-n; 5b-c, e, g; 6d-g; 7a-b; 8a-b, i-j; EV 1c-f; Ev 3c-g; EV 4b, d; EV 5a-e.
 - * Please indicate the statistical test used for data analysis in the legends of figures 1f; 2g; 3i; 4e; 4h, j, m; 5b, e; 6e, g, 8j, EV 1f; EV 2i; EV 4c; EV 5b, d.
 - * Please note that the white arrows are not defined in the legend of figure 2f; 3h; 6e, g; 7a-b, 8i; EV 3e-f. This needs to be rectified.

* Please note that the black arrows are not defined in the legend of figure Ev 2c-d. This needs to be rectified.

I am therefore returning the manuscript to you for a final round of revision, to allow you to make these modifications and upload the revised files, after which we can hopefully proceed with formal acceptance and production of the manuscript.

With best regards,

Hartmut

9) Digital image enhancement is acceptable practice, as long as it accurately represents the original data and conforms to community standards. If a figure has been subjected to significant electronic manipulation, this must be clearly noted in the figure legend and/or the 'Materials and Methods' section. The editors reserve the right to request original versions of figures and the original images that were used to assemble the figure. Finally, we generally encourage uploading of numerical as well as gel/blot image source data; for details see: embopress.org/page/journal/14602075/authorguide#sourcedata

At EMBO Press, we ask authors to provide source data for the main manuscript figures. Our source data coordinator will contact you to discuss which figure panels we would need source data for and will also provide you with helpful tips on how to upload and organize the files.

Further information is available in our Guide For Authors:

In the interest of ensuring the conceptual advance provided by the work, we recommend submitting a revision within 3 months (19th Jun 2024). Please discuss the revision progress ahead of this time with the editor if you require more time to complete the revisions. Use the link below to submit your revision:

Link Not Available

Prof. Fangwei Wang
Zhejiang University
Life Sciences Institute
866 Yuhangtang Rd
Nano Building Rm 577
Hangzhou, Zhejiang 310058
China

12th Apr 2024

Re: EMBOJ-2023-115677R1

A non-canonical role of the inner kinetochore in regulating sister-chromatid cohesion at centromeres

Dear Fangwei,

Thank you for submitting your final revised manuscript for our consideration. I am pleased to inform you that we have now accepted it for publication in The EMBO Journal.

With kind regards,

Hartmut
